# Bandit Learning with Delayed Impact of Actions

**Wei Tang†, Chien-Ju Ho†, and Yang Liu\***

†Washington University in St. Louis, \*University of California, Santa Cruz

{w.tang, chienju.ho}@wustl.edu, yangliu@ucsc.edu

## Abstract

We consider a stochastic multi-armed bandit (MAB) problem with delayed impact of actions. In our setting, actions taken in the past impact the arm rewards in the subsequent future. This delayed impact of actions is prevalent in the real world. For example, the capability to pay back a loan for people in a certain social group might depend on historically how frequently that group has been approved loan applications. If banks keep rejecting loan applications to people in a disadvantaged group, it could create a feedback loop and further damage the chance of getting loans for people in that group. In this paper, we formulate this delayed and long-term impact of actions within the context of multi-armed bandits. We generalize the bandit setting to encode the dependency of this "bias" due to the action history during learning. The goal is to maximize the collected utilities over time while taking into account the dynamics created by the delayed impacts of historical actions. We propose an algorithm that achieves a regret of $\tilde{\mathcal{O}}(KT^{2/3})$ and show a matching regret lower bound of $\Omega(KT^{2/3})$, where $K$ is the number of arms and $T$ is the learning horizon. Our results complement the bandit literature by adding techniques to deal with actions with long-term impacts and have implications in designing fair algorithms.

## 1 Introduction

Algorithms have been increasingly involved in high-stakes decision making. Examples include approving/rejecting loan applications [23, 37], deciding on employment and compensation [5, 19], and recidivism and bail decisions [1]. Automating these high-stakes decisions has raised ethical concerns on whether it amplifies the discriminative bias against protected classes [52, 14]. There have also been growing efforts towards studying algorithmic approaches to mitigate these concerns. Most of the above efforts have focused on static settings: a utility-maximizing decision maker needs to ensure her actions satisfy some fairness criteria at the decision time, without considering the long-term impacts of actions. However, in practice, these decisions may often introduce long-term impacts to the rewards and well-beings for the human agents involved. For example,

- A regional financial institute may decide on the fraction of loan applications from different social groups to approve. These decisions could affect the development of these groups: The capability of applicants from a group to pay back a loan might depend on the group's socio-economic status, which is influenced by how frequently applications from this group have been approved [6, 18].

- The police department may decide on the amount of patrol time or the probability of patrol in a neighborhood (primarily populated with a demographic group). The likelihood to catch a crime in a neighborhood might depend on how frequent the police decides to patrol this area [28, 26].

These observations raise the following concerns. If being insensitive with the long-term impact of actions, the decision maker risks treating a historically disadvantaged group unfairly. Making things even worse, these unfair and oblivious decisions might reinforce existing biases and make it harder to observe the true potential for a disadvantaged group. While being a relatively under-

explored (but important) topic, several recent works have looked into this problem of delayed impact of actions in algorithm design. However, these studies have so far focused on understanding the impact in a one-step delay of actions [45, 36, 31], or a sequential decision making setting without uncertainty [47, 33, 51, 66, 46, 20, 67].

Our work departs from the above line of efforts by studying the long-term impact of actions in sequential decision making under uncertainty. We generalize the multi-armed bandit setting by introducing the *impact functions* that encode the dependency of the "bias" due to the action history of the learning to the arm rewards. Our goal is to learn to maximize the rewards obtained over time, in which the rewards' evolution could depend on the past actions.

The history-dependency reward structure makes our problem substantially more challenging. In particular, we first show that applying standard bandit algorithms leads to linear regret, i.e., existing approaches will obtain low rewards with a biased learning process. To address this challenge, under relatively mild conditions for the dependency dynamics, we present an algorithm, based on a phased-learning template which smoothes out the historical bias during learning, that achieves a regret of $\tilde{\mathcal{O}}(KT^{2/3})$. Moreover, we show a matching lower regret bound of $\Omega(KT^{2/3})$ that demonstrates that our algorithm is order-optimal. Finally, we conduct a series of simulations showing that our algorithms compare favorably to other state-of-the-art methods proposed in other application domains. From a policy maker's point of view, our paper explores solutions to learn the optimal sequential intervention when the actions taken in the past impact the learning environment in an unknown and long-term manner. We believe our work nicely complements the existing literature that focuses more on the "understanding" of the dynamics [33, 45, 66, 67].

**Related work.** Our work contributes to algorithmic fairness studied in sequential settings. Prior works either study fairness in sequential learning settings without considering long-term impact of actions [34, 49, 27, 7, 30, 54] or explore the delayed impacts of actions with focus on addressing the one-step delayed impacts or sequential learning with full information [33, 45, 6, 31, 51, 18, 20]. Our work differs from the above and studies delayed impacts of actions in sequential decision making under uncertainty. Our formulation bears similarity to reinforcement learning since our impact function encodes memory (and is in fact Markovian [53, 62]), although we focus on studying the exploration-exploitation tradeoff in bandit formulation. Our learning formulation builds on the rich bandit learning literature [42, 3] and is related to non-stationary bandits [60, 8, 9, 43, 38]. Our techniques share similar insights with Lipschitz bandits [39, 59] and combinatorial bandits [13] in that we also assume the Lipschitz reward structure and consider combinatorial action space. There are also recent works that have formulated delayed action impact in bandit learning [56, 38], but in all of these works, the setting and the formulation are different from the ones we consider in the present work. More discussions on related work can be found in Appendix A.

## 2 Problem Setting

We formulate the setting in which an institution sequentially determines how to allocate resource to different groups. For example, a regional financial institute may decide on overall frequency of loan applications to approve from different social groups. The police department may decide on the amount of patrol time allocated to different regions.

The institution is assumed to be a utility maximizer, aiming to maximize the expected reward associated with the allocation policy over time. If we assume the reward[1] for allocating a unit of resource to a group is i.i.d. drawn from some unknown distribution, this problem can be reduced to a standard bandit problem, with each group representing an *arm*. The goal of the institution is then to learn a sequence of arm selections to maximize its cumulative rewards.

In this work, we extend the bandit setting and consider the delayed impact of actions. Below we formalize our setup which introduces *impact functions* to bandit framework.

**Action space.** There are $K$ *base arms*, indexed from $k = 1$ to $K$, with each base arm representing a group. At each discrete time $t$, the institution chooses an action, called a *meta arm*, which specifies

---

[1]The reward could be whether a crime has been stopped or whether the borrower pays the monthly payment on time. For applications that require longer time periods to assess the rewards, the duration of a time step, i.e., the frequency to update the policy, would also need to be adjusted accordingly.

the probability to activate each base arm. Let $\mathcal{P} = \Delta([K])$ be the $(K-1)$-dimensional probability simplex. We denote the meta arm as $\mathbf{p}(t) = \{p_1(t), \ldots, p_K(t)\} \in \mathcal{P}$. Each base arm $k$ is activated independently according to their $p_k(t)$ in $\mathbf{p}(t)$. The institution only observes the reward from the arms that are activated. Our feedback model deviates slightly from the classical bandit feedback and shares similarity to combinatorial bandits: instead of assuming always observing one arm's reward each time, we observe the reward of one arm in expectation, i.e., we can potentially observe no arm's reward or multiple arms' rewards. This modeling choice is mainly needed to resolve a technicality issue and has been adopted in the literature [65, 13]. Most of our algorithms and results extend to the case where only one arm is activated according to $\mathbf{p}(t)$.[2]

**Remark 2.1.** *We can also interpret the meta-arm as specifying the proportion of resources allocated to each base arm. The interpretation impacts the way the rewards are generated (i.e., instead of observing the rewards of the realized base arms, the institution observes the rewards of all base arms with non-zero allocations.) Our analysis utilizes the idea of importance weighting and could deal with both cases in the same framework. To simplify the presentation, we focus on the case of interpreting the meta-arm as probabilities, though our results apply to both interpretations.*

**Delayed impacts of actions.** We consider the scenario in which the rewards of actions are unknown a priori and are influenced by the action history. Formally, let $\mathcal{H}(t) = \{\mathbf{p}(s)\}_{s \in [t]}$ be the action history at time $t$. We define the *impact function* $\mathbf{f}(t) = F(\mathcal{H}(t))$ to summarize the impact of the learner's actions to the reward generated in each group, where $F(\cdot)$ is the function mapping the action history to its current impact on arms' rewards. In the following discussion, we make $F(\cdot)$ implicit and use the vector $\mathbf{f}(t) = \{f_1(t), \ldots, f_K(t)\}$ to denote the impact to each group, where $f_k(t)$ captures the impact of action history to arm $k$.

**Rewards and regret.** The reward for selecting group $k$ at time $t$ depends on both $p_k(t)$ and the historical impact $f_k(t)$. In particular, when the arm representing group $k$ is activated, the institution observes a reward (the instantaneous reward is bounded within $[0, 1]$) drawn i.i.d. from a unknown distribution with mean $r_k(f_k(t)) \in [0, 1]$ and claims the sum of rewards from activated arms as total rewards. $r_k(\cdot)$ is unknown a priori but is Lipschitz continuous (with known Lipschitz constant $L_k \in (0, 1]$) with respect to its input, i.e., a small deviation of the institution's actions has small impacts on the unit reward from each group. When action $\mathbf{p}(t)$ is taken at time $t$, the institution obtains an expected reward

$$U_t(\mathbf{p}(t)) = \sum_{k=1}^{K} p_k(t) \cdot r_k(f_k(t)). \tag{1}$$

As for the impact function, we focus on the setting in which $\mathbf{f}(t)$ is a time-discounted average, with each component $f_k(t)$ defined as

$$f_k(t) = \frac{\sum_{s=1}^{t} p_k(s) \gamma^{t-s}}{\sum_{s=1}^{t} \gamma^{t-s}}, \tag{2}$$

where $\gamma \in [0, 1)$ is the time-discounting factor. [3] Intuitively, $f_k(t)$ is a weighted average with more weights on recent actions. We would like to highlight that our results extend to a more general family of impact functions and do not require the exact knowledge of impact functions (see discussion in Section 5.2). We also note that when $\gamma = 0$, our setting reduces to a special case where the impact function only depends on the current action $p_k(t)$ (*action dependent*), instead of the entire history of actions (discounted by 0 right away). We study this special case of interest in Section 4.

Let $\mathcal{A}$ be the algorithm the institution deploys. The goal of $\mathcal{A}$ is to choose a sequence of actions $\{\mathbf{p}(t)\}$ that maximizes the total utility. The performance of $\mathcal{A}$ is characterized by regret, defined as

$$\mathrm{Reg}(T) = \sup_{\mathbf{p} \in \mathcal{P}} \sum_{t=1}^{T} U_t(\mathbf{p}) - \mathbb{E}\left[\sum_{t=1}^{T} U_t(\mathbf{p}(t))\right], \tag{3}$$

where the expectation is taken on the randomness of algorithm $\mathcal{A}$ and the utility realization.[4]

---

[2]The upper bound becomes $\tilde{\mathcal{O}}(K^{4/3}T^{2/3})$, which is slightly worse than $\tilde{\mathcal{O}}(KT^{2/3})$ in $K$. But the upper bound is still tight in $T$. We provide discussions in Remark D.6.

[3]Here we follow the tradition to define $0^0 = 1$ when $\gamma = 0$.

[4]In this paper, we adopt the standard regret definition and compare against the optimal fixed policy. Another possible regret definition is to compare against the optimal dynamic policy that could change based on the

## 2.1 Exemplary Application of Our Setup

We provide an illustrative example to instantiate our model. Consider a police department who needs to dispatch a number of police officers to $K$ different districts. Each district has a different crime distribution, and the goal (absent additional fairness constraints) might be to maximize the number of crimes caught [22].[5] The effects of police patrol resource allocated to each district may aggregate over time and then impact the crime rate of that district. In other words, the crime rate in each district depends on how frequently the police officers have been dispatched historically in this district.

To simplify the discussion, we normalize the expected police resource to be one unit. Each district $k$ has a default average crime rate $\overline{r}_k \in (0,1)$ at the beginning of the learning process. This crime rate can (at most) be decreased to $\underline{r}_k \in (0, \overline{r}_k)$. All of these are unknown to the police department. The police department makes a resource allocation decision at each time step. We use $r_k(t) \in (0,1)$ to denote the crime rate in district $k$ at time $t$, taking into account the impact of historical decisions. Assume $p_k(t)$ is the amount of police resource dispatched to district $k$ at time $t$ ($\sum_k p_k(t) = 1$ for all $t$), the expected number of crimes caught at district $k$ at time $t$ would be $p_k(t)r_k(t)$. Note that here $p_k(t)$ can be interpreted as the probability of allocating police resource (randomly sending the patrol team to each of the $K$ districts) or the fraction of allocated police resource.

Below we provide one natural example of the interaction between the impact function and the reward. At time step $t+1$, let $\mathcal{H}_k(t) := \{p_k(1), \ldots, p_k(t)\}$ denote the historical decisions of the police department for district $k$. Now given $\mathcal{H}_k(t+1) = \{\mathcal{H}_k(t) \cup p_k(t+1)\}$ where $p_k(t+1)$ is the current decision for district $k$, assume that the crime rate at time $t+1$ in district $k$ is in the following form:

$$r_k(t+1) = \overline{r}_k - f_k(\mathcal{H}_k(t+1)) \times (\overline{r}_k - \underline{r}_k), \tag{4}$$

where $f_k(\cdot) : [0,1]^t \to [0,1]$ is the impact function that summarizes how historical actions would impact the current crime rate. One possible example is $f_k(\mathcal{H}_k(t)) = \frac{\sum_{s=1}^{t} p_k(s)\gamma^{t-s}}{\sum_{s=1}^{t} \gamma^{t-s}}$ as we defined in Equation (2). This impact function has two natural properties:

- When $f_k(\mathcal{H}_k(t)) = 1$ (e.g., $p_k(s) = 1, \forall s \leq t$), the police department keeps dispatching the police officers to district $k$ with probability 1, then district $k$ will reach its lowest crime rate.
- When $f_k(\mathcal{H}_k(t)) \to 0$ (e.g., $p_k(s) \to 0, \forall s \leq t$), the police department rarely dispatch police officers to district $k$, The crime rate in district $k$ will reach its highest level.

In this example, treating each district as an arm and directly applying standard bandit algorithms might reach suboptimal solutions since the reward dynamic is not considered. In this paper, we develop algorithms that can take into account this history-dependent reward dynamic and achieve no-regret learning. Our results hold for a general class of impact functions (under mild conditions) and do not need to assume the exact knowledge of the impact function.

## 3 Overview of Main Results

We summarize our main results in this section. First, we present an important, though perhaps not surprising, negative result: if the institution is not aware of the delayed impact of actions, applying existing standard bandit algorithms in our setting leads to linear regrets. This negative result highlights the importance of designing new algorithms when delayed impact of actions are present. The formal statement and analysis are in Appendix C.

**Lemma 3.1** (Informal). *If the institution is unaware of the delayed impact of actions, applying standard bandit algorithms (including UCB, Thompson Sampling) leads to linear regrets.*

While the negative result might not be surprising, as it resembles similarity to the negative results on applying classic bandit algorithm to a non-stationary setting, it points out the need to design new algorithms for settings with delayed impact of actions. The key challenge introduced by our setting is in estimating the arm rewards: when pulling the same meta arm at different time steps, the institution

---

history. However, calculating the optimal dynamic policy in our setting is nontrivial as it requires to solve an MDP with continuous states.

[5]As discussed by Elzayn et al. [22], there might be other goals besides simply catching criminals, including preventing crime, fostering community relations, and promoting public safety. We use the same goal they adopted for the illustrative purpose.

does not guarantee to obtain rewards drawn from the targeted distribution according to the chosen meta arm, as the arm reward depends on the impact function $\mathbf{f}(t)$. To address this challenge, we note that if the institution keeps pulling the same meta-arm repeatedly, the impact function (and thus the arm reward associated with the meta-arm) would converge to some value. This observation leads to our approaches. We first develop a bandit algorithm that works with impacts that converge "immediately" (or equivalently only depend on "immediate" actions, echoing the case with $\gamma = 0$ in Equation (2)). We then propose a phased-learning reduction template that reduces our general setting to the above one and achieves a sublinear regret.

**Theorem 3.2** (Informal). *There is an algorithm that achieves an optimal regret bound $\tilde{\mathcal{O}}(KT^{2/3})$ for the bandit problem with the impact function defined in Equation (2). In addition, there is a matching lower bound of $\Omega(KT^{2/3})$.*

To provide an overview of our approaches, we start with *action-dependent bandits* (Section 4), where the impact at time $t$ depends only on the action at $t$, i.e., $\mathbf{f}(t) = \mathbf{p}(t)$, namely $\gamma = 0$ in Equation (2). This setting not only captures the one-step impact but also offers a backbone for the phase-learning template for the general history-dependent scenario. In this setting, when a meta-arm $\mathbf{p} = \{p_1, \ldots, p_K\}$ is selected, each base arms $k$ is activated with probability $p_k$, and the institution observes the realized rewards for all activated base arms and receives the sum of them as total rewards. Since we know the probability $p_k$ for activating each base arm, we may apply importance weighting to simulate the case as if the learner is selecting $K$ probabilities and obtain $K$ signals at each time step. This interpretation transforms our problem structure to a setting similar to combinatorial bandits. Furthermore, since both $r_k(\cdot)$ are Lipschitz continuous, we adopt the idea from Lipschitz bandits to discretize the continuous space of each $p_k$. With these ideas combined, we design a UCB-like algorithm that achieves a regret of $\mathcal{O}(KT^{2/3}(\ln T)^{1/3})$.

With the solution of action-dependent bandits, we explore the general *history-dependent bandits* with impact functions following Equation (2) (Section 5). The main idea is to divide total time rounds into phases, and then selecting the same actions in each phase to smooth out impacts of historically made actions, which will then help reduce the problem to an action-dependent one. One challenge is to construct appropriate confidence bound and adjust the length of each phase to account for the historical action bias. With a careful combination with our results for action-dependent bandits, we present an algorithm which can also achieve a regret of the order $\tilde{\mathcal{O}}(KT^{2/3})$. We further proceed to show that this bound is tight and provide numerical experiments.

## 4 Action-Dependent Bandits

In this section, we study action-dependent bandits, in which the impact function $\mathbf{f}(t) = \mathbf{p}(t)$, corresponding to $\gamma = 0$ in Equation (2). Our algorithm starts with a discretization over the space $\mathcal{P}$. Formally, we uniformly discretize $[0, 1]$ for each base arm into intervals of a fixed length $\epsilon$, with carefully chosen $\epsilon$ such that $1/\epsilon$ is an positive integer.[6] Let $\mathcal{P}_\epsilon$ be the space of discretized meta arms, i.e., for each $\mathbf{p} = \{p_1, \ldots, p_K\} \in \mathcal{P}_\epsilon$, $\sum_{k=1}^{K} p_k = 1$ and $p_k \in \{\epsilon, 2\epsilon, \ldots, 1\}$ for all $k$. Let $\mathbf{p}_\epsilon^* := \sup_{\mathbf{p} \in \mathcal{P}_\epsilon} \sum_{k=1}^{K} p_k \cdot r_k(p_k)$ denote the optimal strategy in discretized space $\mathcal{P}_\epsilon$. After a meta arm $\mathbf{p}(t) = \{p_1(t), \ldots, p_K(t)\} \in \mathcal{P}_\epsilon$ is selected, each arm $k$ is *independently* activated with probability $p_k(t)$. From now, we use $\tilde{r}_t(\cdot)$ to denote the realization of corresponding reward. The learner observes activated arms, and observes the instantaneous reward $\tilde{r}_t(p_k(t))$ of each activated arm $k$. We use importance weighting [29] to construct the unbiased realized reward for each of the $K$ elements in $\mathbf{p}$:

$$\widehat{r}_t(p_k(t)) = \begin{cases} \tilde{r}_t(p_k(t))/p_k(t), & \text{arm } k \text{ is activated} \\ 0. & \text{arm } k \text{ is not activated} \end{cases} \tag{5}$$

Since the probability activating arm $k$ is $p_k(t)$, it is easy to see that $\mathbb{E}[\widehat{r}_t(p_k(t))] = \mathbb{E}[\tilde{r}_t(p_k(t))]$. Given the importance-weighted rewards $\{\widehat{r}_t(p_k(t))\}$, we re-frame our problem as choosing a $K$-dimensional probability measure (one value for each base arm). In particular, for each base arm $k$, $p_k$ will take the value from $\{\epsilon, 2\epsilon, \ldots, 1\}$, and we refer to $p_k$ as the *discretized arm*.

**Remark 4.1.** *The above importance-weighting technique enables us to "observe" samples of $r_k(p_k)$ for all base arms $k$ when selecting $\mathbf{p} = \{p_1, \ldots, p_K\}$. This technique helps to bridge the gap between*

---

[6]Smarter discretization generally does not lead to better regret bounds [39].

*the interpretation of whether* **p** *is a probability distribution or an allocation over base arms. Our following techniques can be applied in either interpretation.*

---

**Algorithm 1** Action-Dependent UCB

---

1: **Input:** $K, \epsilon$
2: **Initialization:** For each discretized arm, play an arbitrary meta arm such that this discretized arm is included (if the selection of the arm is not realized, then simply initialize its reward to $0$; otherwise initialize it to the observed reward divided/reweighted by the selection probability).
3: **for** $t = \lceil K/\epsilon \rceil + 1, ..., T$ **do**
4:     Select $\mathbf{p}(t) = \arg\max_{\mathbf{p} \in \mathcal{P}_\epsilon} \text{UCB}_t(\mathbf{p})$ where $\text{UCB}_t(\mathbf{p})$ is defined as in (6).
5:     Arm $k$ is activated w.p. $p_k(t)$ and observe its realized reward $\tilde{r}_t(p_k(t))$.
6:     Update the importance-weighted rewards $\{\hat{r}_t(p_k(t))\}$ as in (5) and update the empirical mean $\{\bar{r}_t(p_k(t))\}$ for each base arm as in (6).
7: **end for**

---

By doing so, our problem is now similar to combinatorial bandits, in which we are choosing $K$ discretized arms and observe the corresponding rewards. Below we describe our UCB-like algorithm based on the reward estimation of discretized arms. We define the set $\mathcal{T}_t(p_k) = \{s \in [t] : p_k \in \mathbf{p}(s)\}$ to record all the time steps such that the deployed meta arm $\mathbf{p}(s)$ contains the discretized arm $p_k$. We can maintain the empirical estimates of the mean reward for each discretized arm and compute the UCB index for each meta arm $\mathbf{p} \in \mathcal{P}_\epsilon$:

$$\bar{r}_t(p_k) = \frac{\sum_{s \in \mathcal{T}_t(p_k)} \hat{r}_s(p_k)}{n_t(p_k)}, \quad \text{UCB}_t(\mathbf{p}) = \sqrt{\frac{K \ln t}{\min_{p_k \in \mathbf{p}} n_t(p_k)}} + \sum_{p_k \in \mathbf{p}} p_k \cdot \bar{r}_t(p_k), \quad (6)$$

where $n_t(p_k)$ is the cardinality of set $\mathcal{T}_t(p_k)$. With the UCB index in place, we are now ready to state our algorithm in Algorithm 1. The next theorem provides the regret bound of Algorithm 1.

**Theorem 4.2.** *Let* $\epsilon = \Theta\big((\ln T/T)^{1/3}\big)$. *The regret of Algorithm 1 (with respect to the optimal arm in non-discretized* $\mathcal{P}$*) is upper bounded as follows:* $\text{Reg}(T) = \mathcal{O}\big(KT^{2/3}(\ln T)^{1/3}\big)$.

*Proof Sketch.* Similar to the proofs of the family of UCB-style algorithms for MAB, after an appropriate discretization, we can derive the regret as the sum of the *badness* (suboptimality of a meta arm) for all (discretized) suboptimal meta arm selection. However, this will cost us an exponential $K$ in the order of final regret bound: this is because we need to take the summation over all feasible suboptimal meta arms, which the number grows exponentially with $K$. To tackle this challenge, we focus on the derivations of badness via tracking the *minimum* suboptimal selections in the space of realized actions (base arms), which enables us to reduce the exponential $K$ to a polynomial $K$. On a high level, our proof proceeds in the following steps:

- In Step 1, we obtain a high probability bound of the estimation error for the expected rewards of meta arms after discretization.

- In Step 2, we bound the probability on deploying a suboptimal meta arm when selected sufficiently many number of times, where we quantify such sufficiency via $\min_{p_k \in \mathbf{p}} n_t(p_k)$, which is the minimum number of selection of a discretized arm contained in a suboptimal meta arm.

- In Step 3 and Wrapping-up step, we bound the expected value of $\min_{p_k \in \mathbf{p}} n_t(p_k)$ and connect the regret for playing suboptimal meta arms $\mathbf{p}$ with the regret incurred by including discretized arms $p_k \notin \mathbf{p}_\epsilon^*$ which are not in optimal strategy (in discretized space).

Finally, the regret bound of Algorithm 1 can be achieved by optimizing the discretization parameter. □

---

**Discussions** Our techniques have close connections to Lipschitz bandits [16, 50] and combinatorial bandits [13, 12]. Given the Lipschitz property of $r_k(\cdot)$, we are able to utilize the idea of Lipschitz bandits to discretize the strategy space and achieve sublinear regret with respect to the optimal strategy in the non-discretized strategy space. Moreover, we achieve a significantly improved regret bound by utilizing the connection between our problem setting and combinatorial bandits. In combinatorial bandits, the learner selects $K$ actions out of action space $\mathcal{M}$ at each time step, where $|\mathcal{M}| = \Theta(K/\epsilon)$

in our setting. Directly applying state-of-the-art combinatorial bandit algorithms [13] in our setting would achieve an instance-independent regret bound of $\mathcal{O}\big(K^{3/4}T^{3/4}(\ln T)^{1/4}\big)$, while we achieve a lower regret of $\mathcal{O}\big(KT^{2/3}(\ln T)^{1/3}\big)$.[7] The reason for our improvement is that, for each base arm, regardless of which probability it was chosen, we can update the reward of the base arm, which provides information for all meta arms that select this arm with a different probability. This reduces the exploration and helps achieving the improvement. In addition to the above improvement, we would like to highlight that another of our main contributions is to extend the action-dependent bandits to the problem of history-dependent bandits, as discussed in Section 5.

Another natural attempt to tackle our problem is to apply EXP3 [4], which achieves sublinear regret even when the arm reward is generated adversarially. However, note that the optimal policy in our setting could be a mixed strategy, while the "sublinear" regret of EXP3 is with respect to a fixed strategy. Therefore, when applying EXP3 over the set of base arms, it still implies a linear regret in our setting. The other option is to apply EXP3 over the set of meta arms. Since the number of meta arms is exponential in $K$, it would incur a regret exponential in $K$ due to the size of meta arms.

## 5   History-Dependent Bandits

We now describe how to utilize our results for action-dependent bandits to solve the history-dependent bandit learning problem, with the impact function specified in Equation (2). The crux of our analysis is the observation that, in history-dependent bandits, if the learner keeps selecting the same strategy $\mathbf{p}$ for a long enough period of time, the expected one-shot utility will be approaching the utility of selecting $\mathbf{p}$ in the action-dependent bandits. More specifically, suppose after time $t$, the current action impact for all arms is $\mathbf{f}(t) = \mathbf{p}^{(\gamma)}(t) = \{p_1^{(\gamma)}(t), \dots, p_K^{(\gamma)}(t)\}$. Assume that the learner is interested in learning about the utility of selecting $\mathbf{p} = \{p_1, \dots, p_K\}$ next. Since the rewards are influenced by $\mathbf{f}(t)$, selecting $\mathbf{p}$ at time $t+1$ does not necessarily give us the utility samples at $U(\mathbf{p})$. Instead, the learner can keep pulling this meta arm for a non-negligible $s$ consecutive rounds to ensure that $\mathbf{f}(t+s)$ approaches $\mathbf{p}$. Following this idea, we decompose the total number of time rounds $T$ into $\lfloor T/L \rfloor$ phases which each phase is associated with $L$ rounds. We denote $m \in [1, \dots, \lfloor T/L \rfloor]$ as the phase index and $\mathbf{p}(m)$ as the selected meta-arm in the $m$-th phase. To summarize the above phased-learning template:

- In each phase $m$, we start with an *approaching stage*: the first $s_a$ rounds of the phase. This stage is used to "move" $\mathbf{f}(t+s)$ with $1 \le s \le s_a$ towards to $\mathbf{p}$.

- In the second stage, namely, *estimation stage*, of each phase: the remaining $L - s_a$ rounds. This stage is used for collecting the realized rewards and estimating the true reward mean on action $\mathbf{p}$.

- Finally, we leverage our tools in action-dependent bandits to decide what meta arm to select in each phase.

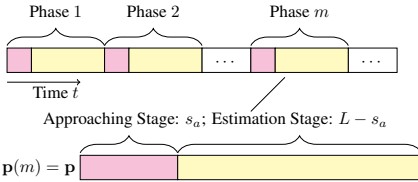

Fig. 1: We deploy $\mathbf{p}$ for all rounds in $m$-th phases, therefore, we use $\mathbf{p}(m) = \mathbf{p}$ to represent $\mathbf{p}(t) = \mathbf{p}$ for simplicity.

Note that even if we keep pulling the arm $k$ with the constant probability $p_k$ in the approaching stage, the action impact in the estimation stage is not exactly the same as meta arm we want to learn, i.e., $\mathbf{f}(t+s) \ne \mathbf{p}$ for $s \in (s_a, L]$, due to the finite length of the stage. However, we can guarantee all $\mathbf{f}(t+s)$ for $s \in (s_a, L]$ is close enough to $\mathbf{p}$ by bounding its approximation error w.r.t $\mathbf{p}$. The above idea enables a more general reduction algorithm that is compatible with any bandit algorithm that solves the action-dependent case. Let $\rho = (L - s_a)/L$ be the ratio of number of rounds in estimation stage of each phase. We present this reduction in Algorithm 2 and a graphical illustration in Figure 1.

### 5.1   History-Dependent UCB

In this section, we show how to utilize the reduction template to achieve a $\tilde{\mathcal{O}}(KT^{2/3})$ regret bound for history-dependent bandits. We first introduce some notations. For each discretized arm $p_k$, similar to action-dependent case, we define $\underline{\Gamma_m(p_k)} :=$

---

[7]We compare our results with a tight regret bound achieved in Theorem 2 of [13]. The detailed derivations are deferred to Appendix D.2.1.

---

**Algorithm 2** Reduction Template

---

1: **Input:** $K, T; \gamma, \epsilon, \rho \in (0, 1), s_a$.
2: **Input:** A bandit algorithm $\mathcal{A}$: History-Dependent UCB (Algorithm 3).
3: Split all rounds into consecutive phases of $L = s_a/(1 - \rho)$ rounds each.
4: **for** $m = 1, \dots$ **do**
5:      Query algorithm $\mathcal{A}$ for its meta arm selection $\mathbf{p}(m) = \mathbf{p}$.
6:      Each phase is separated into two stages:
        1). Approaching stage: $t = L(m - 1) + 1, \dots, L(m - 1) + s_a$;
        2). Estimation stage: $t = L(m - 1) + s_a + 1, \dots, Lm$.
7:      **for** $t = L(m - 1) + 1, \dots, L(m - 1) + s_a$ **do**
8:         Deploy the meta arm $\mathbf{p}$.
9:      **end for**
10:      **for** $t = L(m - 1) + s_a + 1, \dots, Lm$ **do**
11:         Deploy the meta arm $\mathbf{p}$;
12:         Collect the realized rewards $\tilde{r}_t$ of activated arms to estimate the mean reward as in (7).
13:      **end for**
14:      Update $\overline{U}_t^{\text{est}}(\mathbf{p})$ as in (7).
15: **end for**

---

$\left\{ s : \underline{s \in ((i - 1)L + s_a, iL]} \text{ where } p_k \in \mathbf{p}(i), \forall i \in [m] \right\}$ as the set of all time indexes till the end of phase $m$ in estimation stages such that arm $k$ is pulled with probability $p_k$. We define the following empirical $\bar{r}_m^{\text{est}}(p_k)$ computed from our observations and the empirical utility $\overline{U}_m^{\text{est}}(\mathbf{p})$: [8]

$$\bar{r}_m^{\text{est}}(p_k) = \frac{1}{n_m^{\text{est}}(p_k)} \sum_{s \in \Gamma_m(p_k)} \hat{r}_s(p_k^{(\gamma)}(s)), \quad \overline{U}_m^{\text{est}}(\mathbf{p}) = \sum_{p_k \in \mathbf{p}} p_k \cdot \bar{r}_m^{\text{est}}(p_k), \quad (7)$$

where $n_m^{\text{est}}(p_k) := |\Gamma_m(p_k)|$ is the total number of rounds pulling arm $k$ with probability $p_k$ in all estimation stages, and $\hat{r}_s(p_k^{(\gamma)}(s))$ is defined similarly as in Equation (5). We use the smoothed-out frequency $\{p_k^{(\gamma)}(s)\}_{s \in \Gamma_m(p_k)}$ in the estimation stage as an approximation for the discounted frequency right after the approaching stage.

We compute our UCB for each meta arm at the end of each phase. We define and compute $\text{err} := K\gamma^{s_a}(L^* + 1)$, the approximation error incurred after our attempt to smooth out the historical action impact. With these preparations, we present the phased history-dependent UCB algorithm (in companion with Algorithm 2) in Algorithm 3. The main result of this section is given as follows:

---

**Algorithm 3** History-Dependent UCB

---

1: Construct UCB for each meta arm $\mathbf{p} \in \mathcal{P}_\epsilon$ at the end of each phase $m = 1, 2, \dots$, as follows:

$$\text{UCB}_m(\mathbf{p}) = \overline{U}_m^{\text{est}}(\mathbf{p}) + \text{err} + 3\sqrt{\frac{K \ln(L\rho)}{\min_{p_k \in \mathbf{p}} n_m^{\text{est}}(p_k)}}.$$

2: Select $\mathbf{p}(m + 1) = \arg\max_{\mathbf{p}} \text{UCB}_m(\mathbf{p})$ with ties breaking equally.

---

**Theorem 5.1.** *For any constant ratio $\rho \in (0, 1)$ and $\gamma \in (0, 1)$, let $\epsilon = \Theta((\ln(T\rho)/(T\rho))^{1/3})$ and $s_a = \Theta(\ln(\epsilon^{1/3}/K)/\ln \gamma)$. The regret of Algorithm 2 with Algorithm 3 as input bounds as follows:*
$\text{Reg}(T) = \mathcal{O}\big(KT^{2/3}((\ln(T\rho))/\rho)^{1/3}\big).$

For a constant ratio $\rho$, we match the optimal regret order for action-dependent bandits. When $\gamma$ is smaller, the impact function "forgets" the impact of past-taken actions faster, therefore less rounds in approaching stage would be needed (see $s_a$'s dependence in $\gamma$) and this leads to larger $\rho$.

**Remark 5.2.** *The dependence of our regret on the phase length $L$ is encoded in $\rho$. When implementing our algorithm (Section 7), we calculate $L$ via $s_a$ given the ratio $\rho$. We also run simulations of our algorithm on different ratios $\rho$, the results show that the performance of our algorithm are not sensitive w.r.t. specifying $\rho$s - in practice, we do not require the exact knowledge of $\rho$, instead we can afford to use a rough estimation of its upper bound to compute $L$.*

---

[8] `est` in superscript stands for `est`timation stage.

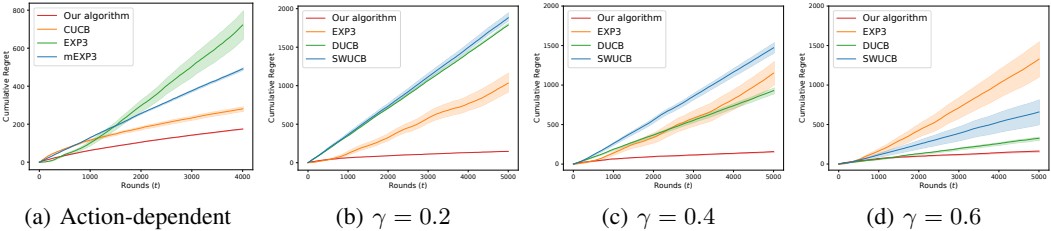

(a) Action-dependent    (b) $\gamma = 0.2$    (c) $\gamma = 0.4$    (d) $\gamma = 0.6$

Fig. 2: (a): Behavior of the different algorithms for action-dependent bandits. (b)-(d): Behavior of the different algorithms for history-dependent bandits on different $\gamma$.

## 5.2 Extension to General Impact Functions

So far, we discuss settings when the impact function is specified as in Equation (2). However, the same technique we presented earlier can be applied for a more general family of impact functions. In particular, as long as the impact function converges after the learner keeps selecting the same action, our result holds. To be more precise, we only require $\mathbf{f}(t)$ to satisfy the condition $|f_k(t+s) - g(p_k)| \leq \gamma^s, \gamma \in (0,1)$ when the learner keeps pulling arm $k$ with probability $p_k$ for $s$ round. The function $g(\cdot)$ can be an arbitrary monotone function as long as it is continuous and differentiable, for example: $g(x) = x$. In fact, the property of $\mathbf{f}(t)$ is only used when we estimate how close $\mathbf{f}$ is to $g(\mathbf{p})$ after the approaching stage with repeatedly selecting $\mathbf{p}$. For a different $\mathbf{f}(t)$, we define new reward mean functions $r'_k(\cdot) = r_k(g(\cdot))$, and tune parameters $\epsilon$ and $s_a$ accordingly to bound the approximation error for $\left| U(\mathbf{p}) - \bar{U}_m^{\text{est}}(\mathbf{p}) \right|$ (change the Lipschitz constant). This way we can follow the same algorithmic template to achieve a similar regret.

Moreover, we do not require exact knowledge of the impact function $\mathbf{f}(t)$. We only require the impact functions to satisfy the above conditions for our algorithms/analysis to hold. With the same arguments, while we assume the reward function $r_k(\cdot)$ is fed with the same impact function $\mathbf{f}$, our formulation generalizes to different impact functions for $r_k(\cdot)$, as long as these impact functions are able to stabilize given a consecutive adoption of the desired action.

## 6 Matching Lower Bounds

For both action- and history-dependent bandit learning problems, we have proposed algorithms that achieve a regret bound of $\tilde{\mathcal{O}}(KT^{2/3})$. We now show the above bounds are order-optimal with respect to $K$ and $T$, i.e., the lower bounds of our action- and history-dependent bandits are both $\Omega(KT^{2/3})$, as summarized below.

**Theorem 6.1.** *Let $T > 2K$ and $K \geq 4$, there exist problem instances that for our action- and history-dependent bandits, respectively, the regret for any algorithm $\mathcal{A}$ follows:* $\inf_{\mathcal{A}} \text{Reg}(T) \geq \Omega(KT^{2/3})$.

For the lower bound proof of action-dependent bandits (included in Appendix F), we following the standard randomized problem instances construction used in combinatorial bandits and Lipschitz bandits and use information inequality to prove the lower bound. For history-dependent bandits, we show that for a general class of reward function $r_k(\cdot)$ which satisfies the *proper* property (see Definition G.1), solving history-dependent bandits is as least as hard as solving action-dependent bandits. Armed with the above derived lower bound of action-dependent bandits, we can then conclude the lower bound of history-dependent bandits.

## 7 Numeric Experiments

We conducted a series of simulations to understand the performance of our algorithms. The detailed setups and discussion are in Appendix I. We first compare our algorithm with some baselines under action-dependent bandits and other non-stationary baselines under history-dependent bandits with different $\gamma$ (the parameter in time-discounted frequency). The results, as shown in Fig. 2, demonstrate that our proposed algorithm consistently outperforms the baseline methods. We also note that the performance of our algorithm is relatively robust w.r.t. difference choices of $\gamma$, i.e., the time-

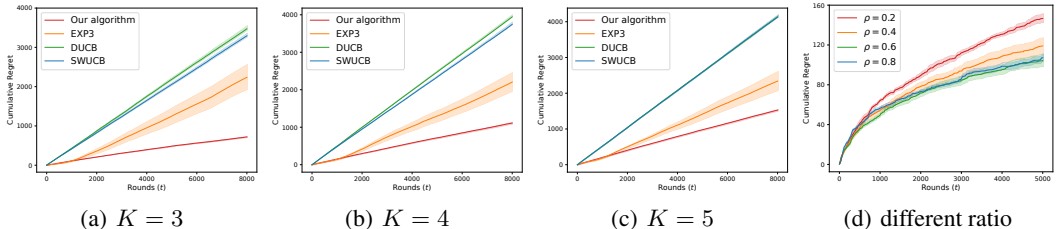

| (a) $K = 3$ | (b) $K = 4$ | (c) $K = 5$ | (d) different ratio |

Fig. 3: (a)-(c): Behavior of the different algorithms on different $K$, the remaining parameters are the same with the simulations in comparison on different $K$. (d): The performance of our algorithms on different ratios, we set $K = 2$ and remaining parameters are also same as before.

discounting factor for the impact. One explanation is that our algorithm utilizes repeated pulling to smooth out historical bias. Given the exponential-decaying nature of time discounting, the amount of pulling required for the impact to converge does not depend on $\gamma$ too heavily. As shown in our theoretical regret upper bounds, gamma can be absorbed with other numeric constants, and when the time horizon increases, the effect of gamma on our algorithm's performance is diminishing, which aligns with our empirical observation. We also examine our algorithm with larger number of base arms $K$ and different ratios $\rho$. The results, as in Figure 3, show that our algorithm outperforms other baselines when $K$ goes large. Furthermore, in our regret bounds (see Theorem 5.1), the regret scales linearly w.r.t $K$. Though the presented results absorb other numeric constants, it is expected to see that the slope of the regret curve is proportionally increasing along with increasing $K$. The results also suggest that our algorithm is not sensitive to different $\rho$, though one could see the regret is slightly lower when $\rho$ is increasing, which is expected from our regret bound.

# 8 Conclusion and Future Work

We explore a multi-armed bandit problem in which actions have delayed impacts to the arm rewards. We propose algorithms that achieve a regret of $\tilde{\mathcal{O}}(KT^{2/3})$ and provide a matching lower regret bound of $\Omega(KT^{2/3})$. Our results complement the bandit literature by exploring the action history dependent biases in bandits. While our model have its limitations, it captures an important but relatively under-explored angle in algorithmic fairness, the long-term impact of actions in sequential learning settings. We hope our study will open more discussions along this direction.

## Acknowledgments and Disclosure of Funding

We thank the anonymous reviewers for their valuable comments. This work is supported in part by the Office of Naval Research Grant N00014-20-1-2240 and the National Science Foundation (NSF) FAI program in collaboration with Amazon under grant IIS-1939677 and IIS-2040800.

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
