| Setup | Notations | Explanations | |
|---|---|---|---|
| | $K; T$ | the number of (base) arms; time horizon | |
| | $k; t$ | arm index; time round $t$ | $k \in [K], t \in [T]$ |
| | $\mathcal{P}$ | probability simplex | $\mathcal{P} \in [0,1]^K$ |
| | $\epsilon$ | discretization parameter | $\epsilon \in [0,1]$ |
| basic setup | $\mathcal{P}_\epsilon$ | probability simplex after discretization with $\epsilon$ | $\mathcal{P}_\epsilon \subset \mathcal{P}$ |
| | $\mathbf{p}$ | meta arm/mixed strategy | $\mathbf{p} \in \mathcal{P}$ |
| | $\mathbf{p}^*$ | optimal meta arm | |
| | $\mathbf{p}^*_\epsilon$ | optimal meta arm in $\mathcal{P}_\epsilon$ | |
| | $r_k(\cdot)$ | expected reward function of arm $k$ | $r_k : [0,1] \to [0,1]$ |
| | $\lambda_k$ | arm $k$'s hyperparameter on tradeoff the expected reward and fairness | |
| | $p_k, p_k(t)$ | the probability on pulling arm $k$, at time $t$ | $p_k, p_k(t) \in [0,1]$ |
| | $L^r_k$ | the Lipschitz constant of $r_k$ | |
| | $L^\pi_k$ | the Lipschitz constant of $\pi_k$ | |
| | $L^*$ | the maximum $L^* = \max(1 + L^r_k + |\lambda_k|L^\pi_k)$ | |
| | $\tilde{r}_t$ | the realized reward at time $t$ | |
| | $\widehat{r}_t$ | the importance weighted reward | |
| | $\bar{r}_t(p_k)$ | the empirical reward mean of discretized arm $p_k$ | |
| | $\mathrm{Reg}(t)$ | cumulative regret till time $t$ | |
| | $\Delta_\mathbf{p}$ | the badness of meta arm $\mathbf{p}$ $t$ | |
| | $\mathbf{p}(t) = \{p_k(t)\}_{k \in [K]}$ | the meta arm deployed in time round $t$ | |
| | $\mathbf{f}(t) = \{f_k(t)\}_{k \in [K]}$ | actions impact function | |
| action dependent bandit | $n_t(p_k)$ | the number of times when pulling arm $k$ with prob $p_k$ till time $t$ | |
| | $N_t(\mathbf{p})$ | number of pulls of meta arm $\mathbf{p}$ till time $t$ | |
| | $\mathcal{S}(p_k)$ | the set of all meta arms which contain $p_k$ | $\mathcal{S}(p_k) = \{\mathbf{p}, p_k \in \mathbf{p}\}$ |
| | $N_t(\mathcal{S}(p_k))$ | total number of pulls of all meta arms in $\mathcal{S}(p_k)$ | $N_t(\mathcal{S}(p_k)) = \sum_{\mathbf{p} \in \mathcal{S}(p_k)} N_t(\mathbf{p})$ |
| | $p_{\min}(\mathbf{p})$ | $p_{\min}(\mathbf{p}) = \arg\min_{p \in \mathbf{p}} n_t(p_k)$ for some $t$ | |
| | $\overline{U}_t(\mathbf{p})$ | the empirical reward mean of meta arm $\mathbf{p}$ | |
| | $U(\mathbf{p})$ | the expected reward of meta arm $\mathbf{p}$ | |
| history dependent bandit | $\gamma$ | time-discounted factor | $\gamma \in (0,1)$ |
| | $L$ | the length of phase | $L \in \mathbb{N}^+$ |
| | $s_a$ | the length of approaching stage | $s_a \in \mathbb{N}^+$ |
| | $\rho$ | the ratio of estimation stages over each phase | $\rho \in (0,1)$ |
| | $m$ | the index of each phase | |
| | $\Gamma_t(p_k)$ | the set of all indexes which arm $k$ is pulled with prob $p_k$ | |
| | $\mathbf{p}^{(\gamma)} = \{p_k^{(\gamma)}\}_{k \in [K]}$ | the time-discounted empirical frequency | |
| | $\bar{r}^{\mathrm{est}}_m(p_k)$ | the empirical reward mean of discretized arm $p_k$ in all estimation stages | |
| | $\overline{U}^{\mathrm{est}}_t(\widehat{\mathbf{p}})$ | the empirical reward mean of meta arm $\mathbf{p}$ in all estimation stages | |
| | $n^{\mathrm{est}}_m(p_k)$ | the number of rounds that arm $k$ is pulled with prob $p_k$ in the first $m$ phases | |

Table 1: The summary of notations.

# A   Related Work

Our learning framework is based on the rich bandit learning literature [3, 42]. However, instead of making the standard assumption of i.i.d. or adversarial rewards, we consider the setting in which the arm reward depends on the action history. The settings most similar to ours are non-stationary bandits, including *restless* bandits [60, 8, 25, 63, 21], in which the reward of each arm changes over time regardless of whether the arm is pulled, and *rested* bandits [43, 58, 17], in which the reward of arm evolves only when it is pulled. In contrast, our model encodes a generic dependency of actions taken in the past and our setting is sort of a mix between the above two. On one hand, the reward of each arm is *restless*, because even if we do not select a particular arm at step $t$, the arm's underlying state will continue to evolve (this is represented by our definition of $\mathbf{f}(t)$), which will change the expected reward to be seen in the future. On the other hand, the changing of rewards does depend on actions, so in this sense, it is related to rested bandit. Technically, due to the presence of historical bias, we allow the learner to learn the optimal strategy in a continuous space which is built on the *probabilistic* simplex over all arms. Meanwhile, our work distinguishes from prior works in that our proposed framework does not require the exact knowledge of dependency function except to the extent of a Lipschitz property and a convergence property.

Our formulation bears similarity to reinforcement learning since our impact function encodes memory (and is in fact Markovian [53, 62]), although we focus on studying the exploration-exploitation tradeoff in bandit formulation. Our techniques and approaches share similar insights with Lipschitz bandits [39, 59, 50, 16] and combinatorial bandits [11, 15, 13, 12] in that we also assume the Lipschitz reward structure and consider combinatorial action space. However, our setting is different since the arm reward explicitly depends on the learner's action history. We have a detailed regret comparisons with the regret of directly applying techniques in combinatorial bandits to our setting in Section 4.

There are several works that formulate delayed feedback in online learning [35, 64, 56, 38, 55, 24, 10, 41]. We discuss the ones that are mostly related to ours. In particular, Pike-Burke et al. [56] considers

the setting in which the observed reward is a sum of a number of previously generated rewards which happen to arrive in the given round. Joulani et al. [35] and Vernade et al. [64] focus on the setting where either feedback or reward is delayed. Our work differs from the above works in that, in our setting, the reward of the arm is influenced by the action history while the above works still consider stationary rewards (though the reward realization could be delayed). There have also been works that study the setting that explores different generative process of reward distribution of arms, e.g., the reward of the arm depends on strategic or biased human behavior [32, 48, 61]. The more closer works to ours include considering the arm of the reward is an increasing concave function of the time since it was last played Kleinberg & Immorlica [38], or decreases as it was played more time [43, 58]. Our work differs from the above in that we formalize an impact function that permits more general form of the reward evolvement as a function of the history of arm plays.

Our work also has implications in algorithmic fairness. one related line of works have studied fairness in the sequential learning setting, however they do not consider long-term impact of actions [34, 7, 49, 30, 27, 44]. For the explorations of delayed impacts of actions, the studies so far have focus on addressing the one-step delayed impacts or a multi-step sequential setting with full information [31, 33, 45, 51, 18, 6]. Our work differs from the above and studies delayed impacts of actions in sequential decision making under uncertainty.

## B  Lagrangian Formulation

While our setting follows standard bandit settings and aims to maximize the utility, it can be extended to incorporate fairness constraints as commonly seen in the discussion of algorithmic fairness. For example, consider the notion of group fairness, which aims to achieve approximate parity of certain measures across groups. Let $\pi_i(f_i(t)) \in [0, 1]$ be the fairness measure for group $i$ (which could reflect the socioeconomic status of the group). One common approach is to impose constraints to avoid the group disparity. Let $\tau \in [0, 1]$ be the tolerance parameter, the fairness constraints at $t$ can be written as: $|\pi_i(f_i(t)) - \pi_j(f_j(t))| \leq \tau, \forall i, j \in [K]$. $\pi_i(\cdot)$ is unknown a priori and is dependent on the historical impact. Incorporating the fairness constraints would transform the goal of the institution as a constrained optimization problem:

$$\max_{\mathbf{p} \in \mathcal{P}} \sum_{t=1}^{T} U_t(\mathbf{p}(t)) \quad \text{s.t.} \quad |\pi_i(f_i(t)) - \pi_j(f_j(t))| \leq \tau, \forall i, j \in [K], \forall t \in [T].$$

We can then utilize the Lagrangian relaxation: impose the fairness requirement as soft constraints and obtain an unconstrained optimization problem with a different utility function. As long as we also observe (bandit) feedback on the fairness measures at every time step, the techniques developed in this work can be extended to include fairness constraints.

To simplify the presentation, we fix a time $t$ and drop the dependency on $t$ in the notations.

**Definition B.1.** *The Lagrangian $\mathcal{L} : \mathcal{P} \times \Lambda^2 \to \mathbb{R}$ where $\Lambda \subseteq \mathbb{R}_+^{\binom{K}{2}}$ of our problem can be formulated as:*

$$\mathcal{L}(\mathbf{p}, \lambda) := \sum_{k=1}^{K} p_k r_k(f_k) - \sum_{c=1}^{\binom{K}{2}} \lambda_c^+ \left( \pi_{i_c}(f_{i_c}) - \pi_{j_c}(f_{j_c}) - \tau \right) - \sum_{c=1}^{\binom{K}{2}} \lambda_c^- \left( \pi_{j_c}(f_{j_c}) - \pi_{i_c}(f_{i_c}) - \tau \right),$$

*where $\lambda^+, \lambda^- \in \Lambda$. The notation $(i_c, j_c) \in \{(i, j)_{1 \leq i < j \leq K}\}$ is a pair of combination and $c \in [K(K-1)/2]$ is the index of each pair of this combination.*

The problem then reduces to jointly maximize over $\mathbf{p} \in \mathcal{P}$ and minimize over $\lambda^+, \lambda^- \in \Lambda$. Rearranging and with a slight abuse of notations, we have the following equivalent optimization problem:

$$\max_{\mathbf{p} \in \mathcal{P}} \min_{\lambda^+, \lambda^-} \sum_{k=1}^{K} p_k(t) r_k(f_k(t)) + \lambda_k \pi_k(f_k(t)) + \tau \sum_{c=1}^{\binom{K}{2}} (\lambda_c^+ + \lambda_c^-), \tag{8}$$

where $\lambda_k := -\sum_{c:i_c=k}(\lambda_c^+ - \lambda_c^-) + \sum_{c:j_c=k}(\lambda_c^+ - \lambda_c^-)$. Due to the uncertainty of reward function $r_k(\cdot)$ and fairness measure $\pi_k(\cdot)$ (recall that our fairness criteria is defined as the parity of socioeconomic status cross different groups, which we can only observe the realization drawn from an

unknown distribution), we treat the above optimization problem as a hyperparameter optimization: similar to choosing hyperparameters (the Lagrange multipliers: $\lambda^+$ and $\lambda^-$) based on a validation set in machine learning tasks. Therefore, given a fixed set of $\lambda^+$ and $\lambda^-$, the problem in (8) can be reduced to the following:

$$\max_{\mathbf{p} \in \mathcal{P}} \sum_{k=1}^{K} p_k(t) \cdot r_k(f_k(t)) + \lambda_k \cdot \pi_k(f_k(t)). \tag{9}$$

## C  Negative Results

In this section, we show that an online algorithm which ignores its action's impact would suffer linear regret. We consider two general bandit algorithms: TS (Thompson Sampling) and a mean-converging family of algorithms (which includes UCB-like algorithms). These are the two most popular and robust bandit algorithms that can be applied to a wide range of scenarios. We prove the negative results respectively. In particular, we construct problem instances that could result in linear regret if the deployed algorithm ignore the action's impact.

**Example 1.** *Considering the following Bernoulli bandit instance with two arms, indexed by arm 1 and arm 2, i.e., $K = 2$. For any $\epsilon \in [0, 1/2)$, define the expected reward of each arm as follows:*

- *arm 1: $r_1(p) = p/(1 - \epsilon) \cdot \mathbb{1}(p \leq 1 - \epsilon) + (2 - \epsilon - p) \cdot \mathbb{1}(p \geq 1 - \epsilon), \quad \forall p \in [0, 1]$*
- *arm 2: $r_2(p) = p/(2\epsilon) \cdot \mathbb{1}(p \leq \epsilon) + (-\frac{1}{2}p + \frac{1}{2}(1 + \epsilon)) \cdot \mathbb{1}(p \geq \epsilon), \quad \forall p \in [0, 1]$*

It is easy to see that $\mathbf{p}^* = \{1 - \epsilon, \epsilon\}$ is the optimal strategy for the above bandit instance.

We first prove the negative result of Thompson Sampling using the above example. The Thompson Sampling algorithm can be summarized as below.

---
**Algorithm 4** Thompson Sampling
---
1: $S_i = 0, F_i = 0$.
2: **for** $t = 1, 2, \ldots,$ **do**
3:    For each arm $i = 1, 2$, sample $\theta_i(t)$ from the `Beta`$(S_i + 1, F_i + 1)$ distribution.
4:    Play arm $a_t := \arg\max_i \theta_i(t)$ and observe reward $\tilde{r}_t$.
5:    If $\tilde{r}_t = 1$, then $S_{a_t} = S_{a_t} + 1$, else $F_{a_t} = F_{a_t} + 1$.
6: **end for**

---

**Lemma C.1.** *For the reward structure defined in Example 1, Thompson Sampling would suffer linear regret if it doesn't consider the action's impact it deploys at every time round, namely, it takes the sample mean as the true mean reward of each arm.*

Before we proceed, we first prove the following strong law of large numbers in Beta distribution. We note that the below two lemmas are not new results and can be found in many statistical books. We provide proofs here for the sake of making the current work self-contained.

**Lemma C.2.** *Consider the Beta distribution* `Beta`$(a\alpha + 1, b\alpha + 1)$ *whose pdf is defined as $f(x, \alpha) = \frac{[x^a(1-x)^b]^\alpha}{B(a\alpha+1, b\alpha+1)}$, where $B(\cdot)$ is the beta function, then for any positive $(a, b)$ such that $a + b = 1$, when $\alpha \to \infty$, the limit of $f(x, \alpha)$ can be characterized by Dirac delta function $\delta(x - a)$.*

*Proof.* By Stirling's approximation, we can write the asymptotics of beta function as follows:

$$B(x, y) \sim \sqrt{2\pi} \frac{x^{x-0.5} y^{y-0.5}}{(x+y)^{x+y-0.5}}.$$

Thus, when $\alpha \to \infty$, i.e., for large $a\alpha + 1$ and $b\alpha + 1$, we can approximate the pdf $f(x, \alpha)$ in the following:

$$f(x, \alpha) \sim \sqrt{\frac{a+2}{2\pi ab}} h^\alpha(x),$$

where $h(x) := (x/a)^a \left(\frac{1-x}{b}\right)^b$. It's easy to see that $h(x)$ has a unique maximum at $a$, by invoking Lemma C.3 will complete the proof. □

**Lemma C.3.** *Let $h : [0,1] \to \mathbb{R}^+$ be any bounded measurable non-negative function with a unique maximum at $x^*$, and suppose $h$ is continuous at $x^*$. For $\lambda > 0$ define $h_\lambda(x) = C_\lambda h^\lambda(x)$ where $C_\lambda$ normalizes such that $\int_0^1 h_\lambda(x)dx = 1$. Consider any continuous function $f$ defined on $[0,1]$ and $\epsilon > 0$, then we have $\lim_{\lambda \to \infty} \int_{h(x) \leq h(x^*) - \epsilon} h_\lambda(x)f(x)dx = 0$ and $\lim_{\lambda \to \infty} \int_0^1 h_\lambda(x)f(x)dx = f(x^*)$.*

*Proof.* For any $\delta > 0$, we have

$$\left| \int_0^1 h_\lambda(x)f(x)dx - f(x^*) \right|$$

$$= \left| \int_0^1 h_\lambda(x)\big(f(x) - f(x^*)\big)dx \right|$$

$$\leq \left| \int_{|x-x^*| \leq \delta} h_\lambda(x)\big(f(x) - f(x^*)\big)dx \right| + \left| \int_{|x-x^*| > \delta} h_\lambda(x)\big(f(x) - f(x^*)\big)dx \right|$$

$$\leq \left| \int_{|x-x^*| \leq \delta} h_\lambda(x)\big(f(x) - f(x^*)\big)dx \right| + \max \big|f(x) - f(x^*)\big| \left| \int_{|x-x^*| > \delta} h_\lambda(x)dx \right|.$$

For any $\delta > 0$, and due to the continuous property of $f$ on $x^*$, which further implies that there exists a constant $c > 0$ such that $|f(x) - f(x^*)| < \delta/2$ whenever $|x - x^*| < c$. Thus, given $c > \delta$, we have

$$\left| \int_0^1 h_\lambda(x)f(x)dx - f(x^*) \right| \leq \epsilon/2 + \max \big|f(x) - f(x^*)\big| \left| \int_{|x-x^*| > \delta} h_\lambda(x)dx \right|.$$

It suffices to show that the second term in RHS of above inequality will converge to 0 as $\lambda \to \infty$. Let $||h||_{\infty,\delta}$ denote the $L^\infty$ norm of $h$ when $h$ is restricted to $\{|x - x^*| > \delta\}$. Note that for any nonnegative integrable functions $h$, we have

$$\lim_{\lambda \to \infty} \left( \int_0^1 h^\lambda(x)dx \right)^{1/\lambda} = ||h||_\infty.$$

Recall the definition of $C_\lambda = \frac{1}{\int_0^1 h^\lambda(x)dx}$, thus, we have $\lim_{\lambda \to \infty} C_\lambda^{1/\lambda} = \frac{1}{||h||_\infty}$, which immediately showing that

$$\left( \int_{|x-x^*| > \delta} h_\lambda(x)dx \right)^{1/\lambda} = C_\lambda^{1/\lambda} \left( \int_{|x-x^*| > \delta} h^\lambda(x)dx \right)^{1/\lambda},$$

which further implies that $||h||_{\infty,\delta}/||h||_\infty < 1$. Thus, there must exist $\lambda_0$ such that $\forall \lambda > \lambda_0$,

$$\left( \int_{|x-x^*| > \delta} h_\lambda(x)dx \right)^{1/\lambda} < \gamma < 1. \tag{10}$$

Since $\gamma < 1$, we then have $\lim_{\lambda \to \infty} \gamma^\lambda = 0$, this implies the second term of RHS of (10) converging to 0 as $\lambda \to \infty$. $\square$

We now ready to prove Lemma C.1.

*Proof.* We prove this by contradiction. Let $\mathrm{Reg}(T)$ denote the expected regret incurred by TS up to time round $T$, and $N_t(\mathbf{p}) = \sum_{s=1}^t \mathbb{1}(\mathbf{p}(s) = \mathbf{p})$ denote the number of rounds when the algorithm deploys the (mixed) strategy $\mathbf{p} \in \Delta_K$. Furthermore, let $S_i(t)$(resp. $F_i(t)$) denote the received $1_s$(resp. $0_s$) of arm $i$ up to time round $t$. Recall that in Thompson Sampling, we have $\mathbb{P}(a_t = 1) = \mathbb{P}\big(\theta_1(t) > \theta_2(t)\big)$. By the reward function defined in Example 1, it's immediate to see that

$$S_1(T) \geq (1 - \epsilon)N_T(\mathbf{p}^*); \quad F_1(T) \leq T - N_T(\mathbf{p}^*); \quad S_2(T) \geq 0.5\epsilon N_T(\mathbf{p}^*); \quad F_2(T) \geq 0.5\epsilon N_T(\mathbf{p}^*).$$

Now suppose Thompson Sampling achieves sublinear regret, i.e., $\mathrm{Reg}(T) = o(T)$, which implies following

$$\lim_{T \to \infty} \frac{T - N_T(\mathbf{p}^*)}{T} = 0.$$

Thus, by the strong law of large numbers and invoking Lemma C.2, the sample $\theta_1(T+1) \sim$ `Beta`$(S_1(T), F_1(T))$ and $\theta_2(T+1) \sim$ `Beta`$(S_2(T), F_2(T))$ will converge as follows:

$$\lim_{T \to \infty} \theta_1(T+1) = 1; \quad \lim_{T \to \infty} \theta_2(T+1) = 0.5.$$

Then it's almost surely that $\lim_{T \to \infty} \mathbb{P}(a_{T+1} = 1) = \lim_{T \to \infty} \mathbb{P}(\theta_1(T+1) > \theta_2(T+1)) = 1$. This leads to following holds for sure

$$S_1(s+1) = S_1(s) + 1, \forall s > T.$$

Thus, consider the regret incurred from the $(T+1)-$th round to $(2T)-$th round, the regret will be

$$\text{Reg}(2T) - \text{Reg}(T) = \sum_{s=T+1}^{2T} U(\mathbf{p}(s)) = 0.5 T \epsilon,$$

where the second equality follows that $\mathbf{p}(s) = (1,0)$ holds almost surely from $T+1$ to $2T$. This shows that $\lim_{T \to \infty} \frac{\mathbb{E}[\text{Reg}(2T)]}{2T} = \epsilon/4$, which contradicts that the algorithm achieves the sublinear regret. $\square$

We now show that a general class of algorithms, which are based on *mean-converging*, will suffer linear regret if it ignores the action's impact. This family of algorithms includes UCB algorithm in classic MAB problems.

**Definition C.4** (Mean-converging Algorithm [57]). *Define $I_k(t) = \{s : a_s = k, s < t\}$ as the set of time rounds such the arm $k$ is chosen. Let $\bar{r}_k(t) = \frac{1}{|I_k(t)|} \sum_{s \in I_k(t)} \tilde{r}_s$ be the empirical mean of arm $k$ up to time $t$. The mean-converging algorithm $\mathcal{A}$ assigns $s_k(t)$ for each arm $k$ if following holds true:*

- *$s_k(t)$ is the function of $\{\tilde{r}_s : s \in I_k(t)\}$ and time $t$;*

- *$\mathbb{P}(s_k(t) = \bar{r}_k(t)) = 1 \quad$ if $\quad \liminf_t \frac{|I_k(t)|}{t} > 0$.*

**Lemma C.5.** *For the reward structure defined in Example 1, the mean-converging Algorithm will suffer linear regret if it mistakenly take the sample mean as the true mean reward of each arm.*

*Proof.* We prove above lemma by contradiction. Let $N_t^{\mathcal{A}}(\mathbf{p})$ denote the number of plays with deploying the strategy $\mathbf{p}$ by algorithm $\mathcal{A}$ till time $t$. Suppose a mean-converging Algorithm $\mathcal{A}$ achieves sublinear regret, then it must have $\lim_{T \to \infty} N_T^{\mathcal{A}}(\mathbf{p}^*)/T > 0$ and $\lim_{T \to \infty} (T - N_T^{\mathcal{A}}(\mathbf{p}^*))/T = o(T)$. By the definition of mean-converging algorithm and recall the reward structure defined in Example 1, the score $s_T(1)$ assigned to arm 1 by the algorithm $\mathcal{A}$ must be converging to 1, and the score of $s_T(2)$ assigned to arm 2 must be converging to 0.5. By the strong law of large numbers, it suffices to show that $\mathbb{P}(\mathbf{p}(t) = \{1, 0\}) = 1, \forall t \geq T + 1$, which implies the algorithm $\mathcal{A}$ would suffer linear regret after $T$ time rounds and thus completes the proof. $\square$

# D    Missing Proofs for Action-Dependent Bandits

## D.1    The naive method that directly utilize techniques from Lipschitz bandits

We first give a naive approach which directly applies Lipschitz bandit technique to our action-dependent setting. Recall that each meta arm $\mathbf{p}$ specifies the probability $p_k \in [0, 1]$ for choosing each base arm $k$. We *uniformly* discretize each $p_k$ into intervals of a fixed length $\epsilon$, with carefully chosen $\epsilon$ such that $1/\epsilon$ is an positive integer. Let $\mathcal{P}_\epsilon$ be the space of discretized meta arms, i.e., for each $\mathbf{p} = \{p_1, \ldots, p_K\} \in \mathcal{P}_\epsilon, \sum_{k=1}^K p_k = 1$ and $p_k \in \{0, \epsilon, 2\epsilon, \ldots, 1\}$ for all $k$. We then run standard bandit algorithms on the finite set $\mathcal{P}_\epsilon$.

There is a natural trade-off on the choice of $\epsilon$, which controls the complexity of arm space and the discretization error. show that, with appropriately chosen $\epsilon$, this approach can achieve sublinear regret (with respect to the optimal arm in the non-discretized space $\mathcal{P}$).

**Lemma D.1.** *Let $\epsilon = \Theta\big(\big(\frac{\ln T}{T}\big)^{\frac{1}{K+1}}\big)$. Running a bandit algorithm which achieves optimal regret $\mathcal{O}(\sqrt{|\mathcal{P}_\epsilon| T \ln T})$ on the strategy space $\mathcal{P}_\epsilon$ attains the following regret (w.r.t. the optimal arm in non-discretized $\mathcal{P}$): $\text{Reg}(T) = \mathcal{O}\big(T^{\frac{K}{K+1}} (\ln T)^{\frac{1}{K+1}}\big)$.*

*Proof.* As mentioned, we *uniformly* discretize the interval $[0, 1]$ of each arm into interval of a fixed length $\epsilon$. The strategy space will be reduced as $\mathcal{P}_\epsilon$, which we use this as an approximation for the full set $\mathcal{P}$. Then the original infinite action space will be reduces as finite $\mathcal{P}_\epsilon$, and we run an off-the-shelf MAB algorithm $\mathcal{A}$, such as UCB1 or Successive Elimination, that only considers these actions in $\mathcal{P}_\epsilon$. Adding more points to $\mathcal{P}_\epsilon$ makes it a better approximation of $\mathcal{P}$, but also increases regret of $\mathcal{A}$ on $\mathcal{P}_\epsilon$. Thus, $\mathcal{P}_\epsilon$ should be chosen so as to optimize this tradeoff. Let $\mathbf{p}_\epsilon^* := \sup_{\mathbf{p} \in \mathcal{P}_\epsilon} \sum_{k=1}^K p_k r_k(p_k)$ denote the best strategy in discretized space $\mathcal{P}_\epsilon$. At each round, the algorithm $\mathcal{A}$ can only hope to approach expected reward $U(\mathbf{p}_\epsilon^*)$, and together with additionally suffering *discretization error*:

$$\mathtt{DE}_\epsilon := U(\mathbf{p}^*) - U(\mathbf{p}_\epsilon^*).$$

Then the expected regret of the entire algorithm is:

$$\begin{aligned}
\mathrm{Reg}(T) &= T \cdot U(\mathbf{p}^*) - \mathtt{Reward}(\mathcal{A}) \\
&= T \cdot U(\mathbf{p}_\epsilon^*) - \mathtt{Reward}(\mathcal{A}) + T(U(\mathbf{p}^*) - U(\mathbf{p}_\epsilon^*)) \\
&= \mathbb{E}[\mathrm{Reg}_\epsilon(T)] + T \cdot \mathtt{DE}_\epsilon,
\end{aligned}$$

where $\mathtt{Reward}(\mathcal{A})$ is the total reward of the algorithm, and $\mathrm{Reg}_\epsilon(T)$ is the regret relative to $U(\mathbf{p}_\epsilon^*)$. If $\mathcal{A}$ attains optimal regret $\mathcal{O}(\sqrt{KT \ln T})$ on any problem instance with time horizon $T$ and $K$ arms, then,

$$\mathrm{Reg}(T) \le \mathcal{O}(\sqrt{|\mathcal{P}_\epsilon| T \ln T}) + T \cdot \mathtt{DE}_\epsilon.$$

Thus, we need to choose $\epsilon$ to get the optimal trade-off between the size of $\mathcal{P}_\epsilon$ and its discretization error. Recall that $r_k(\cdot)$ is Lipschitz-continuous with the constant of $L_k$, thus, we could bound the $\mathtt{DE}_\epsilon$ by restricting $\mathbf{p}_\epsilon^*$ to be nearest w.r.t $\mathbf{p}^*$. Let $L^* = \max_{k \in [K]}(1 + L_k)$, then it's easy to see that

$$\mathtt{DE}_\epsilon = \Omega(KL^*\epsilon).$$

Thus, the total regret can be bounded above from:

$$\mathrm{Reg}(T) \le \mathcal{O}\left(\sqrt{(1/\epsilon + 1)^{K-1} T \ln T}\right) + \Omega(TKL^*\epsilon).$$

By choosing $\epsilon = \Theta\left(\left(\frac{\ln T}{T(L^*)^2}\right)^{\frac{1}{K+1}}\right)$ we obtain:

$$\mathrm{Reg}(T) \le \mathcal{O}(cT^{\frac{K}{K+1}}(\ln T)^{\frac{1}{K+1}}).$$

where $c = \Theta\left(K(L^*)^{\frac{K-1}{K+1}}\right)$. $\qquad\square$

### D.2 Missing Discussions and Proofs of Theorem 4.2

**Step 1: Bounding the error of $|\overline{U}(\mathbf{p}) - U(\mathbf{p})|$.** For any $\mathbf{p} = \{p_1, \ldots, p_K\}$, define the empirical reward $\overline{U}_t(\mathbf{p}) = \sum_{k=1}^K p_k \bar{r}_t(p_k)$. The first step of our proof is to bound $\mathbb{P}(|\overline{U}_t(\mathbf{p}) - U(\mathbf{p})| \le \delta)$ for each meta arm $\mathbf{p} = \{p_1, \ldots, p_K\}$ with high probability.[9] Using the Hoeffding's inequality, we obtain

$$\begin{aligned}
\mathbb{P}(|\overline{U}_t(\mathbf{p}) - U(\mathbf{p})| \ge \delta) &= \mathbb{P}\left(\left|\sum_k \frac{\sum_{s \in \mathcal{T}_t(p_k)} \widehat{r}_s(p_k)}{n_t(p_k)} - \sum_k p_k r(p_k)\right| \ge \delta\right) \\
&\le 2\exp\left(-\frac{2\delta^2}{\sum_k \frac{1}{n_t(p_k)}}\right) \le 2\exp\left(-\frac{2\delta^2 n_t(p_{\min}(\mathbf{p}))}{K}\right),
\end{aligned}$$

where $p_{\min}(\mathbf{p}) := \arg\min_{p_k \in \mathbf{p}} n_t(p_k)$. By choosing $\delta = \sqrt{\frac{K \ln t}{n_t(p_{\min}(\mathbf{p}))}}$ in the above inequality, for each meta arm $\mathbf{p}$ at time $t$, we have that $|\overline{U}_t(\mathbf{p}) - U(\mathbf{p})| \le \sqrt{K \ln t / n_t(p_{\min}(\mathbf{p}))}$, with the probability at least $1 - 2/t^2$.

---

[9] We use $\delta$ to denote the estimation error, as $\epsilon$ has been used as the discretization parameter.

**Step** 2**: Bounding the probability on deploying suboptimal meta arm.** With the above high probability bound we obtain in Step 1, we can construct an UCB index for each meta arm $\mathbf{p} \in \mathcal{P}_\epsilon$:

$$\text{UCB}_t(\mathbf{p}) = \overline{U}_t(\mathbf{p}) + \sqrt{\frac{K \ln t}{n_t(p_{\min}(\mathbf{p}))}}. \tag{11}$$

The above constructed UCB index gives the following guarantee:

**Lemma D.2.** *At any time round $t$, for a suboptimal meta arm $\mathbf{p}$, if it satisfies $n_t(p_{\min}(\mathbf{p})) \geq 4K \ln t / \Delta_{\mathbf{p}}^2$, then $\text{UCB}_t(\mathbf{p}) < \text{UCB}_t(\mathbf{p}_\epsilon^*)$ with the probability at least $1 - 4/t^2$. Thus, for any $t$,*

$$\mathbb{P}\left(\mathbf{p}(t) = \mathbf{p} | n_t(p_{\min}(\mathbf{p})) \geq 4K \ln t / \Delta_{\mathbf{p}}^2\right) \leq 4t^{-2},$$

*where $\Delta_{\mathbf{p}}$ denotes the badness of meta arm $\mathbf{p}$.*

*Proof.* We prove this lemma by considering two "events" which occur with high probability: (1) the UCB index of each meta arm will concentrate on the true mean utility of $\mathbf{p}$; (2) the empirical mean utility of each meta arm $\mathbf{p}$ will also concentrate on the true mean utility of $\mathbf{p}$. We then show that the probability of either one of the events not holding is at most $4/t^2$. By a union bound we prove above desired lemma.

$$
\begin{aligned}
\text{UCB}_t(\mathbf{p}) &= \sum_{k=1}^{K} p_k \bar{r}_t(p_k) + \sqrt{\ln t \frac{K}{n_t(p_{\min}(\mathbf{p}))}} \\
&\overset{(a)}{\leq} \sum_{k=1}^{K} p_k \bar{r}_t(p_k) + \Delta_{\mathbf{p}}/2 < \left(\sum_{k=1}^{K} p_k r_k(p_k) + \Delta_{\mathbf{p}}/2\right) + \Delta_{\mathbf{p}}/2 && \text{By Event 1} \\
&= \sum_{k=1}^{K} p_{k,\epsilon}^* r_k(p_{k,\epsilon}^*) < \sum_{k=1}^{K} p_{k,\epsilon}^* \bar{r}_t(p_{k,\epsilon}^*) + \sqrt{\ln t \frac{K}{n_t(p_{\min}(\mathbf{p}_\epsilon^*))}} && \text{By Event 2} \\
&= \text{UCB}_t(\mathbf{p}_\epsilon^*),
\end{aligned}
$$

where $\mathbf{p}_\epsilon^* = (p_{1,\epsilon}^*, \ldots, p_{K,\epsilon}^*)$. The first inequality (a) comes from that $n_t(p_{\min}(\mathbf{p})) \geq \frac{4K \ln t}{\Delta_{\mathbf{p}}^2}$ and the probability of third inequality or fifth inequality not holding is at most $4/t^2$. $\qquad\square$

Intuitively, Lemma D.2 essentially shows that for a meta arm $\mathbf{p}$, if its $n_t(p_{\min}(\mathbf{p}))$ is sufficiently sampled with respect to $\Delta_{\mathbf{p}}$, that is, sampled at least $4K \ln t / \Delta_{\mathbf{p}}^2$ times, we know that the probability that we hit this suboptimal meta arm is very small.

**Step** 3**: Bounding the $\mathbb{E}[n_T(p_{\min}(\mathbf{p}))]$.** Ideally, we would like to bound the number of the selections on deploying the suboptimal meta arm, i.e., $N_T(\mathbf{p})$, in a logarithmic order of $T$. However, if we proceed to bound this by separately considering each meta arm, the final regret bound will have an order with exponent in $K$ since the number of meta arms grows exponentially in $K$. Instead, we turn to bound $\mathbb{E}[n_T(p_{\min}(\mathbf{p}))]$. Recall that by the definitions of $n_T(p)$ and $p_{\min}(\mathbf{p})$, the pulls of $\mathbf{p}$ is upper bounded by its $n_T(p_{\min}(\mathbf{p}))$. This quantity will help us to reduce the exponential $K$ to the polynomial $K$. This is formalized in the following lemma.

**Lemma D.3.** *For each suboptimal meta arm $\mathbf{p} \neq \mathbf{p}_\epsilon^*$, we have that $\mathbb{E}[n_T(p_{\min}(\mathbf{p}))] \leq \frac{4K \ln T}{\Delta_{\mathbf{p}}^2} + \mathcal{O}(1)$.*

*Proof.* To simplify notations, for each discretized arm $p_k$, we define the notion of *super set* $\mathcal{S}(p_k) = \{\mathbf{p} : p_k \in \mathbf{p}\}$ which contains all the meta arms that include this discretized arm. For suboptimal meta

arm $\mathbf{p} \neq \mathbf{p}_\epsilon^*$ and its $p_{\min}(\mathbf{p})$, we have

$$\mathbb{E}[n_T(p_{\min}(\mathbf{p}))]$$

$$\overset{(a)}{=} 1 + \mathbb{E}\left[\sum_{t=\lceil K/\epsilon \rceil + 1}^{T} \mathbb{1}\left(\mathbf{p}(t) = \mathbf{p}, \mathbf{p} \in \mathcal{S}(p_{\min}(\mathbf{p}))\right)\right]$$

$$= 1 + \mathbb{E}\left[\sum_{t=\lceil K/\epsilon \rceil + 1}^{T} \mathbb{1}\left(\mathbf{p}(t) = \mathbf{p}, \mathbf{p} \in \mathcal{S}(p_{\min}(\mathbf{p})); n_t(p_{\min}(\mathbf{p})) < \frac{4K \ln t}{\Delta_{\mathbf{p}}^2}\right)\right]$$

$$+ \mathbb{E}\left[\sum_{t=\lceil K/\epsilon \rceil + 1}^{T} \mathbb{1}\left(\mathbf{p}(t) = \mathbf{p}, \mathbf{p} \in \mathcal{S}(p_{\min}(\mathbf{p})); n_t(p_{\min}(\mathbf{p})) \geq \frac{4K \ln t}{\Delta_{\mathbf{p}}^2}\right)\right]$$

$$\overset{(b)}{\leq} \frac{4K \ln T}{\Delta_{\mathbf{p}}^2} + \mathbb{E}\left[\sum_{t=\lceil K/\epsilon \rceil + 1}^{T} \mathbb{1}\left(\mathbf{p}(t) = \mathbf{p}, \mathbf{p} \in \mathcal{S}(p_{\min}(\mathbf{p})); n_t(p_{\min}(\mathbf{p})) \geq \frac{4K \ln t}{\Delta_{\mathbf{p}}^2}\right)\right]$$

$$= \frac{4K \ln T}{\Delta_{\mathbf{p}}^2} + \sum_{t=\lceil K/\epsilon \rceil + 1}^{T} \mathbb{P}\left(\mathbf{p}(t) = \mathbf{p}, \mathbf{p} \in \mathcal{S}(p_{\min}(\mathbf{p})); n_t(p_{\min}(\mathbf{p})) \geq \frac{4K \ln t}{\Delta_{\mathbf{p}}^2}\right)$$

$$= \frac{4K \ln T}{\Delta_{\mathbf{p}}^2} + \sum_{t=\lceil K/\epsilon \rceil + 1}^{T} \mathbb{P}\left(\mathbf{p}(t) = \mathbf{p}, \mathbf{p} \in \mathcal{S}(p_{\min}(\mathbf{p})) \middle| n_t(p_{\min}(\mathbf{p})) \geq \frac{4K \ln t}{\Delta_{\mathbf{p}}^2}\right) \mathbb{P}\left(n_t(p_{\min}(\mathbf{p})) \geq \frac{4K \ln t}{\Delta_{\mathbf{p}}^2}\right)$$

$$\overset{(c)}{\leq} \frac{4K \ln T}{\Delta_{\mathbf{p}}^2} + \frac{2\pi^2}{3}.$$

We add 1 in the first equality to account for 1 (step (a)) initial pull of every discretized arm by the algorithm (the initialization phase). In step (b), suppose for contradiction that the indicator $\mathbb{1}\left(\mathbf{p}(t) = \mathbf{p}, \mathbf{p} \in \mathcal{S}(p_{\min}(\mathbf{p})); n_t(p_{\min}(\mathbf{p})) < S\right)$ takes value of 1 at more than $S - 1$ time steps, where $S = \frac{4K \ln T}{\Delta_{\mathbf{p}}^2}$. Let $\tau$ be the time step at which this indicator is 1 for the $(S-1)$-th time. Then the number of pulls of all meta arms in $\mathcal{S}(p_{\min}(\mathbf{p}))$ is at least $L$ times until time $\tau$ (including the initial pull), and for all $t \geq \tau$, $n_t(p_{\min}(\mathbf{p})) \geq S$ which implies $n_t(p_{\min}(\mathbf{p})) \geq \frac{4K \ln t}{\Delta_{\mathbf{p}}^2}$. Thus, the indicator cannot be 1 for any $t \geq \tau$, contradicting the assumption that the indicator takes value of 1 more than $L$ times. This bounds $1 + \mathbb{E}\left[\sum_{t \geq \lceil K/\epsilon \rceil + 1} \mathbb{1}\left(\mathbf{p}(t) = \mathbf{p}, \mathbf{p} \in \mathcal{S}(p_{\min}(\mathbf{p})); n_t(p_{\min}(\mathbf{p})) < S\right)\right]$ by $S$. In step (c), we apply the lemma D.2 to bound the first conditional probability term and use the fact that the probabilities cannot exceed 1 to bound the second probability term. $\square$

We use this connection in the following step to reduce the computation of regret on pulling all suboptimal meta arms so that to calculate the regret via the summation over discretized arms.

**Wrapping up: Proof of Theorem 4.2.** We are now ready to prove Theorem 4.2. We first define notations that are helpful for our analysis. To circumvent the summation over all feasible suboptimal arms $\{\mathbf{p}\}$, for each discretized arm $p_k$, we define the notion of *super set* $\mathcal{S}(p_k) := \{\mathbf{p} : p_k \in \mathbf{p}\}$ which contains all suboptimal meta arms that include this discretized arm. With a slight abuse of notations, we also sort all meta arms in $\mathcal{S}(p_k)$ as $\mathbf{p}_1, \mathbf{p}_2, \ldots, \mathbf{p}_{I(p_k)}$ in ascending order of their expected rewards, where $I(p_k) := |\mathcal{S}(p_k)|$ is the cardinality of the super set $\mathcal{S}(p_k)$. For $\mathbf{p}_l \in \mathcal{S}(p_k)$, we also define $\Delta_l^{p_k} := \Delta_{\mathbf{p}_l}$ where $l \in [I(p_k)]$, and specifically $\Delta_{\min}^{p_k} := \min_{\mathbf{p} \in \mathcal{S}(p_k)} \Delta_{\mathbf{p}} = \Delta_{I(p_k)}^{p_k}$; $\Delta_{\max}^{p_k} := \max_{\mathbf{p} \in \mathcal{S}(p_k)} \Delta_{\mathbf{p}} = \Delta_1^{p_k}$. Let $\text{Reg}_\epsilon(T)$ denote the regret relative to the best strategy in the discretized space parameterized by $\epsilon$. With these notations, we first establish the following instance-dependent regret.

**Lemma D.4.** *Following the* UCB *designed in (11), we have the following instance-dependent regret on the discretized arm space:* $\text{Reg}_\epsilon(T) \leq \lceil K/\epsilon \rceil \cdot (\Delta_{\max} + \mathcal{O}(1)) + \sum_{p_k : \Delta_{\min}^{p_k} > 0} 8K \ln T / \Delta_{\min}^{p_k}$, *where* $\Delta_{\max} := \max_{p_k} \Delta_{\max}^{p_k}$.

*Proof.* Note that by definition, we can compute the regret $\text{Reg}_\epsilon(T)$ as follows:

$$\text{Reg}_\epsilon(T) = \sum_{\mathbf{p} \in \mathcal{P}_\epsilon} \mathbb{E}[N_T(\mathbf{p})]\Delta_\mathbf{p} \leq \sum_{p_k} \sum_{l \in [I(p_k)]} \mathbb{E}[N_T(\mathbf{p}_l)]\Delta_l^{p_k}. \tag{12}$$

Observe that, by Lemma D.3, for each discretized arm $p_k$, there are two possible cases:

- There exists a meta arm $\mathbf{p}_l \in \mathcal{S}(p_k)$, and its $p_{\min}(\mathbf{p}_l) = p_k$. Then by linearity of expectation, we can bound the expectation of total number of pulls for all $\mathbf{p}_{l'} \in \mathcal{S}(p_k)$ as follows

$$\sum_{\mathbf{p}_{l'} \in \mathcal{S}(p_k)} \mathbb{E}[N_T(\mathbf{p}_{l'})] = \mathbb{E}[n_T(p_k)] \leq \frac{4K \ln T}{(\Delta_{\min}^{p_k})^2} + \mathcal{O}(1).$$

- There exists no meta arm $\mathbf{p} \in \mathcal{S}(p_k)$, and $p_{\min}(\mathbf{p})$ for each $\mathbf{p}$ is $p_k$. In this case, for each $\mathbf{p}_l \in \mathcal{S}(p_k)$, there always exists another discretized arm $p'$ that is included in $\mathbf{p}_l$ such that $p' = p_{\min}(\mathbf{p}_l)$ but $p' \neq p_k$. Thus, for each $\mathbf{p}_l \in \mathcal{S}(p_k)$, together with other meta arms which also include discretized arm $p'$ as $\mathbf{p}_l$, we have that

$$\sum_{\mathbf{p} \in \bigcup_{p' \in \mathbf{p}} \mathbf{p}} \mathbb{E}[N_T(\mathbf{p})] = \sum_{\mathbf{p} \in \mathcal{S}(p')} \mathbb{E}[N_T(\mathbf{p})]$$
$$= \mathbb{E}[n_T(p')] \leq \frac{4K \ln T}{(\Delta_{\min}^{p'})^2} + \mathcal{O}(1).$$

The above observations imply that even though we can not find any meta arm $\mathbf{p}$ in $\mathcal{S}(p_k)$ such that $p_{\min}(\mathbf{p}) = p_k$, we can always carry out similar analysis by finding another discretized arm $p' \in \mathbf{p}$ but $p' \neq p_k$, such that $p' = p_{\min}(\mathbf{p})$. Thus, for each discretized arm $p_k$, we can focus on the case where $p_k$ is able to attain the minimum $n_t(p_k)$ for some $\mathbf{p} \in \mathcal{S}(p_k)$. For analysis convenience, instead of looking at the counter of $\mathbf{p}$, i.e., $n_t(p_{\min}(\mathbf{p}))$, we will define a counter $c(p_k)$ for each discretized arm $p_k$ and the value of $c(p_k)$ at time $t$ is denoted by $c_t(p_k)$. The update of $c_t(p_k)$ is as follows: For a round $t > \lceil K/\epsilon \rceil$ (here $\lceil K/\epsilon \rceil$ is the number of rounds needed for initialization), let $\mathbf{p}(t)$ be the meta arm selected in round $t$ by the algorithm. Let $p_k = \arg\min_{p_k \in \mathbf{p}(t)} c_{t-1}(p_k)$. We increment $c(p_k)$ by one, i.e., $c_t(p_k) = c_{t-1}(p_k) + 1$. In other words, we find the discretized arm $p_k$ with the smallest counter in $\mathbf{p}(t)$ and increment its counter. If such $p_k$ is not unique, we pick an arbitrary discretized arm with the smallest counter. Note that the initialization gives $\sum_{p_k} c_{\lceil K/\epsilon \rceil}(p_k) = \lceil K/\epsilon \rceil$. It is easy to see that for any $p_k = p_{\min}(\mathbf{p})$, we have $n_t(p_k) = c_t(p_k)$.

With the above change of counters, Lemma D.2 and Lemma D.3 then have the implication on selecting discretized arm $p_k \notin \mathbf{p}_\epsilon^*$ given its counter $c_t(p_k)$. To see this, for each $\mathbf{p}_l \in \mathcal{S}(p_k)$, we define sufficient selection of discretized arm $p_k$ with respect to $\mathbf{p}_l$ as $p_k$ being selected $4K \ln T / (\Delta_l^{p_k})^2$ times and $p_k$'s counter $c(p_k)$ being incremented in these selected instances. Then Lemma D.2 tells us when $p_k$ is sufficiently selected with respect to $\mathbf{p}_l$, the probability that the meta arm $\mathbf{p}_l$ is selected by the algorithm is very small. On the other hand, when $p_k$'s counter $c(p_k)$ is incremented, but if $p_k$ is under-selected with respect to $\mathbf{p}_l$, we incur a regret of at most $\Delta_j^{p_k}$ for some $j \leq l$.

Define $C_T(\Delta) := \frac{4K \ln T}{\Delta^2}$, the number of selection that is considered sufficient for a meta arm with reward $\Delta$ away from the optimal strategy $\mathbf{p}_\epsilon^*$ with respect to time horizon $t$. With the above analysis, we define following two situations for the counter of each discretizad arm:

$$c_T^{l,\mathbf{suf}}(p_k) := \sum_{t=\lceil K/\epsilon \rceil + 1}^{T} \mathbb{1}\left(\mathbf{p}(t) = \mathbf{p}_l, c_t(p_k) > c_{t-1}(p_k) > C_T(\Delta_l^{p_k})\right),$$

$$c_T^{l,\mathbf{und}}(p_k) := \sum_{t=\lceil K/\epsilon \rceil + 1}^{T} \mathbb{1}\left(\mathbf{p}(t) = \mathbf{p}_l, c_t(p_k) > c_{t-1}(p_k), c_{t-1}(p_k) \leq C_T(\Delta_l^{p_k})\right).$$

Clearly, we have $c_T(p_k) = 1 + \sum_{l \in I(p_k)} \left(c_T^{l,\mathbf{suf}}(p_k) + c_T^{l,\mathbf{und}}(p_k)\right)$. With these notations, we can write (12) as follows:

$$\text{Reg}_\epsilon(T) \leq \mathbb{E}\left[\sum_{p_k} \left(\Delta_{\max}^{p_k} + \sum_{l \in [I(p_k)]} \left(c_T^{l,\mathbf{suf}}(p_k) + c_T^{l,\mathbf{und}}(p_k)\right) \cdot \Delta_l^{p_k}\right)\right]. \tag{13}$$

The proof of this lemma will complete after establishing following two claims:

Claim 1: $\mathbb{E}\left[\sum_{p_k}\sum_{l\in[I(p_k)]}c_T^{l,\mathtt{suf}}(p_k)\right]\leq\lceil K/\epsilon\rceil\cdot\mathcal{O}(1).$ (14)

Claim 2: $\mathbb{E}\left[\sum_{p_k}\sum_{l\in[I(p_k)]}c_T^{l,\mathtt{und}}(p_k)\Delta_l^{p_k}\right]\leq\sum_{p_k}\left((4K\ln T)/\Delta_{\min}^{p_k}+4K\ln T\left(1/\Delta_{\min}^{p_k}-1/\Delta_{\max}^{p_k}\right)\right).$ (15)

We now first prove the Claim 1 as in (14), i.e., for any $t>\lceil K/\epsilon\rceil$, we have following upper bound over counters of sufficiently selected discretized arms. To see this, by definition of $c_T^{l,\mathtt{suf}}(p_k)$, it reduces to show that for any $T\geq t>\lceil K/\epsilon\rceil$,

$$\mathbb{E}\left[\sum_{p_k}\sum_{l\in[I(p_k)]}\mathbb{1}\left(\mathbf{p}(t)=\mathbf{p}_l,c_t(p_k)>c_{t-1}(p_k)>C_T(\Delta_l^{p_k})\right)\right]$$
$$=\sum_{p_k}\sum_{l\in[I(p_k)]}\mathbb{P}\left(\mathbf{p}(t)=\mathbf{p}_l,p_k=p_{\min}(\mathbf{p}_l);\forall p\in\mathbf{p}_l,c_{t-1}(p)>C_T(\Delta_l^{p_k})\right)$$
$$\overset{(a)}{\leq}\lceil 4K/\epsilon\rceil\cdot t^{-2},$$

where the last step (a) is due to Lemma D.2, thus (14) follows from a simple series bound.

We now proceed to analyze the discretized arms that are not sufficiently included in the meta arm chosen by the algorithm and prove the Claim 2 as in (15). For any under-selected discretized arm $p_k$, its counter $c(p_k)$ will increase from 1 to $C_T(\Delta_{\min}^{p_k})$. To simplify the notation, we set $C_T(\Delta_0^{p_k})=0$. Suppose that at round $t$, $c(p_k)$ is incremented, and $c_{t-1}(p_k)\in(C_T(\Delta_{j-1}^{p_k}),C_T(\Delta_j^{p_k})]$ for some $j\in[I(p_k)]$. Notice that we are only interested in the case that $p_k$ is under-selected. In particular, if this is indeed the case, $\mathbf{p}(t)=\mathbf{p}_l$ for some $l\geq j$. (Otherwise, $\mathbf{p}(t)$ is sufficiently selected based on the counter value $c_{t-1}(p_k)$.) Thus, we will suffer a regret of $\Delta_l^{p_k}\leq\Delta_j^{p_k}$ (step (a)). As a result, for counter $c_t(p_k)\in(C_T(\Delta_{j-1}^{p_k}),C_T(\Delta_j^{p_k})]$, we will suffer a total regret for those playing suboptimal meta arms that include under-selected discretized arms at most $(C_T(\Delta_j^{p_k})-C_T(\Delta_{j-1}^{p_k}))\cdot\Delta_j^{p_k}$ in rounds that $c_t(p_k)$ is incremented (step (b)). In what follows we establish the above analysis rigorously.

$$\sum_{l\in[I(p_k)]}c_T^{l,\mathtt{und}}(p_k)\Delta_l^{p_k}$$
$$=\sum_{t=\lceil K/\epsilon\rceil+1}^{T}\sum_{l\in[I(p_k)]}\mathbb{1}\left(\mathbf{p}(t)=\mathbf{p}_l,c_t(p_k)>c_{t-1}(p_k),c_{t-1}(p_k)\leq C_T(\Delta_l^{p_k})\right)\cdot\Delta_l^{p_k}$$
$$=\sum_{t=\lceil K/\epsilon\rceil+1}^{T}\sum_{l\in[I(p_k)]}\sum_{j=1}^{l}\mathbb{1}\left(\mathbf{p}(t)=\mathbf{p}_l,c_t(p_k)>c_{t-1}(p_k),c_{t-1}(p_k)\in(C_T(\Delta_{j-1}^{p_k}),C_T(\Delta_j^{p_k})]\right)\cdot\Delta_l^{p_k}$$
$$\overset{(a)}{\leq}\sum_{t=\lceil K/\epsilon\rceil+1}^{T}\sum_{l\in[I(p_k)]}\sum_{j=1}^{l}\mathbb{1}\left(\mathbf{p}(t)=\mathbf{p}_l,c_t(p_k)>c_{t-1}(p_k),c_{t-1}(p_k)\in(C_T(\Delta_{j-1}^{p_k}),C_T(\Delta_j^{p_k})]\right)\cdot\Delta_j^{p_k}$$
$$\leq\sum_{t=\lceil K/\epsilon\rceil+1}^{T}\sum_{l,j\in[I(p_k)]}\mathbb{1}\left(\mathbf{p}(t)=\mathbf{p}_l,c_t(p_k)>c_{t-1}(p_k),c_{t-1}(p_k)\in(C_T(\Delta_{j-1}^{p_k}),C_T(\Delta_j^{p_k})]\right)\cdot\Delta_j^{p_k}$$
$$=\sum_{t=\lceil K/\epsilon\rceil+1}^{T}\sum_{j\in[I(p_k)]}\mathbb{1}\left(\mathbf{p}(t)\in\mathcal{S}(p_k),c_t(p_k)>c_{t-1}(p_k),c_{t-1}(p_k)\in(C_T(\Delta_{j-1}^{p_k}),C_T(\Delta_j^{p_k})]\right)\cdot\Delta_j^{p_k}$$
$$\overset{(b)}{\leq}\sum_{j\in[I(p_k)]}(C_T(\Delta_j^{p_k})-C_T(\Delta_{j-1}^{p_k}))\cdot\Delta_j^{p_k}.$$

Now, we can compute the regret incurred by selecting the meta arm which includes under-selected discretized arms:

$$\sum_{p_k} \sum_{l \in [I(p_k)]} c_T^{l,\text{und}}(p_k) \Delta_l^{p_k} \leq \sum_{p_k} \sum_{j \in [I(p_k)]} (C_T(\Delta_j^{p_k}) - C_T(\Delta_{j-1}^{p_k})) \cdot \Delta_j^{p_k}$$

$$= \sum_{p_k} \left( C_T(\Delta_{\min}^{p_k}) \Delta_{\min}^{p_k} + \sum_{j \in [I(p_k)-1]} C_T(\Delta_j^{p_k}) \cdot (\Delta_j^{p_k} - \Delta_{j+1}^{p_k}) \right)$$

$$\leq \sum_{p_k} \left( C_T(\Delta_{\min}^{p_k}) \Delta_{\min}^{p_k} + \int_{\Delta_{\min}^{p_k}}^{\Delta_{\max}^{p_k}} C_t(x) dx \right)$$

$$= \sum_{p_k} \left( \frac{4K \ln T}{\Delta_{\min}^{p_k}} + 4K \ln T \left( \frac{1}{\Delta_{\min}^{p_k}} - \frac{1}{\Delta_{\max}^{p_k}} \right) \right). \tag{16}$$

Equipped with the above set of results, the bound of regret (13) follows by combing the bounds in (14) and (15). □

To achieve instance-independent regret bound, we need to deal with the case when the meta-arm gap $\Delta_{\min}^{p_k}$ is too small, leading the regret to approach infinite. Nevertheless, one can still show that when $\Delta_{\min}^{p_k} \leq 1/\sqrt{T}$, the regret contributed by this scenario scales at most $\mathcal{O}(\sqrt{T})$ at time horizon $T$.

**Lemma D.5.** *Following the UCB designed in (11), we have:* $\text{Reg}_\epsilon(T) \leq \mathcal{O}\big(K\sqrt{T \ln T/\epsilon}\big).$

*Proof.* Following the proof of Lemma D.4, we only need to consider the meta arms that are played when they are under-sampled. We particularly need to deal with the situation when $\Delta_{\min}^{p_k}$ is too small. We measure the threshold for $\Delta_{\min}^{p_k}$ based on $c_T(p_k)$, i.e., the counter of disretized arm $p_k$ at time horizon $T$. Let $\{T(p_k), \forall p_k\}$ be a set of possible counter values at time horizon $T$. Our analysis will then be conditioned on the event that $\mathcal{E}(p_k) = \{c_T(p_k) = T(p_k)\}$. By definition,

$$\mathbb{E}\big[ \sum_{l \in [I(p_k)]} c_T^{l,\text{und}}(p_k) \cdot \Delta_l^{p_k} \mid \mathcal{E}(p_k) \big]$$

$$= \sum_{t=\lceil K/\epsilon \rceil+1}^{T} \sum_{l \in [I(p_k)]} \mathbb{1}\left(\mathbf{p}(t) = \mathbf{p}_l, c_t(p_k) > c_{t-1}(p_k), c_{t-1}(p_k) \leq C_T(\Delta_l^{p_k}) \mid \mathcal{E}(p_k)\right) \cdot \Delta_l^{p_k}.$$
$$\tag{17}$$

We define $\Delta^*(T(p_k)) := \left( \frac{4K \ln T}{T(p_k)} \right)^{1/2}$, i.e., $C_T(\Delta^*(T(p_k))) = T(p_k)$. To achieve *instance-independent* regret bound, we consider following two cases:
**Case 1:** $\Delta_{\min}^{p_k} > \Delta^*(T(p_k))$, we thus have

$$\mathbb{E}\big[ \sum_{l \in [I(p_k)]} c_T^{l,\text{und}}(p_k) \cdot \Delta_l^{p_k} \mid \mathcal{E}(p_k) \big] \leq \mathcal{O}\left( \sqrt{4K \ln T \cdot T(p_k)} \right). \tag{18}$$

**Case 2:** $\Delta_{\min}^{p_k} < \Delta^*(T(p_k))$. Let $l^* := \min\{l \in I(p_k) : \Delta_l^{p_k} > \Delta^*(T(p_k))\}$. Observe that we have $\Delta_{l^*}^{p_k} \leq \Delta^*(T(p_k))$ and the counter $c(p_k)$ never go beyond $T(p_k)$, we thus have

$$(17) \leq (C_T(\Delta^*(T(p_k))) - C_T(\Delta_{l^*-1}^{p_k})) \cdot \Delta^*(T(p_k)) + \sum_{j \in [l^*-1]} (C_T(\Delta_j^{p_k}) - C_T(\Delta_{j-1}^{p_k})) \cdot \Delta_j^{p_k}$$

$$\leq C_T(\Delta^*(T(p_k))) \cdot \Delta^*(T(p_k)) + \int_{\Delta^*(T(p_k))}^{\Delta_{\max}^{p_k}} C_T(x) dx \leq \mathcal{O}\left( \sqrt{K \ln T \cdot T(p_k)} \right). \tag{19}$$

Thus, combining (18) and (19), we have

$$\mathbb{E}\big[ \sum_{p_k : \Delta_{\min}^{p_k} > 0} \sum_{l \in [I(p_k)]} c_T^{l,\text{und}}(p_k) \cdot \Delta_l^{p_k} \mid \mathcal{E}(p_k) \big] \leq \sum_{p_k : \Delta_{\min}^{p_k} > 0} \mathcal{O}(\sqrt{K \ln T \cdot T(p_k)})$$

$$\overset{(a)}{\leq} \mathcal{O}(K\sqrt{T \ln T/\epsilon}),$$

where (a) is by Jesen's inequality and $\sum_{p_k} T(p_k) \leq KT/\epsilon$. Put all pieces together, we have the instance-independent regret bound as stated in the lemma. Observe that the final inequality does not depend on the event $\mathcal{E}(p_k)$, we thus can drop this conditional expectation. □

With the above lemma in hand, picking $\epsilon = \Theta((\ln T/T)^{1/3})$ will give us desired result in Theorem 4.2. [10]

**Remark D.6.** *When only one arm is activated according to* $\mathbf{p}(t)$*, the Hoeffding's inequality is adapted as follows:*

$$\mathbb{P}\big(|\overline{U}_t(\mathbf{p}) - U(\mathbf{p})| \geq \delta\big) \leq \sum_k \mathbb{P}\big(|p_k \bar{r}(p_k) - p_k r(p_k)| \geq \delta/K\big)$$

$$\leq \sum_k 2 \exp\big(-2\delta^2 n_t(p_k)/K^2\big) \leq 2K \exp\big(-2\delta^2 n_t(p_{\min}(\mathbf{p}))/K^2\big).$$

*The below analysis carries over with accordingly changing* $\delta = \sqrt{\frac{K \ln t}{n_t(p_{\min}(\mathbf{p}))}}$ *to* $\delta = \sqrt{\frac{K^2 \ln(\sqrt{K}t)}{n_t(p_{\min}(\mathbf{p}))}}$, *and the condition of* $n_t(p_{\min}(\mathbf{p}))$ *in Lemma D.2 is changed to* $4K^2 \ln(\sqrt{K}t)/\Delta_{\mathbf{p}}^2$ *to account for larger* $\delta$*. As a result, the instance-independent regret bound in Lemma D.5 is changed to* $\mathcal{O}\left(K\sqrt{KT\ln(\sqrt{K}T)/\epsilon}\right)$*. Together with the discretization error, one can then optimize the choice of* $\epsilon$ *to get* $\tilde{\mathcal{O}}(K^{4/3}T^{2/3})$ *regret bound.*

### D.2.1 Regret Bound Comparison with [13]

In the work [13], the authors study the setting when pulling the meta arm, each base arm in (or possibly other base arm) this meta arm will be triggered and played as a result. Back to our setting, this is saying that when pulling a meta arm $\mathbf{p} = (p_1, \ldots, p_K)$, each base arm $k$ will be triggered with its corresponding probability (discretized arm) $p_k$. The authors in [13] discuss a general setting which allows complex reward structure where only requires two mild conditions. In particular, one of the condition they need for expected reward of playing a meta arm is the bounded smoothness (cf., Definition 1 in [13].). In the Theorem 2 of [13], the authors give results when the function used to characterize bounded smoothness is $f(x) = \gamma \cdot x^\omega$ for some $\gamma > 0$ and $\omega \in (0,1]$. In more detail, they achieve a regret bound $\mathcal{O}\left(\frac{2\gamma}{2-\omega}\left(\frac{12|\mathcal{M}|\ln T}{p^*}\right)^{\omega/2} \cdot T^{1-\omega/2} + |\mathcal{M}| \cdot \Delta_{\max}\right)$ where $p^* \in (0,1)$ is the minimum triggering probability across all base arms and $\Delta_{\max}$ is the largest badness of the suboptimal meta arm in discretized space. [11] Adapt to our setting, by inspection, we have $\gamma = L^*, \omega = 1, p^* = \epsilon, |\mathcal{M}| = \Theta(K/\epsilon)$, and $\Delta_{\max} = \Theta(KL^*)$. Substituting these values to the above bound, ignoring constant factors and combining with the discretization error, we have

$$\mathcal{O}\left(\left(\frac{K\ln T}{\epsilon^2}\right)^{1/2} \cdot T^{1/2} + K^2/\epsilon\right) + \mathcal{O}(TK\epsilon).$$

Picking $\epsilon = \Theta(\ln T/(KT))^{1/4}$ will give us result.

## E Proof of Theorem 5.1 for History-dependent Bandits

In this section, we provide the analysis of Theorem 5.1. The analysis follows a similar structure to the one used in the proof of the regret bound in Theorem 4.2. However, due to the existence of historical bias, we need to perform a careful computation when handling the high-probability bounds. Specifically, we need to prove that, after deploying $\mathbf{p}$ consecutively for moderate long rounds (tuning $s_a$), the approximation error $|U(\mathbf{p}) - \overline{U}_m^{\text{est}}(\mathbf{p})|$ is small enough. The analysis is provided below.

**Step 1: Bounding the small error of** $|U(\mathbf{p}) - \overline{U}_m^{\text{est}}(\mathbf{p})|$ **with high-probability.** Our first step is to ensure the empirical mean reward estimation we obtain from the information we collected in all the estimation stages will approximate well the true mean of meta arm we want to deploy.

To return a high-probability error bound, we first bound the approximation error incurred due to the dependency of history of arm selection ("historical bias"). This is summarized below.

---

[10]Here the choice of $\epsilon$ absorbs Lipschitz constant of $r_k(\cdot)$.

[11]For simplicity, the bound we present here omits a non-significant term.

**Lemma E.1.** *Keeping deploying* $\mathbf{p} = \{p_1, \ldots, p_K\}$ *in the approaching stage with* $s_a$ *rounds, and collect all reward feedback in the following estimation stage for the empirical estimation of rewards generated by* $\mathbf{p}$*, one can bound the approximation error as follows:*

$$\mathbb{E}\left[\left|\overline{U}_m^{est}(\mathbf{p}) - \overline{U}(\mathbf{p})\right|\right] \leq K\gamma^{s_a}(L^* + 1),$$

*where* $\overline{U}(\mathbf{p})$ *denote the empirical mean of rewards if the instantaneous reward is truly sampled from mean reward function according to* $\mathbf{p}$*.*

*Proof.* The proof of this lemma is mainly built on analyzing the convergence of $\mathbf{p}^{(\gamma)}$ via pulling the base arms with the same probability consistently. For the ease of presentation, let us suppose $t = mL$ and let $t_m^{\text{est}} := \frac{t}{L}(L - s_a) = m(L - s_a)$ be the total number of estimation rounds in the first $m$ phases. Thus, at the end of the approaching stage, we have

$$\widehat{p}_k^{(\gamma)}(t + s_a) = \frac{p_k(t + s_a)\gamma^0 + \ldots + p_k(t + 1)\gamma^{s_a - 1} + (1 + \gamma + \ldots + \gamma^{t-1})\gamma^{s_a}\widehat{p}_k^{(\gamma)}(t)}{1 + \gamma + \ldots + \gamma^{t + s_a - 1}},$$

where $\widehat{p}_k^{(\gamma)}(t) = \frac{p_k(t)\gamma^0 + \ldots + p_k(1)\gamma^{t-1}}{1 + \gamma + \ldots + \gamma^{t-1}}$. Recall that during the approaching stage, we consistently pull arm $k$ with the same probability $p_k$. Thus, the approximation error of $\widehat{p}_k^{(\gamma)}(t + s_a)$ w.r.t. $p_k$ can be computed as:

$$\left|\widehat{p}_k^{(\gamma)}(t + s_a) - p_k\right| = \left|\frac{p_k(1 - \gamma^{s_a}) + \widehat{p}_k^{(\gamma)}(t)\gamma^{s_a}(1 - \gamma^t)}{1 - \gamma^{t + s_a}} - p_k\right| \leq \frac{\gamma^{s_a}(1 - \gamma^t)}{1 - \gamma^{t + s_a}} < \gamma^{s_a}.$$

Recall that $U(\mathbf{p}) = \sum_{p_k \in \mathbf{p}} p_k r_k(p_k)$. In the estimation stage, we approximate all the realized utility as the utility generated by the meta arm $\mathbf{p}$. However, note that we actually cannot compute the empirical value of $\overline{U}(\mathbf{p})$, instead, we use $\overline{U}_m^{\text{est}}(\mathbf{p}(t + s_a))$ of each phase as an approximation of $\overline{U}(\mathbf{p})$, i.e., we approximate all $\mathbf{p}^{(\gamma)}(t + s), \forall s \in (s_a, L]$ as $\mathbf{p}(t + s_a)$ and use $\mathbf{p}(t + s_a)$ as the approximation of $\mathbf{p}$. Recall that for any $s \in (s_a, L]$, we have:

$$\left|\widehat{p}_k^{(\gamma)}(t + s) - p_k\right| = \left|\frac{\gamma^s(1 - \gamma^t)(\widehat{p}_k^{(\gamma)}(t) - p_k)}{1 - \gamma^{t + s}}\right| \leq \frac{\gamma^s(1 - \gamma^t)}{1 - \gamma^{t + s}} < \frac{\gamma^{s_a}(1 - \gamma^t)}{1 - \gamma^{t + s_a}} < \gamma^{s_a}.$$

Thus, the approximation error on the empirical estimation can be computed as follows:

$$\mathbb{E}\left[\left|\overline{U}_m^{\text{est}}(\mathbf{p}(t + s_a)) - \overline{U}(\mathbf{p})\right|\right] = \mathbb{E}\left[\left|\sum_{p_k^{(\gamma)} \in \mathbf{p}(t + s_a)} p_k^{(\gamma)}\bar{r}_{t+s_a}^{\text{est}}(p_k^{(\gamma)}) - \sum_{p_k \in \mathbf{p}} p_k \bar{r}_{t+s_a}^{\text{est}}(p_k)\right|\right]$$

$$= \left|\sum p_k^{(\gamma)}\mathbb{E}\left[\bar{r}_{t+s_a}^{\text{est}}(p_k^{(\gamma)})\right] - \sum p_k\mathbb{E}\left[\bar{r}_{t+s_a}^{\text{est}}(p_k)\right]\right|$$

$$= \left|\sum p_k^{(\gamma)} r_k(p_k^{(\gamma)}) - \sum p_k r_k(p_k)\right|$$

$$= \left|\sum \left(p_k^{(\gamma)}\left(r_k(p_k^{(\gamma)}) - r_k(p_k)\right) + r_k(p_k)(p_k^{(\gamma)} - p_k)\right)\right|$$

$$\leq \sum \left|\gamma^{s_a} L_k p_k^{(\gamma)} + r_k(p_k)\gamma^{s_a}\right| \leq K\gamma^{s_a}(L^* + 1).$$

$\square$

With the approximation error at hand, we can then bound the error of $\left|U(\mathbf{p}) - \overline{U}_m^{\text{est}}(\mathbf{p})\right|$ with high probability:

**Lemma E.2.** *With probability at least* $1 - \frac{6}{\left(L\rho m\right)^2}$*, we have*

$$\left|U(\mathbf{p}) - \overline{U}_m^{est}(\mathbf{p})\right| \leq err + 3\sqrt{\frac{K\ln\left(L\rho m\right)}{n_m^{est}(p_{\min}(\mathbf{p}))}},$$

*where* $p_{\min}(\mathbf{p}) = \arg\min_{p_k \in \mathbf{p}} n_m^{est}(p_k)$*.*

*Proof.* We first decompose $\left|U(\mathbf{p}) - \overline{U}_m^{\text{est}}(\mathbf{p}_e^{(\gamma)})\right|$ as $\left|U(\mathbf{p}) - \overline{U}(\mathbf{p})\right| + \left|\overline{U}(\mathbf{p}) - \overline{U}_m^{\text{est}}(\mathbf{p})\right|$ and then apply union bound.

$$\mathbb{P}\left(\left|U(\mathbf{p}) - \overline{U}_m^{\text{est}}(\mathbf{p}(t+s_a))\right| \geq \delta\right)$$

$$\leq \mathbb{P}\left(\left|U(\mathbf{p}) - \overline{U}(\mathbf{p})\right| + \left|\overline{U}(\mathbf{p}) - \overline{U}_m^{\text{est}}(\mathbf{p}(t+s_a))\right| \geq \delta\right) \qquad \text{By triangle inequality}$$

$$= \mathbb{P}\Bigg(\left|U(\mathbf{p}) - \overline{U}(\mathbf{p})\right| + \left|\overline{U}_m^{\text{est}}(\mathbf{p}(t+s_a)) - \mathbb{E}[\overline{U}_m^{\text{est}}(\mathbf{p}(t+s_a))] - \right.$$

$$\left. (\overline{U}(\mathbf{p}) - \mathbb{E}[\overline{U}(\mathbf{p})]) + \mathbb{E}[\overline{U}(\mathbf{p})] - \mathbb{E}[\overline{U}_m^{\text{est}}(\mathbf{p}(t+s_a))]\right| \geq \delta\Bigg)$$

$$\leq \mathbb{P}\Bigg(2\left|U(\mathbf{p}) - \overline{U}(\mathbf{p})\right| + \left|\overline{U}_m^{\text{est}}(\mathbf{p}(t+s_a)) - \mathbb{E}[\overline{U}_m^{\text{est}}(\mathbf{p}(t+s_a))]\right| \geq \delta - \texttt{err}\Bigg)$$

$$\overset{(a)}{\leq} 3\mathbb{P}\left(\left|U(\mathbf{p}) - \overline{U}(\mathbf{p})\right| \geq \frac{\delta - \texttt{err}}{3}\right) \leq 6\exp\left(-\frac{2n_m^{\text{est}}(p_{\min}(\mathbf{p}))(\delta - \texttt{err})^2}{9K}\right),$$

where in step (a), we use the Hoeffding's Inequality on Weighted Sums and Lemma E.1. $\qquad \square$

**Step 2: Bounding the probability on deploying suboptimal meta arm.** Till now, with the help of the above high probability bound on the empirical reward estimation, the history-dependent reward bandit setting is largely reduced to an action-dependent one with a certain approximation error. Then, similar to our argument on upper bound of action-dependent bandits, we have the following specific Lemma for history-dependent bandits:

**Lemma E.3.** *At the end of each phase, for a suboptimal meta arm* $\mathbf{p}$*, if it satisfies* $n_m^{est}(p_{\min}(\mathbf{p})) \geq \frac{9K\ln\left(L\rho m\right)}{\left(\Delta_{\mathbf{p}}/2 - err\right)^2}$*, then with the probability at least* $1 - \frac{12}{\left(L\rho m\right)^2}$*, we have* $\texttt{UCB}_m(\mathbf{p}) < \texttt{UCB}_m(\mathbf{p}^*)$*, i.e.,*

$$\mathbb{P}\left(\mathbf{p}(m+1) = \mathbf{p}\,\Big|\,n_m^{est}(p_{\min}(\mathbf{p})) \geq \frac{9K\ln\left(L\rho m\right)}{\left(\frac{\Delta_{\mathbf{p}}}{2} - err\right)^2}\right) \leq \frac{12}{\left(L\rho m\right)^2}.$$

*Proof.* To prove the above lemma, we construct two high-probability events. Event 1 corresponds to that the UCB index of each meta arm concentrates on the true mean utility of $\mathbf{p}$; Event 2 corresponds to that the empirical mean utility of each approximated meta arm $\mathbf{p}^{(\gamma)}$ concentrates on the true mean utility of $\mathbf{p}$. The probability of Event 1 or Event 2 not holding is at most $4/t^2$. By the definition of the constructed UCB, we'll have

$$\texttt{UCB}_m(\mathbf{p}) = \overline{U}_m^{\text{est}}(\mathbf{p}(t+s_a)) + \texttt{err} + 3\sqrt{\frac{K\ln\left(L\rho m\right)}{n_m^{\text{est}}(p_{\min}(\mathbf{p}))}} \overset{(a)}{\leq} \overline{U}_m^{\text{est}}(\mathbf{p}(t+s_a)) + \Delta_{\mathbf{p}}/2$$

$$\overset{(b)}{<} (U(\mathbf{p}) + \Delta_{\mathbf{p}}/2) + \Delta_{\mathbf{p}}/2 \qquad\qquad\qquad\qquad \text{By Event 1}$$

$$= U(\mathbf{p}_\epsilon^*) \overset{(c)}{<} \texttt{UCB}_m(\mathbf{p}_\epsilon^*), \qquad\qquad\qquad\qquad\qquad \text{By Event 2}$$

where the first inequality (a) is due to $n_m^{\text{est}}(p_{\min}(\mathbf{p})) \geq \frac{9K\ln(L\rho m)}{(\Delta_{\mathbf{p}}/2 - \texttt{err})^2}$, and the probability of step (b) or (c) not holding is at most $12/(L\rho m)^2$. $\qquad \square$

The above lemma implies that we will stop deploying suboptimal meta arm $\mathbf{p}$ and further prevent it from incurring regret as we gather more information about it such that $\texttt{UCB}_m(\mathbf{p}) < \texttt{UCB}_m(\mathbf{p}_\epsilon^*)$.

**Step 3: Bounding the $\mathbb{E}[n_m^{\text{est}}(p_{\min}(\mathbf{p}))]$.** The results we obtain in Step 2 implies following guarantee:

**Lemma E.4.** *For each suboptimal meta arm* $\mathbf{p} \neq \mathbf{p}^*$*, we have following:*

$$\mathbb{E}[n_m^{est}(p_{\min}(\mathbf{p}))] \leq \frac{9K\ln\left(L\rho m\right)}{\left(\Delta_{\mathbf{p}}/2 - err\right)^2} + \frac{2\pi^2}{L - s_a}.$$

*Proof.* For notation simplicity, suppose $t = mL$. For each suboptimal arm $\mathbf{p} \neq \mathbf{p}_\epsilon^*$, and suppose there exists $p_{\min}(\mathbf{p}) \notin \mathbf{p}_\epsilon^*$ such that $p_{\min}(\mathbf{p}) = \arg\min_{p_k \in \mathbf{p}} n_t^{\mathtt{est}}(p_k)$, then

$$\mathbb{E}[n_t^{\mathtt{est}}(p_{\min}(\mathbf{p}))]$$

$$= (L - s_a)\mathbb{E}\left[\sum_{i=1}^{m} \mathbb{1}\left(\mathbf{p}(i) = \mathbf{p}, \mathbf{p} \in \mathcal{S}(p_{\min}(\mathbf{p}))\right)\right]$$

$$= (L - s_a)\mathbb{E}\left[\sum_{i=1}^{m} \mathbb{1}\left(\mathbf{p}(i) = \mathbf{p}, \mathbf{p} \in \mathcal{S}(p_{\min}(\mathbf{p})); n_i^{\mathtt{est}}(p_{\min}(\mathbf{p})) < \frac{9K \ln\left(i(L - s_a)\right)}{(\Delta_{\mathbf{p}}/2 - \mathtt{err})^2}\right)\right] +$$

$$(L - s_a)\mathbb{E}\left[\sum_{i=1}^{m} \mathbb{1}\left(\mathbf{p}(i) = \mathbf{p}, \mathbf{p} \in \mathcal{S}(p_{\min}(\mathbf{p})); n_i^{\mathtt{est}}(p_{\min}(\mathbf{p})) \geq \frac{9K \ln\left(i(L - s_a)\right)}{(\Delta_{\mathbf{p}}/2 - \mathtt{err})^2}\right)\right]$$

$$\overset{(a)}{\leq} \frac{9K \ln\left(t_m^{\mathtt{est}}\right)}{(\Delta_{\mathbf{p}}/2 - \mathtt{err})^2} + (L - s_a)\mathbb{E}\left[\sum_{i=1}^{m} \mathbb{1}\left(\mathbf{p}(i) = \mathbf{p}, \mathbf{p} \in \mathcal{S}(p_{\min}(\mathbf{p})); n_i^{\mathtt{est}}(p_{\min}(\mathbf{p})) \geq \frac{9K \ln\left(i(L - s_a)\right)}{(\Delta_{\mathbf{p}}/2 - \mathtt{err})^2}\right)\right]$$

$$= \frac{9K \ln\left(t_m^{\mathtt{est}}\right)}{(\Delta_{\mathbf{p}}/2 - \mathtt{err})^2} + (L - s_a)\sum_{i=1}^{m} \mathbb{P}\left(\mathbf{p}(i) = \mathbf{p}, \mathbf{p} \in \mathcal{S}(p_{\min}(\mathbf{p})) \middle| n_i^{\mathtt{est}}(p_{\min}(\mathbf{p})) \geq \frac{9K \ln\left(i(L - s_a)\right)}{(\Delta_{\mathbf{p}}/2 - \mathtt{err})^2}\right) \cdot$$

$$\mathbb{P}\left(n_i^{\mathtt{est}}(p_{\min}(\mathbf{p})) \geq \frac{9K \ln\left(i(L - s_a)\right)}{(\Delta_{\mathbf{p}}/2 - \mathtt{err})^2}\right)$$

$$\leq \frac{9K \ln\left(t_m^{\mathtt{est}}\right)}{(\Delta_{\mathbf{p}}/2 - \mathtt{err})^2} + (L - s_a)\sum_{i=1}^{m} \frac{12}{(i(L - s_a))^2} \leq \frac{9K \ln\left(t_m^{\mathtt{est}}\right)}{(\Delta_{\mathbf{p}}/2 - \mathtt{err})^2} + \frac{2\pi^2}{L - s_a}.$$

In step (a), suppose for contradiction that the indicator $\mathbb{1}\left(\mathbf{p}(i) = \mathbf{p}, \mathbf{p} \in \mathcal{S}(p_{\min}(\mathbf{p})); n_i^{\mathtt{est}}(p_{\min}(\mathbf{p})) < S\right)$ takes value of 1 at more than $S - 1$ time steps, where $S = \frac{9K \ln(i(S - s_a))}{(\Delta_{\mathbf{p}}/2 - \mathtt{err})^2}$. Let $\tau$ be the phase at which this indicator is 1 for the $(S - 1)$-th phase. Then the number of pulls of all meta arms in $\mathcal{S}(p_{\min}(\mathbf{p}))$ is at least $L$ times until time $\tau$ (including the initial pull), and for all $i > \tau$, $n_i(p_{\min}(\mathbf{p})) \geq S$ which implies $n_i^{\mathtt{est}}(p_{\min}(\mathbf{p})) \geq \frac{9K \ln(i(S - s_a))}{(\Delta_{\mathbf{p}}/2 - \mathtt{err})^2}$. Thus, the indicator cannot be 1 for any $i \geq \tau$, contradicting the assumption that the indicator takes value of 1 more than $S$ times. This bounds $1 + \mathbb{E}\left[\sum_{i=1}^{m} \mathbb{1}\left(\mathbf{p}(i) = \mathbf{p}, \mathbf{p} \in \mathcal{S}(p_{\min}(\mathbf{p})); n_i^{\mathtt{est}}(p_{\min}(\mathbf{p})) < S\right)\right]$ by $S$. $\qquad\square$

**Wrapping up: Proof of Theorem 5.1.** Following the similar analysis in Section 3, we can also get an instance-dependent regret bound for history-dependent bandits:

**Lemma E.5.** *Following the UCB designed in Algorithm 3, we have following instance-dependent regret on discretized arm space for history-dependent bandits:*

$$\mathrm{Reg}_\epsilon(T) \leq \mathcal{O}\left(\frac{K\Delta_{\max}}{L\epsilon\rho^2}\right) + \sum_{p_k}\left(\frac{9K \ln\left(T\rho\right)}{\rho}\left(\frac{\Delta_{\min}^{p_k}}{(\Delta_{\min}^{p_k}/2 - \boldsymbol{err})^2} + \frac{2}{\Delta_{\min}^{p_k}/2 - \boldsymbol{err}}\right)\right).$$

*Proof.* For notation simplicity, we include all initialization rounds to phase 0 and suppose the time horizon $T = ML$. Note that by definitions, we can compute the regret $\mathrm{Reg}_\epsilon(T)$ as follows:

$$\mathrm{Reg}_\epsilon(T) = \sum_{\mathbf{p} \in \mathcal{P}_\epsilon} \mathbb{E}[N_T(\mathbf{p})]\Delta_{\mathbf{p}} \leq \sum_{p_k} \sum_{\mathbf{p}_l \in \mathcal{S}(p_k)} \mathbb{E}[N_T(\mathbf{p}_l)]\Delta_l^{p_k}. \tag{20}$$

where $N_t(\mathbf{p}) = K + L \sum_{m=1}^{M} \mathbb{1}\left(\mathbf{p}(m) = \mathbf{p}\right)$, where $K$ here accounts for the initialization. Follow the same analysis in action-dependent bandits, we can also define a counter $c^{\mathtt{est}}(p_k)$ for each discretized arm $p_k$ and the value of $c^{\mathtt{est}}(p_k)$ at phase $m$ is denoted by $c_m^{\mathtt{est}}(p_k)$. But different from the action-dependent bandit setting, we update the counter $c^{\mathtt{est}}(p_k)$ only when we start a new phase. In particular, for a phase $m \geq 1$, let $\mathbf{p}(m)$ be the meta arm selected in the phase $m$ by the algorithm. Let $p_k = \arg\min_{p_k \in \mathbf{p}(m)} c_m^{\mathtt{est}}(p_k)$. We increment $c_m^{\mathtt{est}}(p_k)$ by one, i.e., $c_m^{\mathtt{est}}(p_k) = c_{m-1}^{\mathtt{est}}(p_k) + 1$. In other words, we find the discretized arm $p_k$ with the smallest counter in $\mathbf{p}(m)$ and increment its counter. If such $p_k$ is not unique, we pick an arbitrary discretized arm with the smallest counter. Note

that the initialization gives $\sum_{p_k} c_0^{\text{est}}(p_k) = \lceil K/\epsilon \rceil$. It is easy to see that for any $p_k = p_{\min}(\mathbf{p})$, we have $n_m(p_k) = L\rho \cdot c_m(p_k)$.

Like in action-dependent bandits, we also define $C_M^{\text{est}}(\Delta) := \frac{9K\ln(ML\rho)}{L\rho(\Delta/2-\text{err})^2}$, the number of selection that is considered sufficient for a meta arm with reward $\Delta$ away from the optimal strategy $\mathbf{p}_\epsilon^*$ with respect to phase horizon $M$. With the above notations, we define following two situations for the counter of each discretized arm:

$$c_M^{\text{est},l,\text{suf}}(p_k) := \sum_{m=1}^M \mathbb{1}\left(\mathbf{p}(m) = \mathbf{p}_l, c_m^{\text{est}}(p_k) > c_{m-1}^{\text{est}}(p_k) > C_M^{\text{est}}(\Delta_l^{p_k})\right) \tag{21}$$

$$c_M^{\text{est},l,\text{und}}(p_k) := \sum_{m=1}^M \mathbb{1}\left(\mathbf{p}(m) = \mathbf{p}_l, c_m^{\text{est}}(p_k) > c_{m-1}^{\text{est}}(p_k), c_{m-1}^{\text{est}}(p_k) \leq C_M^{\text{est}}(\Delta_l^{p_k})\right). \tag{22}$$

Clearly, we have $c_M^{\text{est}}(p_k) = 1 + \sum_{l \in I(p_k)} \left(c_M^{\text{est},l,\text{suf}}(p_k) + c_M^{\text{est},l,\text{und}}(p_k)\right)$. With these notations, we can write (20) as follows:

$$\text{Reg}_\epsilon(T) \leq \mathbb{E}\left[\sum_{p_k}\left(\Delta_{\max}^{p_k} + L \cdot \sum_{l \in [I(p_k)]}\left(c_M^{\text{est},l,\text{suf}}(p_k) + c_M^{\text{est},l,\text{und}}(p_k)\right) \cdot \Delta_l^{p_k}\right)\right]. \tag{23}$$

We now first show that for any $m \geq 1$, we have following upper bound over counters of sufficiently selected discretized arms:

$$\mathbb{E}\left[L \cdot \sum_{p_k}\sum_{l \in [I(p_k)]} c_M^{l,\text{suf}}(p_k)\right] \leq \mathcal{O}\left(\frac{K}{L\epsilon\rho^2}\right). \tag{24}$$

To see this, by definition of $c_M^{\text{est},l,\text{suf}}(p_k)$, it reduces to show that for any $M \geq m > 1$,

$$\mathbb{E}\left[L \cdot \sum_{p_k}\sum_{l \in [I(p_k)]} \mathbb{1}\left(\mathbf{p}(m) = \mathbf{p}_l, c_m^{\text{est}}(p_k) > c_{m-1}^{\text{est}}(p_k) > C_M^{\text{est}}(\Delta_l^{p_k})\right)\right]$$

$$= L \cdot \sum_{p_k}\sum_{l \in [I(p_k)]} \mathbb{P}\left(\mathbf{p}(m) = \mathbf{p}_l, p_k = p_{\min}(\mathbf{p}_l); \forall p \in \mathbf{p}_l, L\rho \cdot c_{m-1}^{\text{est}}(p) > \frac{9K\ln(ML\rho)}{\left(\Delta_l^{p_k}/2 - \text{err}\right)^2}\right)$$

$$\overset{(a)}{\leq} \lceil 12LK/\epsilon \rceil \cdot (ML\rho)^{-2},$$

where the last step (a) is due to Lemma E.3, thus (24) follows from a simple series bound.

We now proceed to analyze the discretized arms that are not sufficiently included in the meta arm chosen by the algorithm. For any under-selected discretized arm $p_k$, its counter $c^{\text{est}}(p_k)$ will increase from 1 to $C_M^{\text{est}}(\Delta_{\min}^{p_k})$. To simplify the notation, we set $C_M^{\text{est}}(\Delta_0^{p_k}) = 0$. Suppose that at phase $m \geq 1$, $c^{\text{est}}(p_k)$ is incremented, and $c_{m-1}^{\text{est}}(p_k) \in (C_M^{\text{est}}(\Delta_{j-1}^{p_k}), C_M^{\text{est}}(\Delta_j^{p_k})]$ for some $j \in [I(p_k)]$. Notice that we are only interested in the case that $p_k$ is under-selected. In particular, if this is indeed the case, $\mathbf{p}(m) = \mathbf{p}_l$ for some $l \geq j$. (Otherwise, $\mathbf{p}(m)$ is sufficiently selected based on the counter value $c_{m-1}^{\text{est}}(p_k)$.) Thus, we will suffer a regret of $\Delta_l^{p_k} \leq \Delta_j^{p_k}$ (step (a)). As a result, for counter $c_m^{\text{est}}(p_k) \in (C_M^{\text{est}}(\Delta_{j-1}^{p_k}), C_M^{\text{est}}(\Delta_j^{p_k})/L]$, we will suffer a total regret for those playing suboptimal meta arms that include under-selected discretized arms at most $(C_M^{\text{est}}(\Delta_j^{p_k}) - C_M^{\text{est}}(\Delta_{j-1}^{p_k})) \cdot \Delta_j^{p_k}$ in rounds that $c_m^{\text{est}}(p_k)$ is incremented (step (b)). In what follows we establish the above analysis

rigorously.

$$\sum_{l\in[I(p_k)]} c_M^{\text{est},l,\text{und}}(p_k)\Delta_l^{p_k}$$

$$= \sum_{m=1}^{M}\sum_{l\in[I(p_k)]} \mathbb{1}\left(\mathbf{p}(m)=\mathbf{p}_l, c_m^{\text{est}}(p_k)>c_{m-1}^{\text{est}}(p_k), c_{m-1}^{\text{est}}(p_k)\le C_M^{\text{est}}(\Delta_l^{p_k})\right)\cdot\Delta_l^{p_k}$$

$$= \sum_{m=1}^{M}\sum_{l\in[I(p_k)]}\sum_{j=1}^{l} \mathbb{1}\left(\mathbf{p}(m)=\mathbf{p}_l, c_m^{\text{est}}(p_k)>c_{m-1}^{\text{est}}(p_k), c_{m-1}^{\text{est}}(p_k)\in(C_M^{\text{est}}(\Delta_{j-1}^{p_k}),C_M^{\text{est}}(\Delta_j^{p_k})]\right)\cdot\Delta_l^{p_k}$$

$$\overset{(a)}{\le} \sum_{m=1}^{M}\sum_{l\in[I(p_k)]}\sum_{j=1}^{l} \mathbb{1}\left(\mathbf{p}(m)=\mathbf{p}_l, c_m^{\text{est}}(p_k)>c_{m-1}^{\text{est}}(p_k), c_{m-1}^{\text{est}}(p_k)\in(C_M^{\text{est}}(\Delta_{j-1}^{p_k}),C_M^{\text{est}}(\Delta_j^{p_k})]\right)\cdot\Delta_j^{p_k}$$

$$\le \sum_{m=1}^{M}\sum_{l,j\in[I(p_k)]} \mathbb{1}\left(\mathbf{p}(m)=\mathbf{p}_l, c_m^{\text{est}}(p_k)>c_{m-1}^{\text{est}}(p_k), c_{m-1}^{\text{est}}(p_k)\in(C_M^{\text{est}}(\Delta_{j-1}^{p_k}),C_M^{\text{est}}(\Delta_j^{p_k})]\right)\cdot\Delta_j^{p_k}$$

$$= \sum_{m=1}^{M}\sum_{j\in[I(p_k)]} \mathbb{1}\left(\mathbf{p}(m)\in\mathcal{S}(p_k), c_m^{\text{est}}(p_k)>c_{m-1}^{\text{est}}(p_k), c_{m-1}^{\text{est}}(p_k)\in(C_M^{\text{est}}(\Delta_{j-1}^{p_k}),C_M^{\text{est}}(\Delta_j^{p_k})]\right)\cdot\Delta_j^{p_k}$$

$$\overset{(b)}{\le} \sum_{j\in[I(p_k)]} (C_M^{\text{est}}(\Delta_j^{p_k})-C_M^{\text{est}}(\Delta_{j-1}^{p_k}))\cdot\Delta_j^{p_k}.$$

Now, we can compute the regret incurred by selecting the meta arm which includes under-selected discretized arms:

$$L\cdot\sum_{p_k}\sum_{l\in[I(p_k)]} c_M^{\text{est},l,\text{und}}(p_k)\cdot\Delta_l^{p_k}$$

$$\le L\cdot\sum_{p_k}\sum_{j\in[I(p_k)]} (C_M^{\text{est}}(\Delta_j^{p_k})-C_M^{\text{est}}(\Delta_{j-1}^{p_k}))\cdot\Delta_j^{p_k}$$

$$= L\cdot\sum_{p_k}\left(C_M^{\text{est}}(\Delta_{\min}^{p_k})\Delta_{\min}^{p_k}+\sum_{j\in[I(p_k)-1]} C_M^{\text{est}}(\Delta_j^{p_k})\cdot(\Delta_j^{p_k}-\Delta_{j+1}^{p_k})\right)$$

$$\le L\cdot\sum_{p_k}\left(C_M^{\text{est}}(\Delta_{\min}^{p_k})\Delta_{\min}^{p_k}+\int_{\Delta_{\min}^{p_k}}^{\Delta_{\max}^{p_k}} C_M^{\text{est}}(x)dx\right)$$

$$= \sum_{p_k}\left(\frac{9K\ln(ML\rho)}{\rho(\Delta_{\min}^{p_k}/2-\texttt{err})^2}\cdot\Delta_{\min}^{p_k}+9K\ln(ML\rho)/\rho\cdot\int_{\Delta_{\min}^{p_k}}^{\Delta_{\max}^{p_k}}\frac{1}{(x/2-\texttt{err})^2}dx\right)$$

$$= \sum_{p_k}\left(\frac{9\Delta_{\min}^{p_k}K\ln(ML\rho)}{\rho(\Delta_{\min}^{p_k}/2-\texttt{err})^2}+\frac{9K\ln(ML\rho)}{\rho}\left(\frac{2}{\frac{\Delta_{\min}^{p_k}}{2}-\texttt{err}}-\frac{2}{\Delta_{\max}^{p_k}/2-\texttt{err}}\right)\right)$$

$$\le \sum_{p_k}\left(\frac{9K\ln(ML\rho)}{\rho}\left(\frac{\Delta_{\min}^{p_k}}{(\Delta_{\min}^{p_k}/2-\texttt{err})^2}+\frac{2}{\Delta_{\min}^{p_k}/2-\texttt{err}}\right)\right).$$

Combing the bound established in (24) will complete the proof. □

The instance-independent regret on discretized arm space is summarized in following lemma:

**Lemma E.6.** *Following the* `UCB` *designed in Algorithm* 3*, the instance-independent regret is given as*
$\text{Reg}_\epsilon(T)\le\mathcal{O}\left(K\cdot\sqrt{T\ln(T\rho)/(\rho\epsilon)}+K/(L\epsilon\rho^2)\right).$

*Proof.* Following the proof action-dependent bandits, we only need to consider the meta arms that are played when they are under-sampled. We particularly need to deal with the situation when $\Delta_{\min}^{p_k}$ is too small. We measure the threshold for $\Delta_{\min}^{p_k}$ based on $c_M^{\text{est}}(p_k)$, i.e., the counter of disretized

arm $p_k$ at phase horizon $M$. Let $\{M(p_k), \forall p_k\}$ be a set of possible counter values at time horizon $M$. Our analysis will then be conditioned on the event that $\mathcal{E}(p_k) := \{c_M^{\text{est}}(p_k) = M(p_k)\}$. By definition,

$$\mathbb{E}\big[\sum_{l \in [I(p_k)]} c_M^{\text{est},l,\text{und}}(p_k) \cdot \Delta_l^{p_k} \mid \mathcal{E}(p_k)\big]$$

$$= \sum_{m=1}^{M} \sum_{l \in [I(p_k)]} \mathbb{1}\big(\mathbf{p}(m) = \mathbf{p}_l, c_m^{\text{est}}(p_k) > c_{m-1}^{\text{est}}(p_k), c_{m-1}^{\text{est}}(p_k) \leq C_M^{\text{est}}(\Delta_l^{p_k}) \mid \mathcal{E}(p_k)\big) \cdot \Delta_l^{p_k}.$$

$$(25)$$

We define $\Delta^*(M(p_k)) := 2\left(\frac{9K\ln(ML\rho)}{L\rho \cdot M(p_k)}\right)^{1/2} + 2\text{err.}$ thus we have $C_M^{\text{est}}(\Delta^*(M(p_k))) = M(p_k)$.
To achieve *instance-independent* regret bound, we consider following two cases:
**Case 1:** $\Delta_{\min}^{p_k} > \Delta^*(M(p_k))$, clearly we have $\Delta_{\min}^{p_k}/2 > \text{err}$. Thus,

$$L \cdot \mathbb{E}\big[\sum_{l \in [I(p_k)]} c_M^{\text{est},l,\text{und}}(p_k) \cdot \Delta_l^{p_k} \mid \mathcal{E}(p_k)\big] \leq \mathcal{O}\left(\sqrt{\frac{K\ln(T\rho) \cdot LM(p_k)}{\rho}}\right). \qquad (26)$$

**Case 2:** $\Delta_{\min}^{p_k} < \Delta^*(M(p_k))$. Let $l^* := \min\{l \in I(p_k) : \Delta_l^{p_k} > \Delta^*(M(p_k))\}$. Observe that we have $\Delta_{l^*}^{p_k} \leq \Delta^*(M(p_k))$ and the counter $c^{\text{est}}(p_k)$ never go beyond $M(p_k)$, we thus have

$$L \cdot (25) \leq L(C_M^{\text{est}}(\Delta^*(M(p_k))) - C_M^{\text{est}}(\Delta_{l^*-1}^{p_k})) \cdot \Delta^*(M(p_k)) + \sum_{j \in [l^*-1]} L(C_M^{\text{est}}(\Delta_j^{p_k}) - C_M^{\text{est}}(\Delta_{j-1}^{p_k})) \cdot \Delta_j^{p_k}$$

$$\leq LC_M^{\text{est}}(\Delta^*(M(p_k))) \cdot \Delta^*(M(p_k)) + L \int_{\Delta^*(M(p_k))}^{\Delta_{\max}^{p_k}} C_M^{\text{est}}(x)dx$$

$$\leq \mathcal{O}\left(\sqrt{\frac{K\ln(T\rho) \cdot LM(p_k)}{\rho}}\right). \qquad (27)$$

Combining (26) and (27), and with Jesen's inequality and $\sum_{p_k} M(p_k) \leq KM/\epsilon$ will give us desired result. Put all pieces together, we have the instance-independent regret bound as stated in the lemma. The final inequality does not depend on the event $\mathcal{E}(p_k)$, we thus can drop this conditional expectation. $\qquad \square$

Combining with the discretization error, we have

$$\text{Reg}(T) \leq \mathcal{O}\left(K \cdot \sqrt{T\ln(T\rho)/(\rho\epsilon)} + K/(L\epsilon\rho^2)\right) + \mathcal{O}(K\epsilon T).$$

Picking

$$\epsilon = \mathcal{O}\left(\frac{\ln(T\rho)}{T\rho}\right)^{1/3}; \quad s_a = \mathcal{O}\left(\frac{1/3\ln\left(\frac{\ln(T\rho)}{T\rho}\right) - \ln(L^*K)}{\ln\gamma}\right).$$

We will obtain the results as stated in the theorem.

## F  Lower Bound of Action-Dependent Bandits

In this section, we derive the lower regret bound of bandits with action-dependent feedback, showing that the upper regret bound of our Algorithm 1 is optimal in the sense that it matches this lower bound in terms of the dependency on $T$ and $K$. Note that, by importance-weighting technique, we can construct an unbiased estimation of each base arm's reward. In the below discussion, we rephrase our problem as the *combinatorial Lipschitz bandit with constraint*, henceforth called `CombLipBwC`, which directly operates on the observations of all base arms:

**Definition F.1** (`CombLipBwC`). *Let action set $\mathcal{P}$ available to the learner be a continuous space, consisted of $K$ unit-range base arms, i.e., $\mathcal{P} \subset [0,1]^K$. At each time, the learner needs to select a meta arm $\mathbf{p}(t) = \{p_1(t), \ldots, p_K(t)\}$ in which each discretized arm $p_k(t) \in [0,1]$ is selected from $k$-th unit range, with the constraint such that $\sum_k p_k(t) = 1$. And then the learner will observe rewards $\{\tilde{r}_t(p_k(t))\}_{k \in [K]}$ for all base arms with the mean of each $\mathbb{E}[\tilde{r}_t(p_k(t))] = r_k(p_k(t))$.*

Our main result of this section is summarized in the following theorem:

**Theorem F.2.** *Let $T > 2K$ and $K \geq 4$, there exists a problem instance such that for any algorithm $\mathcal{A}$ for our action-dependent bandits , we have $\inf_{\mathcal{A}} \mathrm{Reg}(T) \geq \Omega(KT^{2/3})$.*

The high-level intuition for deriving the above lower bound is that we first construct a reduction from `CombLipBwC` to a discretized *combinatorial bandit problem with the action constraint* $\sum_k p_k(t) = 1$ - we refer to this latter problem setting as `CombBwC`. Then we show that the regret incurred within `CombLipBwC` is lower bounded by the regret incurred with `CombBwC`. To finish the proof, we bound the worst-case regret from below of `CombBwC` by taking an average over a conveniently chosen class of problem instances.

## F.1 Randomized problem instances and definitions

We now construct a reduction for proving the lower bound of `CombLipBwC`. Specifically, we will construct a distribution $\mathcal{D}$ over a set of problem instances (we also call each instance an adversary, since the instances are adversarially constructed) of `CombLipBwC`, while each problem instance will be uniquely mapped to a problem instance in `CombBwC`. The construction is similar to the one used in [39].

These new instances are associated with $0 - 1$ rewards. For each base arm $k \in [K]$, all the discretized arms $p$ have mean reward $r_k(p) = 1/2$ except those near the unique best discretized arm $p_k^*$ with $r_k(p_k^*) = 1/2 + \epsilon$. Here $\epsilon > 0$ is a parameter to be adjusted later in the analysis. Due to the requirement of Lipschitz condition, a smooth transition is needed in the neighborhood of each $p_k^*$. More formally, we define the following function $r_k(\cdot)$ for base arm $k$:

$$r_k(p) = \begin{cases} 1/2, & \forall p \in [0,1] : |p - p_k^*| \geq \epsilon/L_k \\ 1/2 + \epsilon - L_k \cdot |p - p_k^*|, & \forall p \in [0,1] : |p - p_k^*| < \epsilon/L_k \end{cases} \tag{28}$$

Fix $N_p \in \mathbb{N}$ and partition all base arms $[0,1]$ into $N_p$ disjoint intervals of length $1/N_p$. Then the above functions indicate that each interval with the length of $2\epsilon$ will either contain a bump or be completely flat. For the sake of simplifying presentation, in the analysis below, we'll focus on the case where the Lipschitz constant is $L_k = 1, \forall k \in [K]$. Formally,

**Definition F.3.** *We define 0-1 rewards problem instances $\mathcal{I}(\mathbf{p}^*, \epsilon)$ for `CombLipBwC` indexed by a random permutation $\mathbf{p}^* = \{p_k^*\}_{k \in [K]}$, which satisfies following property:*

- *$\sum_k p_k^* = 1$ and each $p_k^*$ takes the value from $\{(2j-1)\epsilon\}_{j \in [N_p]}$.*
- *The reward function of base arm $k$ is defined in (28), and the optimal action of arm $k$ is $p_k^*$.*

In combinatorial bandits, the learner selects a subset of ground arms subject to some pre-defined constraints. Adapting to our model, we denote this discretized action space $\mathcal{M}$ as the set of $K \times N_p$ binary matrices $\{0,1\}^{K \times N_p}$:

$$\mathcal{M} = \{\mathbf{a} \in \{0,1\}^{K \times N_p} : \forall k \in [K], \sum_{j=1}^{N_p} a_{k,j} = 1\},$$

where $a_{k,j} \in \{0,1\}$ is the indicator random variable such that $a_{k,j} = 1$ means that the $j$-th discretized arm probability is selected for the $k$-th base arm. Note that this space has not included the action constraint that we're planning to impose on `CombLipBwC`.

We now construct the problem instances for `CombBwC` such that each problem instance $\mathcal{I}(\mathbf{p}^*, \epsilon)$ in `CombLipBwC` has a corresponding problem instance in `CombBwC`.

**Definition F.4.** *We define 0-1 rewards problem instances $\mathcal{J}(\mathbf{l}^*, \epsilon)$ for `CombBwC` indexed by $\mathbf{l}^* = \{l_k^*\}_{k \in [K]}$, such that $l_k^* = (p_k^*/\epsilon + 1)/2$. Therefore, $l_k^* \in [N_p]$ and the mean reward of $\mathcal{J}(\mathbf{l}^*, \epsilon)$ is defined as follows: for any $t \in \{1, \ldots, T\}$,*

$$\mathbb{E}[\tilde{r}_t(l_k(t))] = \begin{cases} 1/2, & l_k(t) \neq l_k^* \\ 1/2 + \epsilon, & l_k(t) = l_k^* \end{cases} \tag{29}$$

Observe that with one more action constraint, the feasible action space of `CombBwC` will be a constrained space of $\mathcal{M}$, which we denote by $\Pi = \{\mathbf{a} \in \{0,1\}^{K \times N_p} : \forall k \in [K], \sum_{j=1}^{N_p} a_{k,j} = 1, \sum_{i=1}^{K} l_{k,j} a_{k,j} = K - 1 + N_p\}$.

We now next show that for any algorithm $\mathcal{A}_\mathcal{I}$ trying to solve the problem instance $\mathcal{I}(\mathbf{p}^*, \epsilon)$ in CombLipBwC, we can construct an algorithm $\mathcal{A}_\mathcal{J}$ that needs to solve a corresponding problem instance $\mathcal{J}(\mathbf{p}^*, \epsilon)$ in CombBwC.

The intuition of the construction routine is as follows. With the above defined $KN_\mathbf{p}$ intervals in hand and the deliberately designed reward structure, whenever an algorithm chooses a meta arm $\mathbf{p} = \{p_1, \ldots, p_K\}$ such that each discretized arm $p_k$ falls into an interval of this base arm $k$, choosing the center of this interval is best. Thus, if we restrict to discretized arms that are centers of the intervals of all base arms, we then have a family of problem instances of CombBwC, where the reward function is exactly defined in (29).

---

**Routine** A routine inbetween $\mathcal{A}_\mathcal{I}$ and $\mathcal{A}_\mathcal{J}$

**Input:** A CombLipBwC instance $\mathcal{I}$, a CombBwC instance $\mathcal{J}$ and an algorithm $\mathcal{A}_\mathcal{I}$ for solving $\mathcal{I}$.
**for** round $t = 1, \ldots$ **do**
    $\mathcal{A}_\mathcal{I}$ selects a meta arm $\mathbf{p}(t) = \{p_1(t), \ldots, p_K(t)\}$;
    $\mathcal{A}_\mathcal{J}$ selects arm $\mathbf{l}(t) = \{l_1(t), \ldots, l_K(t)\}$ such that $p_k(t) \in \big[(2l_k(t) - 1)\epsilon - \epsilon, (2l_k(t) - 1)\epsilon + \epsilon\big), \forall k \in [K]$;
    $\mathcal{A}_\mathcal{J}$ observes $\{\tilde{r}(l_k(t))\}$;
    $\mathcal{A}_\mathcal{I}$ observes $\{\tilde{r}(p_k(t))\}$;
**end for**

---

Furthermore, with above construction routine, we have following guarantee:

**Lemma F.5.** *The regret incurred by $\mathcal{A}_\mathcal{I}$, which is for the problem instance $\mathcal{I}(\mathbf{p}^*, \epsilon)$, is lower bounded by the regret incurred by $\mathcal{A}_\mathcal{J}$ for the problem instance $\mathcal{J}(\mathbf{l}^*, \epsilon)$:*

$$\mathbb{E}[\text{Reg}_{2\epsilon}(T)|\mathcal{I}, \mathcal{A}_\mathcal{I}] \geq \mathbb{E}[\text{Reg}_{2\epsilon}(T)|\mathcal{J}, \mathcal{A}_\mathcal{J}]. \tag{30}$$

*Proof.* As we can see, each instance $\mathcal{J}(\mathbf{l}^*, \epsilon)$ corresponds to an instance $\mathcal{I}(\mathbf{p}^*, \epsilon)$ of CombLipBwC. In particular, each $k$-th base arm in $\mathcal{J}$ corresponds to the base arm $k$ in $\mathcal{I}$, and more specifically, each discretized arm $j \in [N_\mathbf{p}]$ in $k$-th base arm corresponds to the all possible discretized arms $p$ such that $p \in [(2j - 1) \cdot \epsilon - \epsilon, (2j - 1) \cdot \epsilon + \epsilon)$. In other words, we can view $\mathcal{J}$ as a discrete version of $\mathcal{I}$. In particular, we have $r_k(j|\mathcal{J}) = r_k(p), \forall p \in [(2j - 1)\epsilon - \epsilon, (2j - 1)\epsilon + \epsilon)$, where $r_k(\cdot)$ is the reward function for base arm $k$ in $\mathcal{I}$, and $r_k(\cdot|\mathcal{J})$ is the reward function for base arm $k$ in $\mathcal{J}$.

Given an arbitrary algorithm $\mathcal{A}_\mathcal{I}$ for a problem instance $\mathcal{I}$ of CombLipBwC, we can use it to construct an algorithm $\mathcal{A}_\mathcal{J}$ to solve the corresponding problem instance $\mathcal{J}$ in CombBwC. To see this, at each round, $\mathcal{A}_\mathcal{I}$ is called and an action is selected $\mathbf{p}(t)$. This action corresponds to an action $\mathbf{l}(t)$ in CombBwC such that for each discretized arm $p_k(t) \in \mathbf{p}(t)$, it falls into the interval $[(2l_k(t) - 1)\epsilon - \epsilon, (2l_k(t) - 1)\epsilon + \epsilon)$ where $l_k(t) \in \mathbf{l}(t)$. Then algorithm $\mathcal{A}_\mathcal{J}$ will observe $\{\tilde{r}(l_k(t))\}$ and receive the reward $\sum_k \tilde{r}(l_k(t))$. After that, $\sum_k \tilde{r}(l_k(t))$ and $\mathbf{p}(t)$ will be further used to compute reward $\sum_k \tilde{r}(p_k(t))$ such that $\mathbb{E}[\sum_k \tilde{r}(p_k(t))] = \sum_{k \in [K]} r(l_k(t))$, and feed it back to $\mathcal{A}_\mathcal{I}$.

At each round, let $\mathbf{p}(t)$ and $\mathbf{l}(t)$ denote the action chosen by the $\mathcal{A}_\mathcal{I}$ and $\mathcal{A}_\mathcal{J}$, since we have $r_k(l_k(t)) \geq r_k(p_k(t))$ and best arm of the problem instance $\mathcal{I}$ and $\mathcal{J}$ has the same mean reward $K(1/2 + \epsilon)$, this completes the proof. $\qquad\square$

### F.2 Lower bound the $\mathbb{E}[\text{Reg}_{2\epsilon}(T)|\mathcal{J}, \mathcal{A}_\mathcal{J}]$

With Lemma F.5 stating the relationship between $\mathbb{E}[\text{Reg}_{2\epsilon}(T)|\mathcal{I}, \mathcal{A}_\mathcal{I}]$ and $\mathbb{E}[\text{Reg}_{2\epsilon}(T)|\mathcal{J}, \mathcal{A}_\mathcal{J}]$ as derived in (30), we can lower bound the $\mathbb{E}[\text{Reg}_{2\epsilon}(T)|\mathcal{A}_\mathcal{I}]$ via deriving the lower bound for $\mathbb{E}[\text{Reg}_{2\epsilon}(T)|\mathcal{A}_\mathcal{J}]$.

The structure of the proof is similar to that of [2], while the main difference is that we construct a different set of adversaries to bound the probability of the learner on achieving "good event" (will be specified later). At a high level, our proof builds on the following 4 steps: from step 1 to 3 we restrict our attention to the case of deterministic strategies for the learner, and then we show how to extend the results to arbitrary and randomized strategies by Fubini's theorem in step 4.

**Step 1: Regret Notions.** We will also call that the learner is playing against the $\mathbf{l}^*$-adversary when the current instance is $\mathcal{J}(\mathbf{l}^*, \epsilon)$. We denote by $\mathbb{E}_{\mathbf{l}^*}[\cdot]$ the expectation with respect to the reward generation process of the $\mathbf{l}^*$-adversary. Without the loss of generality, we assume $K$ is an even number. We write $\mathbb{P}_{(2h-1,2h),\mathbf{l}^*}$ for the probability distribution of $(j_{2h-1,t}, j_{2h,t})$ when the learner faces the $\mathbf{l}^*$-adversary. Thus, against the $\mathbf{l}^*$-adversary, we have

$$\mathbb{E}_{\mathbf{l}^*}[\mathrm{Reg}_{2\epsilon}(T)] = \mathbb{E}_{\mathbf{l}^*} \sum_{t=1}^{T} \sum_{h=1}^{K/2} 2\epsilon \mathbb{1}(\{j_{2h-1,t} \neq l_{2h-1}^*, j_{2h,t} \neq l_{2h}^*\}) = T \cdot 2\epsilon \sum_{h=1}^{K/2} \Big(1 - \mathbb{P}_{(2h-1,2h),\mathbf{l}^*}(G_T)\Big),$$

where $G_T$ denotes the *good event* such that $\{j_{2h-1,T} = l_{2h-1}^*, j_{2h,T} = l_{2h}^*\}$ holds simultaneously for base arm $2h-1$ and $2h$. For a particular distribution $\mathbf{l}^* \sim \mathcal{D}$ for all random adversaries, and let $\mathbb{P}(\mathbf{l}^*)$ denote the support of the adversary $\mathbf{l}^*$. Because the maximum value is always no less than the mean, we have

$$\sup_{\mathbf{l}^* \in \mathcal{J}_\epsilon} \mathbb{E}_{\mathbf{l}^*}[\mathrm{Reg}_{2\epsilon}(T)] \geq T \cdot 2\epsilon \sum_{h=1}^{K/2} \Big(1 - \sum_{\mathbf{l}^* \in \mathcal{J}_\epsilon} \mathbb{P}(\mathbf{l}^*) \cdot \mathbb{P}_{(2h-1,2h),\mathbf{l}^*}(G_T)\Big). \tag{31}$$

**Step 2: Information Inequality** Let $\mathbb{P}_{-(2h-1,2h),\mathbf{l}^*}$ be the probability distribution of $(j_{2h-1,t}, j_{2h,t})$ against the adversary which plays like the $\mathbf{l}^*$-adversary except that in the $(2h-1, 2h)$−th base arms, where the rewards of all discretized arms are drawn from a Bernoulli distribution of parameter $1/2$. We refer to it as $(-h, \mathbf{l}^*)$-adversary. Let $\mathcal{J}_\epsilon$ denote the set of all possible $\mathbf{l}^*$ adversaries and $\mathcal{D}$ be the distribution over $\mathbf{l}^*$ in which $\mathbf{l}^*$ is sampled uniformly at random.

**Lemma F.6.** *Let $n_{-h}(K - 1 + N_p - m), \forall m \in \{2, \ldots, 1 + N_p\}$ denote the total number of the combinations of $(j_k)_{k \neq 2h-1,2h}$ such that $\sum_{i \neq 2h-1,2h} j_k = K - 1 + N_p - m$. Then we have*

$$\frac{1}{|\mathcal{J}_\epsilon|} \sum_{\mathbf{l}^* \in \mathcal{J}_\epsilon} \mathbb{P}_{(2h-1,2h),\mathbf{l}^*}(G_T) \leq \sum_{m=2}^{N_p+1} \frac{n_{-h}(K-1+N_p-m)}{|\mathcal{J}_\epsilon|} + c\epsilon \sqrt{\frac{T}{|\mathcal{J}_\epsilon|} \sum_{m=2}^{N_p+1} n_{-h}(K-1+N_p-m)}, \tag{32}$$

*where $c$ is a constant.*

*Proof.* Let $\mathrm{KL}(\cdot)$ be the Kullback-Leibler divergence operator. By Pinsker's inequality, we have

$$\mathbb{P}_{(2h-1,2h),\mathbf{l}^*}(G_T) \leq \mathbb{P}_{-(2h-1,2h),\mathbf{l}^*}(G_T) + \sqrt{\frac{1}{2}\mathrm{KL}(\mathbb{P}_{-(2h-1,2h),\mathbf{l}^*}, \mathbb{P}_{(2h-1,2h),\mathbf{l}^*})}, \quad \forall \mathbf{l}^* \in \mathcal{J}_\epsilon.$$

Then by the concavity of the square root,

$$\frac{1}{|\mathcal{J}_\epsilon|} \sum_{\mathbf{l}^* \in \mathcal{J}_\epsilon} \mathbb{P}_{(2h-1,2h),\mathbf{l}^*}(G_T)$$

$$\leq \frac{1}{|\mathcal{J}_\epsilon|} \sum_{\mathbf{l}^* \in \mathcal{J}_\epsilon} \mathbb{P}_{-(2h-1,2h),\mathbf{l}^*}(G_T) + \sqrt{\frac{1}{2|\mathcal{J}_\epsilon|} \sum_{\mathbf{l}^* \in \mathcal{J}_\epsilon} \mathrm{KL}(\mathbb{P}_{-(2h-1,2h),\mathbf{l}^*}, \mathbb{P}_{(2h-1,2h),\mathbf{l}^*})}.$$

We introduce $n_h(m), \forall m \in \{2, \ldots, 1 + N_p\}$ to denote the total number of combinations of $(j_{2h-1}, j_{2h})$ such that $j_{2h-1} + j_{2h} = m$. Then by definition, it is easy to see that $n_h(m) = m - 1$, and furthermore

$$\sum_{m=2}^{N_p+1} n_h(m) \cdot n_{-h}(K - 1 + N_p - m) = |\mathcal{J}_\epsilon|. \tag{33}$$

Let $\mathcal{D}$ be the distribution over $\mathbf{l}^*$ in which $\mathbf{l}^*$ is sampled uniformly at random, i.e., $\mathbb{P}(\mathbf{l}^*) = \frac{1}{|\mathcal{J}_\epsilon|}$, then by the symmetry of the adversary $(-h, \mathbf{l}^*)$, we have

$$
\begin{aligned}
\sum_{\mathbf{l}^* \in \mathcal{J}_\epsilon} \mathbb{P}(\mathbf{l}^*) \cdot \mathbb{P}_{-(2h-1,2h),\mathbf{l}^*}(G_T) &= \sum_{m=2}^{N_{\mathrm{p}}+1} \sum_{\mathbf{l}^*: \sum_{k \neq 2h-1,2h} l_k^* = K-1+N_{\mathrm{p}}-m} \mathbb{P}(\mathbf{l}^*) \cdot \mathbb{P}_{-(2h-1,2h),\mathbf{l}^*}(G_T) \\
&= \sum_{m=2}^{N_{\mathrm{p}}+1} \frac{1}{n_h(m)} \sum_{\mathbf{l}^*: \sum_{k \neq 2h-1,2h} l_k^* = K-1+N_{\mathrm{p}}-m} \mathbb{P}(\mathbf{l}^*) \\
&= \sum_{m=2}^{N_{\mathrm{p}}+1} \frac{1}{n_h(m)} \frac{n_h(m) \cdot n_{-h}(K-1+N_{\mathrm{p}}-m)}{|\mathcal{J}_\epsilon|} \\
&= \sum_{m=2}^{N_{\mathrm{p}}+1} \frac{n_{-h}(K-1+N_{\mathrm{p}}-m)}{|\mathcal{J}_\epsilon|}. \qquad \text{By (33)}
\end{aligned}
$$

$\square$

**Step 3: Bounding** $\mathrm{KL}(\mathbb{P}_{-(2h-1,2h),\mathbf{l}^*}, \mathbb{P}_{(2h-1,2h),\mathbf{l}^*})$ **via the chain rule.** We now proceed to bound the value of $\mathrm{KL}(\mathbb{P}_{-(2h-1,2h),\mathbf{l}^*}, \mathbb{P}_{(2h-1,2h),\mathbf{l}^*})$.

**Lemma F.7.** $\mathrm{KL}(\mathbb{P}_{-(2h-1,2h),\mathbf{l}^*}^T, \mathbb{P}_{(2h-1,2h),\mathbf{l}^*}^T) \leq \frac{c\epsilon^2 T}{1-4\epsilon^2} \mathbb{P}_{-(2h-1,2h),\mathbf{l}^*}(G_T)$, *where $c$ is the constant value.*

*Proof.* Given any sequence of observed rewards up to time $T$, which denoted by $W_T \in \{1, \ldots, K\}^T$, the empirical distribution of plays, and, in particular, the probability distribution of $(j_{2h-1,t}, j_{2h,t})$ conditional on the fact that $W_T$ will be the same for all adversaries. Thus, if we denote by $\mathbb{P}_{(2h-1,2h),\mathbf{l}^*}^T$ (or $\mathbb{P}_{-(2h-1,2h),\mathbf{l}^*}^T$) the probability distribution of $W_T$ when the learner plays against the $\mathbf{l}^*$-adversary (or the $(-h, \mathbf{l}^*)$-adversary), we can easily show that $\mathrm{KL}(\mathbb{P}_{-(2h-1,2h),\mathbf{l}^*}, \mathbb{P}_{(2h-1,2h),\mathbf{l}^*}) \leq \mathrm{KL}(\mathbb{P}_{-(2h-1,2h),\mathbf{l}^*}^T, \mathbb{P}_{(2h-1,2h),\mathbf{l}^*}^T)$. Then we apply the chain rule for Kullback-Leibler divergence iteratively to introduce the probability distributions $\mathbb{P}_{(2h-1,2h),\mathbf{l}^*}^t$ of the observed rewards $W_t$ up to time $t$ and then will arrive desired result. More formally, we reduce to bound the $\mathrm{KL}(\mathbb{P}_{-(2h-1,2h),\mathbf{l}^*}^T, \mathbb{P}_{(2h-1,2h),\mathbf{l}^*}^T)$,

$$
\begin{aligned}
& \mathrm{KL}(\mathbb{P}_{-(2h-1,2h),\mathbf{l}^*}^T, \mathbb{P}_{(2h-1,2h),\mathbf{l}^*}^T) \\
={} & \mathrm{KL}(\mathbb{P}_{-(2h-1,2h),\mathbf{l}^*}^1, \mathbb{P}_{(2h-1,2h),\mathbf{l}^*}^1) + \\
& \sum_{t=2}^{T} \sum_{w_{t-1} \in \{1,\ldots,K\}^{t-1}} \mathbb{P}_{-(2h-1,2h),\mathbf{l}^*}^{t-1}(w_{t-1}) \mathrm{KL}\big(\mathbb{P}_{-(2h-1,2h),\mathbf{l}^*}(\cdot | w_{t-1}), \mathbb{P}_{(2h-1,2h),\mathbf{l}^*}(\cdot | w_{t-1})\big) \\
={} & \mathrm{KL}(\mathcal{B}_\emptyset, \mathcal{B}_\emptyset') \mathbb{1}(j_{2h-1,1} = l_{2h-1}^*, j_{2h,1} = l_{2h}^*) + \\
& \sum_{t=2}^{T} \sum_{w_{t-1}: j_{2h-1,t-1}=l_{2h-1}^*, j_{2h,t-1}=l_{2h}^*} \mathbb{P}_{-(2h-1,2h),\mathbf{l}^*}^{t-1}(w_{t-1}) \mathrm{KL}(\mathcal{B}_{w_{t-1}}, \mathcal{B}_{w_{t-1}}') \\
={} & \mathrm{KL}(\mathcal{B}_\emptyset, \mathcal{B}_\emptyset') \mathbb{1}(G_1) + \sum_{t=2}^{T} \sum_{w_{t-1}: G_{t-1}} \mathbb{P}_{-(2h-1,2h),\mathbf{l}^*}^{t-1}(w_{t-1}) \mathrm{KL}(\mathcal{B}_{w_{t-1}}, \mathcal{B}_{w_{t-1}}'),
\end{aligned}
$$

where $\mathcal{B}_{w_{t-1}}$ and $\mathcal{B}_{w_{t-1}}'$ are two Bernoulli random variables with parameters in $\{1/2, 1/2 + \epsilon\}$. Due to the fact that $\mathrm{KL}(p, q) \leq \frac{(p-q^2)}{q(1-q)}$, we will have

$$
\mathrm{KL}(\mathcal{B}_{w_{t-1}}, \mathcal{B}_{w_{t-1}}') \leq c \frac{\epsilon^2}{1-4\epsilon^2},
$$

where $c$ is a constant. Taking the summation will complete the proof. $\square$

**Wrapping up: Proof of Theorem F.2 on Deterministic Strategies.** Observe that we can bound

$$\sum_{m=2}^{N_{\mathrm{p}}+1} n_{-h}(K-1+N_{\mathrm{p}}-m)/|\mathcal{J}_\epsilon| = \Omega(1/N_{\mathrm{p}}),$$

which follows the fact that: given $a_1 \le a_2 \le \dots \le a_n$ and $b_1 \le b_2 \le \dots \le b_n$, one will have $n\sum a_i b_i \ge \sum a_i \sum b_i$. Plugging back into Eqs. (32) and (31) and substituting $\epsilon = \Theta(T^{-1/3})$ will get the desired result.

**Step 4: Fubini's theorem for Random Strategies.** For a randomized learner, let $\mathbb{E}_{\mathrm{rand}}$ denote the expectation with respect to the randomization of the learner. Then

$$\frac{1}{|\mathcal{J}_\epsilon|}\sum_{\mathbf{l}^*\in\mathcal{J}_\epsilon}\mathbb{E}\sum_{t=1}^{T}\left(\mathbf{l}(t)^T\mathbf{r}_t - (\mathbf{l}^*)^T\mathbf{r}_t\right) = \mathbb{E}_{\mathrm{rand}}\frac{1}{|\mathcal{J}_\epsilon|}\sum_{\mathbf{l}^*\in\mathcal{J}_\epsilon}\mathbb{E}_{\mathbf{l}^*}\sum_{t=1}^{T}\left(\mathbf{l}(t)^T\mathbf{r}_t - (\mathbf{l}^*)^T\mathbf{r}_t\right).$$

where $\mathbf{r}_t = (r_1(l_1(t)),\dots,r_K(l_K(t)))$, and value of the reward for not realized arms are computed from Eq (5). The interchange of the integration and the expectation is justified by Fubini's Theorem. For every realization of learner's randomization, the results of all earlier steps still follow. This will give us the same lower bound for $\mathbb{E}_{\mathrm{rand}}\frac{1}{|\mathcal{J}_\epsilon|}\sum_{\mathbf{l}^*\in\mathcal{J}_\epsilon}\mathbb{E}_{\mathbf{l}^*}\sum_{t=1}^{T}\left(\mathbf{l}(t)^T\mathbf{r}_t - (\mathbf{l}^*)^T\mathbf{r}_t\right)$ as we have shown above.

## G   Proof of the lower bound in history-dependent bandits

For history-dependent bandits, we show that for a general class of utility function which satisfies the *strictly proper* property (we will shortly elaborate this property), solving history-dependent bandits is as least hard as solving action-dependent bandits. Armed with the above derived lower bound of action-dependent case, we can then conclude the lower bound of history-dependent case. *Strictly Proper Utility Function* is defined as below.

**Definition G.1** (Strictly Proper Utility). *For any mixed strategy $\mathbf{p} \in \mathcal{P}$ and any $\mathbf{q} \neq \mathbf{p}$, the functions $\{r_k\}$ are strictly proper if following holds,*

$$\sum_{p_k\in\mathbf{p}} p_k r_k(p_k) > \sum_{p_k\in\mathbf{p},q_k\in\mathbf{q}} p_k r_k(q_k). \tag{34}$$

With above defined strictly proper utility at hand, we now ready to prove the Theorem 6.1 for history-dependent case.

*Proof.* Let $\mathcal{I}^h$ denote a history-dependent bandits instance whose utility function satisfies above defined strictly proper property, and $\mathcal{I}^a$ denote the associated action-dependent bandit instance whose utility function is the same as that in $\mathcal{I}^h$. Let $\mathbf{f}^*(t) = \{f_k^*\}_{k\in[K]}$ be the discounted frequency at time $t$ when the learner keeps deploying the best-in-hindsight strategy $\mathbf{p}^*$ and $L^* = \max L_k$. Then we can show that

$$\mathbb{E}[\mathrm{Reg}(T)|\mathcal{I}^h] = \sum_{t=1}^{T}U_t(\mathbf{p}^*) - \sum_{t=1}^{T}U_t(\mathbf{p}(t))$$

$$= \sum_{t=1}^{T}\sum_{p_k^*\in\mathbf{p}^*} p_k^* \cdot r_k(f_k^*(t)) - \sum_{t=1}^{T}\sum_{k} p_k(t)\cdot r_k(f_k(t))$$

$$> \sum_{t=1}^{T}\sum_{p_k^*\in\mathbf{p}^*} p_k^* \cdot r_k(f_k^*(t)) - \sum_{t=1}^{T}\sum_{k} p_k(t)\cdot r_k(p_k(t))$$

$$\ge \sum_{t=1}^{T}\sum_{p_k^*\in\mathbf{p}^*} p_k^* \cdot r_k(p_k^*) - \frac{\gamma^2(1-\gamma^{2T-2})KL^*}{1-\gamma^2} - \sum_{t=1}^{T}\sum_{k} p_k(t)\cdot r_k(p_k(t))$$

$$= \mathbb{E}[\mathrm{Reg}(T)|\mathcal{I}^a] - \frac{\gamma^2(1-\gamma^{2T-2})KL^*}{1-\gamma^2} = \Omega(KT^{2/3}),$$

where the first inequality is due to the strict proper property of utility function, and the third inequality is due to the fact that the history-dependent bandits shares the same best-in-hindsight strategy as that in the action-dependent bandit and Lemma E.1. By the regret reduction from the history-dependent bandits to the action-dependent bandit, we can conclude the lower bound of the history-dependent case. □

# H    Optimal dynamic policy v.s. best policy in hindsight

As we mentioned, for action-dependent bandits, the optimal dynamic policy can be characterized by a best-in-hindsight (mixed) strategy computing from following constrained optimization problem: $\max_{\mathbf{p} \in \mathcal{P}} \sum_{k=1}^{K} p_k r_k(p_k)$. While for history-dependent bandits, it is possible that the optimal policy $\mathbf{p}^*$ may not be well-defined due to the fact of reward dependence on action history. However, we argue that when competing against with best-in-hindsight policy, notwithstanding in the face of this kind of reward-history correlation, the value of the optimal strategy is always well-defined in the limit, and this limit value is also characterized by the best-in-hindsight (mixed) strategy computed from action-dependent bandits. To gain intuition, note that the time-discounted frequency $\mathbf{f}(t)$ will be exponentially approach to the fixed strategy $\mathbf{p}$ the learner deploys. As we explain in Section 5, after consistently deploying $\mathbf{p}$ with $s$ rounds, the frequency $\mathbf{f}(t+s)$ will be converging to $\mathbf{p}$ with the exponential decay error $\gamma^s$. Thus, to achieve highest expected reward, the learner should deploy the optimal strategy computed as in action-dependent case.

# I    EVALUATIONS

We conducted a series of simulations to empirically evaluate the performance of our proposed solution with a set of baselines.

## I.1    Evaluations for action-dependent bandits

We first evaluate our proposed algorithm on action-dependent bandits against the following state-of-the-art bandit algorithms.

- **EXP3**: One natural baseline is applying EXP3 [4] on the space of base arms. While EXP3 is designed for adversarial rewards, it is competing with the best fixed arm in hindsight and might not work well in our setting since the optimal strategy is randomized.

- **EXP3-Meta-Arm (mEXP3)**: To make a potentially more fair comparison, we also implement EXP3 on the meta-arm space. We denote it as mEXP3 in the following discussion.

- **CUCB** [13, 65]: This algorithm is designed to solve combinatorial semi-bandit problem, which chooses $m$ arms out of $M$ arms at each round and receives only the rewards of selected arms. Mapping to our setting, $M = K/\epsilon$ represents the total number of discretized arms, $m = K$ is the number of base arms, and the selection of $m$ arms is constrained to satisfy the probability simplex constraint.

In the simulations, we set $K = 2$ for simplicity. Moreover, $r_k(p_k)$ is chosen such that $r_k(p_k)$ is maximized when $0 < p_k < 1$. In particular, we define $r_k(p_k)$ as a scaled Gaussian function : $r_k(p_k) = f(p_k|\tau_k, 0.5)/C_k$, where $f(x|\tau, \sigma^2)$ is the pdf of Gaussian distribution with the mean $\tau$ and variance $\sigma^2$, and $C_k = f(\tau_k|\tau_k, \sigma^2)$ is a constant ensuring $r_k(p_k) \in [0, 1], \forall p_k \in [0, 1]$. For each arm $k$, $\tau_k$ is uniformly draw from 0.45 to 0.55 and the instantaneous reward is drawn from a Bernoulli distribution with the mean of $r_k(p_k(t))$, i.e., $\tilde{r}_t(p_k(t)) \sim \texttt{Bernoulli}(r_k(p_k(t)))$. And the ratio $\rho$ is set to 0.2. For each algorithm we perform 40 runs for each of independent 40 values of the corresponding parameter, and we report the averaged results of these independent runs, where the error bars correspond to $\pm 2$ standard deviations.

The results, shown in Figure 2(a), demonstrate that our algorithm significantly outperforms the baselines. As expected, mEXP3 works better than EXP3 algorithm when $T$ is large, since the former searches the optimal strategies in the meta arm space. Our algorithm outperforms mEXP3 and CUCB since we utilize the problem structure, which reduces the amount of explorations.

## I.2 Evaluations for history-dependent bandits

We now evaluate our proposed algorithm for history-dependent bandits via comparing against the following baselines from non-stationary bandits. Note that, while CUCB performs reasonably well in action-dependent case, it does not apply in history-dependent case, since we cannot *select* the time-discounted frequency (which maps to the arm in CUCB) as required in CUCB.

- **Discounted UCB** [25, 40]: Discounted UCB (DUCB) is an adaptation of the standard UCB policies that relies on a discount factor $\gamma_{\text{DUCB}} \in (0,1)$. This method constructs an UCB : $\bar{r}_t(k, \gamma_{\text{DUCB}}) + c_t(k, \gamma_{\text{DUCB}})$ for the instantaneous expected reward, where the confidence is defined as $c_t(k, \gamma_{\text{DUCB}}) = 2\sqrt{\frac{\xi \ln(n_t)}{N_t(k, \gamma_{\text{DUCB}})}}$, for an appropriate parameter $\xi$, $N_t(k, \gamma_{\text{DUCB}}) = \sum_{s=1}^{t} \gamma_{\text{DUCB}}^{t-s} \mathbb{1}_{(a_s=k)}$, and the discounted empirical average is given by $\bar{r}_t(k, \gamma_{\text{DUCB}}) = \frac{1}{N_t(k, \gamma_{\text{DUCB}})} \cdot \sum_{s=1}^{t} \gamma_{\text{DUCB}}^{t-s} \tilde{r}_s(k) \mathbb{1}_{(a_s=k)}$.

- **Sliding-Window UCB** [25]: Sliding-Window UCB (SWUCB) is a modification of DUCB, instead of averaging the rewards over all past with a discount factor, SWUCB relies on a local empirical average of the observed rewards, for example, using only the $\tau$ last plays. Specifically, this method also constructs an UCB : $\bar{r}_t(k, \tau) + c_t(k, \tau)$ for the instantaneous expected reward. The local empirical average is given by $\bar{r}_t(k, \tau) = \frac{1}{N_t(k, \tau)} \sum_{s=t-\tau+1}^{t} \tilde{r}_s(k) \mathbb{1}_{(a_s=k)}$, $N_t(k, \tau) = \sum_{s=t-\tau+1}^{t} \gamma_{\text{DUCB}}^{t-s} \mathbb{1}_{(a_s=k)}$ and the confidence interval is defined as $c_t(k, \gamma_{\text{DUCB}}) = 2\sqrt{\frac{\xi \ln(\min(t, \tau))}{N_t(k, \tau)}}$.

We use *grid searches* to determine the algorithms' parameters. For example, in DUCB, the discount factor was chosen from $\gamma_{\text{DUCB}} \in \{0.5, 0.6, \ldots 0.9\}$, while the window size of SWUCB was chosen from $\tau \in \{10^2, \ldots, 5 \times 10^2\}$. Besides above algorithms, we also implement the celebrated non-stationary bandit algorithm EXP3.

We chose $K$ and $r_k(p_k)$ to be the same as the experiments in action-dependent case. And the discount factor is chosen as $\gamma_{\text{DUCB}} = 0.8$ and the window size for SWUCB is chosen as 200 via the grid search, and $\xi$ is set to 1. We examine the algorithm performances under different $\gamma$ (the parameter in time-discounted frequency), with smaller $\gamma$ indicating that arm rewards are more influenced by recent actions. As seen in Figures 2(b)-2(d), our algorithm outperforms all baselines in all $\gamma$ but the improvement is more significant with small $\gamma$. This is possibly due to that most non-stationary bandit algorithms have been focusing on settings in which the change of arm rewards over time is not dramatic.

We also examine our algorithm with larger number of base arms $K$ with comparing to above baseline algorithms and the performance of our algorithm on different ratios $\rho$. The results are presented in Figure 3 and show that our algorithm consistently performs better than other baselines when $K$ goes large. The results also suggest that our algorithm is not sensitive to different $\rho$, though one could see the regret is slightly lower when $\rho$ is increasing, which is expected from our regret bound.