# OpenReview forum: "Bandit Learning with Delayed Impact of Actions"
_NeurIPS.cc/2021/Conference — NeurIPS 2021 Poster_

### Official Review · Reviewer_ZiMd · 2021-07-13

**Rating:** 7
**Confidence:** 4

**Summary:**

The paper considers a bandit problem where actions have a discounted impact on future rewards. The authors propose an algorithm that achieves an $O(KT^\frac{2}{3})$ regret bound, assuming that the reward function is Lipschitz and resorting to a discretization of the action space and a waiting time that enables the rewards to stabilize (the last actions are the one that will impact the rewards the most). They provide a matching lower bound.

**Limitations And Societal Impact:**

The way the authors discuss the limitations of their work is satisfactory to me.

**Main Review:**

## Originality
Delayed rewards and rested bandits have already been studied, but the originality of this paper lies in the fact that the actions chosen at a certain time will impact their future rewards decreasingly (their rewards can decrease even at a time step when they are not pulled).
One of my concerns however is that it is the probability of choosing an arm that impacts its evolution. I do not understand in which practical case a probability of choosing an arm is key, rather than its effective choice. I would like the authors to clarify this point.
Also, I wonder what would happen, should the impact function structure depend on the arm (for example if the discount factor depended on the chosen arm)? This would maybe be a more realistic setting.

## Quality
The upper bound of order of $T^{2/3}$ seems satisfactory and is supported by a matching lower bound.
I am a little uncertain about the use of discretization. How should $\epsilon$ be chosen? Is $\epsilon$ a function of the Lipschitz constant? If so, this is an important limitation that should be discussed.


## Clarity
Some parts could be explained in more detail. The intuition behind the lower bound could be explained a little bit in the main paper for example, since it is a little expeditious as is.
I think the paper would really benefit from a discussion of the choice of the length of the approaching stage. How is the learner supposed to set $\rho$? How does it affect $L$?

## Typos
- I think $\delta$ should be $\gamma$, l118 and l172.
- "our model has its limitation" l350

## Edit
I thank the authors for taking the time to write clear answers. The only drawback of the paper that I see is that the fact that probabilities of actions impact their evolution is justified by very specific applications. Let me be clear, these use cases make sense to me but I think they should really be justified more in detail in the final version.

**Time Spent Reviewing:**

5

---

> ### Author Response · Authors · 2021-08-10
> **Response to Reviewer ZiMd**
>
> We sincerely thank the reviewer for the comments and feedback.
>
> ----------
> - **Motivation on the impact evolution**: We would like to first note that while we focus on the analysis of probabilistic arm pulling, our results apply to the natural case of proportional arm pulling (Remarks 2.1 and 4.1). In addition, there are applications that can motivate the setup when it is the probability of choosing an arm that impacts its evolution. Consider the classical security game. At each time, the learner (security department) needs to decide and “announce” her strategy $\mathbf{p}$ (the probability of allocating patrol resources over a set of locations). The attacker/terrorist then adjusts his strategy by responding to $\mathbf{p}$. The impact function naturally depends on the announced $\mathbf{p}$  in this case. Another example: consider each arm being a population of agents. The selection probability of this arm corresponds to some normalized fraction (w.r.t the entire population contributed by all arms) of agents that are offered resources. The impact of arms’ status will depend on the selection probability (i.e., the ratio of agents being selected in the population).
>
> - **Different impact functions for different base arms**: This is a good point. We agree with the reviewer that it is indeed more practical to consider the setting such that each base arm is associated with a different impact function.  For the example mentioned by the reviewer, we believe our results can be generalized to the setting where each base arm has a different discount factor, as what matters in the analysis is that we need to bound the approximation ratio between expected rewards after keeping selecting a same meta arm and the true expected rewards of this meta arm (please see Lemma E.1 in Appendix E). Informally, given a meta arm $\mathbf{p}$ , let $\gamma_\min = \min_{p_k\in\mathbf{p}} \gamma_k$ be the smallest discounted factor of the base arm in $\mathbf{p}$, then the result can be adapted to include $\gamma_\min$.  Generalizing our results to general heterogeneous impact functions would be an interesting future work.
>
> - **$\epsilon$ and its dependency on Lipschitz constant**: The choice of $\epsilon$ needs to balance the discretization error and the number of discretized arms. Larger $\epsilon$ leads to a smaller number of arms but larger discretization error. In our work, $\epsilon$ is optimized to be in the order of $(\ln T)/ T^1/3$. And yes, it is indeed a function of the Lipschitz constant $L$. Similar to traditional Lipschitz bandits (see Theorem 4.1 in [1]), the value of \epsilon is in the order of $L^{1/3}$. Since we are particularly interested in the algorithm’s performance over the time horizon $T$, we omit the Lipschitz constant with other numeric constants in the results presentation. We will clarify this in the revision.
>
> - **The intuition behind the lower bound, how to set rho and how does it affect phase length**:  We sincerely thank the reviewer's suggestions. We will add more details about the proof of lower bound in the main text (due to the space limit, we have to move it to the appendix in the current draft). To give some intuitions here. Informally, our setting is essentially a combinatorial Lipschitz bandit. So we can follow the classic technique “needle in a haystack” to construct hard problem instances. The main technical difficulty in our proof is that our combinatorial structure of the action space differs from the classical one (ours is a probability simplex) and needs additional care. For history-dependent bandits, we show that solving the history-dependent bandits is generally at least as hard as solving action-dependent bandits, utilizing a general class of utility function which satisfies the “strictly proper property”. Together with the lower bound of the action-dependent case, we can conclude the lower bound of the history-dependent case. \
> \
> Recall that $\rho$ denotes the fraction of estimation stage. Conditional on the approximation stage is “long enough” that the impact has converged, we would prefer a larger $\rho$ since it leads to lower regret, as demonstrated in our regret bound and empirical results. Note that this doesn’t mean the length of the approaching stage is negligible - it is instead because the regret incurred from the approaching stage is on a smaller order when T is large, and therefore does not appear in the final bound. This is also because the approaching stage converges fast: note that since our impact function is in the form of time-discounted average, when the same meta-arm is pulled repeatedly, the impact function converges exponentially fast in the number of repeated pulls in the approaching stage. Therefore, when T is large, any positive constant $\rho<1$ (as required in Theorem 5.1) would lead to a long enough approximation stage. When T is small, we need to more carefully choose $\rho$ to balance the convergence of impacts and estimation of rewards. For the effect to $L$, as discussed in Remark 5.2, we calculate $L$ using the length of approaching stage as specified by the ratio $\rho$. When $\rho$ increases, $L$ decreases.
>
> - **Typos**: Thanks for catching these. We will fix the typos in the revision. \
> \
> [1]. Slivkins, Aleksandrs. "Introduction to Multi-Armed Bandits." Foundations and Trends® in Machine Learning 12.1-2 (2019): 1-286.
> ----------
>
> Please let us know if you have any further concerns about the paper. We are happy to answer more questions.

---

### Official Review · Reviewer_L6w5 · 2021-07-16

**Rating:** 6
**Confidence:** 3

**Summary:**

This paper generalizes the stochastic bandit model by redefining the action space as a sequence of resource allocations, with a time discounted action-dependent bias on future resource allocations. The authors provide an example application based on their model, and propose an algorithm with proved near optimal regret upper bound. A lower bound based on the proposed model is provided.

**Limitations And Societal Impact:**

I think the authors have adequately addressed parts of limitation and potential negative societal impact of their work. There could be some other limitations, such as a more general format of function f(), but that could also be a future work. My suggestion would be that the authors could consider adding more insights of why the current function f() look like that, and how to generalize it, so as to cover more applications.
Another tiny concern is that, in Line 118 & 172, I think it should be \gamma instead of \delta.

**Main Review:**

The paper is well written and easy to read, with many details fully explained. I also think that this idea is novel especially the time discounted action impact. Existing works focus more on long-term impact by reward history, for example current action is influenced by a feedback function on reward history, such as https://arxiv.org/pdf/1802.05693.pdf, while this paper considers long-term influence by an impact function on action history, which differs from previous works by new non-trivial problem formulation and policy design.
I have one question: since the value of \epsilon and s_a affect the regret, and they are required as inputs, I am wondering how to tune these values to make the regret under control, or if only we make sure that \epsilon and s_a are in the orders as stated in Theorem 5.1, the we can guarantee a good regret regradless of the coefficients before the order? How can I get some clue when tuning the parameters?

**Time Spent Reviewing:**

3h

---

> ### Author Response · Authors · 2021-08-10
> **Response to Reviewer L6w5**
>
> We sincerely thank the reviewer for the comments and feedback. We appreciate that you find our paper well written and the setting in this paper is novel. We also thank the reviewer for bringing the linked paper to our attention.
>
> ----------
> - **The choices of $\epsilon$ and $s_a$**: The choice of $\epsilon$ is in the order of $O(\ln T/ T^{1/3})$. Once we determine the value of $\epsilon$, then we can determine the value of $s_a$. The coefficients omitted here include Lipschitz constant (where we follow the traditional Lipschitz bandit literature and assuming Lipschitz coefficient is a constant number) and some numeric constants. As we are particularly interested in the algorithm’s performance over the time horizon $T$, the effects of these constants are diminished when we increase the length of time horizon. We will add more clarifications.
>
> - **General impact functions**: Thanks for the suggestions! We will add more discussions      about the current impact function and generalizations. To provide a few immediate generalizations here: Our results hold for “sliding-window” impact function that depends on the actions of past $m$ steps for some constant $m$, e.g., $f_k(t) = \frac{\sum_{i = t-m}^t p_k(i)}{m}$ $f_k(t) = \frac{\sum_{i = t-m}^t p_k^2(i)}{m}$ are both valid impact functions. Applying some transformations to valid impact functions, such $f_k(t) = h_k\left(\frac{\sum_{i=1}^t \gamma^{t-i} p_k(i)}{\sum_{i=1}^t \gamma^{t-i}}\right)$ for any bounded Lipschitz function $h_k$, also lead to valid impact functions.
>
> - **Typo in Line 118 & 172**: Thanks for catching this. We will fix this in the revision.
> ----------
>
> Please let us know if you have any further concerns about the paper. We are happy to answer more questions.

---

### Official Review · Reviewer_aHiG · 2021-07-17

**Rating:** 7
**Confidence:** 4

**Summary:**

In this paper, the authors formulate and study a multi-armed bandit problem with delayed impact
of actions, in which the actions are taken in the past affect the rewards in the future. Under this
setting, the authors design an algorithm that utilizes the dynamics introduced by the delayed
impacts of past actions to look for the optimal meta arm that maximizes the collected utilities
over time. The algorithm is shown to incur regret of at most $\Tilde{O}(KT^{2/3})$, which
matches the theoretical lower bound for this problem.

**Ethics Review Area:**

["I don’t know"]

**Limitations And Societal Impact:**

Weaknesses:
One limitation of the current paper is that most of the analysis on action-dependent and
history-dependent bandits is done using the impact function in Equation (2). Although the
authors remark in Section 5.2 that similar analysis can be applied to impact functions that
converge when one keeps selecting the same meta arm, it is not clear how one should select
the monotone function $g(.)$ for a general impact function. Since this restriction does not really
apply to general functions, it might be better to state it as a formal assumption.
In the regret definition, the authors choose the optimal fixed policy instead of the stronger
optimal dynamic policy as the benchmark. In Appendix H, the authors justify their choice by
arguing that for history-dependent bandits, the optimal strategy in the limit is well defined and
can be characterized as the best-in-hindsight policy. From the intuitive explanations, it is unclear
to me why this holds and might need a more rigorous treatment.

Minor comments:
- On pages 3 and 4, “$\delta = 0$” is a typo and should be $\gamma = 0$.
- In Section 4, it says that $p_k \in \{\epsilon, 2\epsilon, \dots, 1\}$. Can $p_k$ take the
value of 0 as well?
- It might be good to compare the performance of the algorithm of different $rho$ values
against the benchmark algorithms. From Figure 3d alone, it is difficult to tell that the
algorithm is not sensitive to different values of $\rho$.

**Main Review:**

Strengths:
The paper is well written and structured. The model is novel in the current bandit's literature with
delayed impacts due to its consideration of long-term impacts. The paper provides good intuition
as to the applications of MABs with long-delayed impacts (the example in Section 2 well
illustrates the model set-up and the roles of the impact function). The theoretical results for both
action-dependent and history-dependent bandits are well explained. In particular, Algorithm 2/3
uses a simple idea that uses a short approaching stage to achieve the convergence of $f(t)$,
a longer estimation stage to estimate the reward of meta arm $p$, and a UCB-type algorithm to
find a good meta arm. The regret upper bound also matches the theoretical lower bound, which
demonstrates the optimality of the proposed algorithm.

Questions:
- How would one determine the choice of $\rho$ in Algorithm 2? Empirically, the authors
argue that the choice of $\rho$ does not influence the cumulative regret too much. But if
$\rho$ gets too small it seems that the theoretical upper bound worsens. Is there
anything that prevents us from always selecting a large $\rho$?

**Time Spent Reviewing:**

3 hours

---

> ### Author Response · Authors · 2021-08-10
> **Response to Reviewer aHiG**
>
> We sincerely thank the reviewer for the comments and feedback. We also appreciate that you find our paper novel and well written.
>
> ----------
> - **The choice of $\rho$**: Recall that $\rho$ denotes the fraction of estimation stage. Conditional on the approximation stage is “long enough” that the impact has converged, we would prefer a larger $\rho$ since it leads to lower regret, as demonstrated in our regret bound and empirical results. Note that this doesn’t mean the length of the approaching stage is negligible - it is instead because the regret incurred from the approaching stage is on a smaller order when T is large, and therefore does not appear in the final bound. This is also because the approaching stage converges fast: note that since our impact function is in the form of time-discounted average, when the same meta-arm is pulled repeatedly, the impact function converges exponentially fast in the number of repeated pulls in the approaching stage. Therefore, when T is large, any positive constant $\rho<1$ (as required in Theorem 5.1) would lead to a long enough approximation stage. When T is small, we need to more carefully choose $\rho$ to balance the convergence of impacts and estimation of rewards.
>
> - **General impact functions**: We will provide more formal discussion. Here we give a few immediate generalizations of our impact function: Our results hold for “sliding-window” impact function that depends on the actions of past $m$ steps for some constant $m$, e.g., $f_k(t) = \frac{\sum_{i = t-m}^t p_k(i)}{m}$ $f_k(t) = \frac{\sum_{i = t-m}^t p_k^2(i)}{m}$ are both valid impact functions. Applying some transformations to valid impact functions, such $f_k(t) = h_k\left(\frac{\sum_{i=1}^t \gamma^{t-i} p_k(i)}{\sum_{i=1}^t \gamma^{t-i}}\right)$ for any bounded Lipschitz function $h_k$, also lead to valid impact functions.
>
> - **Optimal fixed policy**: We agree with the reviewer and acknowledge that it is more reasonable to define the regret to be competing with the optimal dynamic policy.  However, to our best knowledge, calculating the optimal dynamic policy in our setting is challenging and nontrivial as it requires to solve an MDP with continuous states. Moreover, under some conditions (Definition G.1), the optimal dynamic policy is the same as best fixed policy in hindsight. We think considering a stronger regret definition would be an interesting and important future work. The argument in Appendix H relies on techniques in the literature of proper scoring rules (introduced in Appendix G). We will make the arguments more formal and rigorous.
>
> - **Minor comments**: It is indeed a typo. Thanks for catching it, and we will fix this in the revision. In our algorithm, to ensure the importance-weighted rewards are bounded, we make $p_k$ to start from $\epsilon$. We thank the reviewer for the helpful suggestion on our presentation of Fig. 3d, we will take the reviewer’s suggestion and update accordingly.
> ----------
>
> Please let us know if you have any further concerns about the paper. We are happy to answer more questions.

---

### Official Review · Reviewer_dTZk · 2021-07-22

**Rating:** 7
**Confidence:** 4

**Summary:**

The authors formulate and solve the problem of resource allocation to K groups under the bandit setting where the impact of actions perseveres over time and affects the corresponding reward function. They key idea is to realize that deploying the policy for a certain period of time helps us estimate its utility and therefore mixing this with updating the policy helps us to learn the optimal one.  A regret bound of Otilde(KT^2/3) is shown with a matching lower bound. Experiments are shown on synthetic datasets comparing it with EXP3, DUCB and SWUCB to show the performance improvement in terms of cumulative regret.

**Limitations And Societal Impact:**

Yes

**Main Review:**

Overall, the paper is well-written and solves an important problem for the community.  In particular, it is concerned with resource allocation in delayed settings which are both challenging as well as critically important for healthcare, crime and financial domains.

Pros:

(A) The problem is well-motivated by a crime example of deploying police officers to districts to reduce the rate of incidents. Also, applying standard algorithms is shown to lead to linear regret. These are typical in most of these settings but it is good to be formalized in Lemma 3.1
(B) Similar to EXP3, we see the importance weighting being utilized to redefine the reward function of the observed arms so that the expected reward is maintained.
(C) First the case where the impact functions are limited to the current selection are tackled and this is similar to UCB with the reweighted weights from above. The observation that history-dependent can then solved by utilizing the previous freeze and gather evidence and update the policy cycle is really nice and leads to an optimal regret. This is both theoretically and experimentally validated.

Cons/Questions:

(1) The setting was a bit unclear to me from the delay perspective. So, the reward of say loan applications to a certain group will result in a certain reward but usually there is a delay in observing such rewards. Is it the case that we need both to have a better model? We have impact of the previous decisions affecting the current rewards and also the current rewards and not observed till some future date?
(2) The issue with importance weighting, as we know, is that it can lead to high variance. How does your approach tackle the case where the policy has low probability for some of the arms? Could the usual tricks of having a biased estimator but lower variance help with this as we see with EXP3 -> EXP3-IX?
(3) The regret is compared to a static policy and it would have been nicer if the regret performance was over the optimal dynamic policy.

After rebuttal:

Thanks for the responses. I will keep my score unchanged as it clarifies the questions I had and is aligned with the other reviews as well.


**Time Spent Reviewing:**

5

---

> ### Author Response · Authors · 2021-08-10
> **Response to Reviewer dTZk**
>
> We sincerely thank the reviewer for the comments and feedback.  We also appreciate that you find our paper solving a both challenging and important problem.
>
> ----------
> - **Delayed action impacts and delayed rewards**: This is a good point. We agree that, in many practical scenarios, the rewards might not be immediately observable. There have been prior works [1,2] that have explored the problem that the reward realization might be delayed. In this paper, we focus on studying scenarios where historically made actions would affect the underlying arm rewards. It is definitely interesting to incorporate the two perspectives in the future work. We thank the reviewer for the suggestion, and we will add more discussions on the two settings in the revision.
>
> - **Importance weighting**: This is a great point. Indeed, applying importance weighting increases the variance for the reward estimation of base arm selected with small probability (more formally, the *discretized arm* as defined in L213). However, since our regret is defined with respect to the reward of the meta-arm (more specifically, the expected reward for each randomized selection strategy), the contribution of the high-variance small-probability base arms is multiplied, and therefore discounted proportionally, with the same small probability (the likelihood of obtaining reward from the base arm). More formally, recall that our reward definition of a meta arm $\mathbf{p}$ is $\sum_{k=1}^K p_k \hat{r}(p_k)$ where $\hat{r}(p_k)$ is the importance-weighted (weighted by $p_k$, see eq (5)) reward for base arm $k$. Therefore, although importance weighting might bump up the variance of the weighted reward $\hat{r}(p_k)$, it is also discounted by $p_k$ in our reward estimation, so the convergence of the empirical estimation was not heavily affected by the reweighting. Exploring whether we can utilize the technique of EXP3-IX to improve our bound would be interesting, however, it is non-trivial since EXP3-IX provides a high-probability regret bound while we concern the regret bound in expectation.
>
> - **Regret with respect to the dynamic policy**: We agree with the reviewer that it would be nice to define the regret with respect to the optimal dynamic policy. However, to the best of our knowledge, calculating the optimal dynamic policy in our setting is challenging as it requires solving an MDP with continuous states. So in this work, we focus on the regret definition that is competing with the best fixed mixed strategy. Moreover, under some conditions (Definition G.1), the optimal dynamic policy is the same as best fixed policy in hindsight. We think considering a stronger regret definition would be an interesting and important future work. \
> \
> [1]. Joulani, Pooria, Andras Gyorgy, and Csaba Szepesvári. "Online learning under delayed feedback." International Conference on Machine Learning, 2013.\
> [2]. Pike-Burke, Ciara, et al. "Bandits with delayed, aggregated anonymous feedback." International Conference on Machine Learning, 2018.
> ----------
>
> Please let us know if you have any further concerns about the paper. We are happy to answer more questions.

---

### Official Review · Reviewer_8ugy · 2021-08-03

**Rating:** 7
**Confidence:** 3

**Summary:**

This paper investigates a bandit variant with the changing rewards. The rewards of actions depend on history information, which is very interesting and practical. The idea of combining epsilon-cover (to discretize the action space) and CMAB to design a learning algorithm is very novel. They provide an instance-independent regret upper bound and also a matching regret lower bound, which means their proposed learning algorithm has an order optimal regret guarantee.


**Limitations And Societal Impact:**

I do not see too much discussion about the negative impact of their proposed setting. However, I think the framework proposed has a positive impact on society such as how to dispatch police resources to reduce the crime rate.

**Main Review:**

The paper is quite organized and written. It has several motivating applications included, which contributes to understanding the learning problem. I have some questions.

I am wondering for some special problem instances, is  it possible to have instance-dependent regret bound (takes the O(1/Delta) form) and a matching regret lower bound?

Regarding the content just above Section~5, I still cannot understand why EXP3 would fail. Can authors clarify more?

Regarding the learning algorithm, is there any computational issue to find epsilon covers? If yes, is it possible to have advanced algorithm that does not have any computational issue?


**Time Spent Reviewing:**

6 hours

---

> ### Author Response · Authors · 2021-08-10
> **Response to Reviewer 8ugy**
>
> We sincerely thank the reviewer for the comments and feedback. We also appreciate that you find our paper well motivated and well written.
>
> ----------
> - **Instance-dependent bound**: We thank the reviewer for bringing this up. When proving our instance-independent regret bounds, we have derived the instance-dependent bounds for both action-dependent bandits (Lemma D.4 in Appendix E) and history-dependent bandits (Lemma E.5 in Appendix E), and they do take the form of $O(1/\Delta)$. We only present the instance-independent bounds in the main text due to space constraint.  We do not have matching instance-dependent regret lower bound, but it is definitely an interesting research question for future work.
>
> - **EXP3**: We would like to clarify that the argument is for applying EXP3 over base arm space:  If we apply EXP3 on the base arm space, it suffers linear regrets as it is competing w.r.t best fixed base arm. One alternative is to apply EXP3 on the meta-arm space. However, since there are exponentially many (over K) meta arms, the regret will be exponential in K, which is not desirable. We will make the discussion more clear.
>
> - **Computation of epsilon covers**: The choice of epsilon can be computed efficiently (see Theorem 5.1). We believe there is no computational issue here for finding the value of epsilon.
> ---------
>
> Please let us know if you have any further concerns about the paper. We are happy to answer more questions.

---

### Decision · Program_Chairs · 2021-09-28

**Decision:**

Accept (Poster)

**Comment:**

This paper introduces a novel bandit setting which is certainly of contemporary interest; the delayed impact of actions can well be an important consideration in applications like police patrol policies, and getting this right is important in light of the potential for negative societal impact. The reviewers were uniformly positive on this work. The authors clearly explain why direct application of previous approaches will not achieve the optimal regret, and they successfully devise an algorithm based on a simple intuition (repeatedly pulling the same meta-arm until rewards stabilize) which provably achieves the optimal regret. As this setting is new, another important contribution is proving a lower bound, which allows the authors to claim optimality.

I would like to briefly mention a few minor weaknesses, in the hopes that the camera-ready version of the paper can be stronger. First, essentially nothing is said in the main text about how the lower bounds are proved, and I agree with reviewer ZiMd that the paper's main text would benefit from including the intuition behind the lower bound. Also, similar to reviewer ZiMd, I found the assumption that it is the probability of choosing an arm that impacts the evolution of the arm's reward as odd in some situations. The authors' response to this point, about security games where the probabilities are revealed, does work, but this only makes sense for certain applications. Therefore, I suggest that the authors provide more justification (or at least discussion) of why it is the probability (and not the pulling of an arm itself) which influences the evolution of an arm’s reward.

Overall, this paper provides a very strong contribution and should be of interest not only to researchers in online learning but also the broader community given the motivation for this work. This paper would be a welcome contribution to the NeurIPS 2021 proceedings.

**Consistency Experiment:**

NeurIPS has a long history of experimentation. In 2014, NeurIPS ran an experiment in which 10% of submissions were reviewed by two independent committees to quantify the randomness in the review process. This year, we repeated a variant of this experiment to see how the quality of the review process has changed over time.  This paper was part of the experiment and was therefore assigned to two committees (consisting of reviewers, an Area Chair, and a Senior Area Chair) that reached independent decisions.  If both committees made the same recommendation, this recommendation was followed. If a single committee recommended acceptance, the paper was accepted (with the exception of a few cases in which the other committee identified what we considered a fatal flaw, e.g., an error in a key result).

This copy’s committee reached the following decision: **Accept (Spotlight)**

The other committee assigned to the paper recommended **Accept (Poster)**.  You can find the other set of reviews, along with any follow up discussion with the authors here:
https://openreview.net/forum?id=4QrgRSAAroI